# First results of Antarctic sea ice type retrieval from active and passive microwave remote sensing data

Christian Melsheimer[1], Gunnar Spreen[1], Yufang Ye[2], and Mohammed Shokr[3]

[1]Institute of Environmental Physics (IUP), University of Bremen, Germany
[2]School of Geospatial Engineering and Science, Sun Yat-Sen University, Zhuhai, China
[3]Environment and Climate Change Canada, Toronto, Canada

**Correspondence:** Christian Melsheimer (melsheimer@uni-bremen.de)

**Abstract.**

Polar sea ice is one of the Earth's climate components that has been significantly affected by the recent trend of global warming. While the sea ice area in the Arctic has been decreasing at a rate of about 4% per decade, the multi-year ice (MYI), also called perennial ice, is decreasing at a faster rate of 10%–15% per decade. On the other hand, the sea ice area in the
Antarctic region was slowly increasing at a rate of about 1.5% per decade until 2014 and since then it has fluctuated without a clear trend. However, no data about ice type areas are available from that region, particularly of MYI. Due to differences in physical and crystalline structural properties of sea ice and snow between the two polar regions, it has become difficult to identify ice types in the Antarctic. Until recently, no satellite retrieval scheme was ready for monitoring the distribution and temporal development of Antarctic ice types, particularly MYI, throughout the freezing season and on time scales of several
years. In this study, we have adapted a method for retrieving Arctic sea ice types and partial concentrations using microwave satellite observations to fit the Antarctic sea ice conditions. The core of the retrieval method is a mathematical scheme that needs empirical distributions of the microwave brightness temperature and backscatter input parameters for the different ice types. The first circumpolar, long-term time series of Antarctic sea ice types; MYI, first-year ice and young ice is being established, so far covering years 2013–2021. Qualitative comparison with (a) synthetic aperture radar data, (b) with charts of the development
stage of the sea ice, and with (c) Antarctic polynya distribution data show that the retrieved ice types, in particular the MYI, are reasonable. Although there are still some shortcomings, the new retrieval for the first time allows insight into the interannual evolution and dynamics of Antarctic sea ice types. The current time series can in principle be extended backwards to start in the year 2002 and can be continued with current and future sensors.

## 1 Introduction

As an important component of the global climate system, sea ice affects and reflects changes in other climate components, controls energy and gas fluxes between ocean and atmosphere in polar regions, and it is an important part of the polar marine ecosystem. The Arctic sea ice extent has decreased by 4.1% per decade in the past three decades (Parkinson and Cavalieri, 2012b), while the declining rate for multiyear ice (MYI), ice that has survived at least one summer melt (item 2.6, 2.6.x, JCOMM Expert Team on Sea Ice, 2015), is much higher, 10–15% per decade (Johannessen et al., 1999; Comiso, 2012;

Meredith et al., 2019). This ice, which is also called perennial ice, is the counterpart of seasonal ice which is defined as ice that forms in the beginning of the freezing season and melts in the following summer. Seasonal ice includes young ice (YI), which has thickness less than 30 cm (items 2.1–2.4 in JCOMM Expert Team on Sea Ice, 2015) and the more common first-year ice (FYI), which is thicker than 30 cm (items, 2.5.x in JCOMM Expert Team on Sea Ice, 2015). Three decades ago, MYI covered two thirds of the Arctic Basin. This portion has dropped to one third recently and the MYI has been replaced by FYI (Kwok, 2018).

In contrast to Arctic ice, sea ice extent in the Antarctic has increased by 1.5% per decade since the 1970s until 2014 (Parkinson and Cavalieri, 2012a). Since then, it has strongly decreased and then partly rebounded, so it is too early to quantify any new trend (cf. Ludescher et al., 2019; Parkinson, 2019; Parkinson and DiGirolamo, 2021). Only a small fraction of the sea ice in the Antarctic survives the summer and hence becomes MYI; it is mostly found in the Weddell Sea, but also on the Western side of the Antarctic Peninsula and in small patches around the coast (Stocker et al., 2013).

Almost all MYI in the Antarctic is in fact second-year ice (SYI) because it will usually have drifted out to lower latitudes and melt after at most 2 years. However, it is difficult to discriminate between SYI and older ice using satellite data directly (i.e., not using multi-annual satellite observation or drift data based on multi-temporal satellite imagery). For the remainder of this study, the term MYI usually denotes Antarctic second year ice. While the general distribution and yearly cycle of the different sea ice types in the Antarctic are known to some extent, details, the interannual variations and possible long-term trends are still unclear. The MYI area of the Antarctic may have followed an increasing trend, based on the fact that in the Austral summer the sea ice extent of the Antarctic has increased by 3.6% per decade from 1979 to 2010 (Parkinson and Cavalieri, 2012a); this positive trend in summer is also confirmed – until 2014 – by Parkinson and DiGirolamo (2021). As this is the sea ice which is becoming MYI, one might assume that the MYI area overall increased at a similar rate. In the absence of information on the sea ice type distribution and evolution, the ice mass balance and the heat flux between ocean and atmosphere are still unknown. MYI also is a good proxy for the total ice and snow thickness and thereby the MYI distribution influences the heat flux and serves as an indicator or response to climate forcing.

In the Antarctic, sea ice cover, and particularly MYI, is different in many ways from Arctic ice. There is commonly a thicker snow cover, which may cause the underlying sea ice to be depressed below the water surface and thus the snow-ice interface to be flooded, creating so-called flooded ice. The slush layer resulting from flooding of the basal snow layer with sea water eventually refreezes, resulting in what is called snow ice. In addition, water from partial snow melt that percolates down and refreezes at the bottom of the snow layer forms so-called superimposed ice, which is more common in the Antarctic than in the Arctic (Haas et al., 2001). The ice cover in the Antarctic is also rougher than that in the Arctic because it is exposed to more variable wind conditions and higher wind, which triggers more motion and collisions between ice floes (Wadhams et al., 1986). The turbulent ocean water leads to formation of frazil ice crystalline structure, which is another difference compared to the mostly congealed structure of the Arctic sea ice (Gow et al., 1987; Lange and Eicken, 1991). For these reasons, statistical distributions of radiometric and backscattering observations are expected to be different between the two regions.

Most of the Antarctic MYI is found in the Weddell Sea. The Weddell Gyre transports the MYI away from the coast (where it has survived the summer), towards the north-western and northern Weddell Sea where eventually most of it melts. In turn,

seasonal ice is transported into the Weddell Sea from the north and northeast, can be pressed and compacted against the ice shelves and the coast of the Antarctic peninsula where it survives the summer ad becomes MYI. Spatial and temporal observations of the MYI distribution are needed to better understand these processes. Beside MYI, the other ice types are of interest as well. Along the coasts large landfast ice areas develop regularly (Nihashi and Ohshima, 2015). Note that a substantial amount of the MYI along the East Antarctic coast is actually fast ice (Massom et al., 2010). This is often true MYI (older than two years) which is of great importance for the ecosystem and has effects on buttressing ice shelves (Fraser et al., 2021). In polynyas, new and YI types alter the mass balance and surfaces fluxes. In the marginal ice zone (MIZ), pancake ice is a common occurrence; a substantial part of new ice is actually formed via pancake ice (Lange et al., 1989). Monitoring the dynamic and vast sea ice cover requires satellite observations. A physically consistent time series of sea ice types is needed to ascertain trends and quantify the interaction of sea ice within the climate system. Currently, climate models are not yet able to correctly reproduce realistic future scenarios of the Antarctic sea ice extent and especially regional patterns are not well reproduced (Mahlstein et al., 2013; Polvani and Smith, 2013; Hobbs et al., 2015; Turner et al., 2015; Zunz et al., 2013). In order to improve and better validate climate models, time series with more detailed information about the sea ice, e.g., sea ice thickness or type, are required. Models usually do not directly produce our sea ice types as output variable, but rather sea ice age – which might well be set into relation with sea ice types like FYI and MYI.

The concentration of total sea ice and of the sea ice types (i.e., the area fraction of ice, in per cent, within an observation cell) can be estimated from microwave satellite observations because in this spectral range different ice types have different emission and scattering properties. This has long ago already been used by sea ice concentration retrieval algorithms using passive microwave (i.e., radiometer) data, like the NASA TEAM algorithm which retrieves FYI and MYI concentration. This, however, has been applied almost exclusively to the Arctic. The reason is that the radiometric signature between MYI and seasonal ice (FYI) in the Antarctic was found to be not well suited to carry out a similar analysis in the Antarctic – at least when using the same channels as in the Arctic. Besides, the retrieved Arctic MYI area tends to increase during the cold season which is not possible (see Section 3.2). Sea ice type discrimination based on analysing high-resolution active microwave (i.e., synthetic aperture radar) data usually lacks full and daily coverage. Using radar scatterometer data might give full and frequent coverage, but has ambiguities. There are very few attempts to retrieve sea ice types in the Antarctic, notably by Lythe et al. (1999) and Ozsoy-Cicek et al. (2011), but they are only of regional scope.

Recently, a method has been developed to estimate partial and total concentration of Arctic ice types using a combination of active and passive microwave satellite measurements (i.e., from scatterometer and radiometer, respectively) and additional ancillary data of air temperature and ice drift vectors. The method is based on Environment Canada's Ice Concentration Extractor algorithm, ECICE (Shokr et al., 2008; Shokr and Agnew, 2013) and subsequent correction schemes for mitigating misclassifications caused by melt-refreeze cycles and by snow metamorphosis. These correction schemes use surface temperature from meteorological reanalysis and ice drift from satellite data. This has been successfully applied and tested in the Arctic (Ye et al., 2016a, b) and has recently also been compared to other sea ice type retrieval results (Ye et al., 2019). In this study, we show that this method (i.e., ECICE and the correction schemes) can be adapted to Antarctic conditions. The study is first intended as a "proof of concept", followed by a first look at the interannual evolution (until now unknown) of Antarctic sea ice types since

95 2013. The ultimate goal, however, is of course to eventually fill the data gap in the Antarctic – the lack of ice type, in particular, multiyear ice, data. The only other approach of operationally retrieving sea ice type in the entire Antarctic from remote sensing data is the ice type classification based on microwave radiometer data, recently (summer 2021) released by the Ocean and Sea Ice Satellite Application Facility (OSI-SAF) (Aaboe et al., 2021a).

This paper is organised as follows: In Section 2 we give a brief account of ECICE, the implemented correction schemes, 100 and the adaptation to the Antarctic conditions. In Section 3, results of the Antarctic sea ice type concentration mapping are compared with data from other sources like Sentinel-1 radar images or ice charts, followed by critical discussion of the findings and the first time series of Antarctic MYI, 2013–2021. Section 4 presents a summary and conclusions.

## 2 Estimation of Sea Ice Type Concentration

Satellite-based estimation of the total ice concentration and of partial concentration of different ice types like MYI, FYI and 105 YI can be obtained using the Environment Canada's Ice Concentration Extractor (ECICE). It takes input from any set of satellite observations and produces concentrations of any given ice types (Shokr et al., 2008), and was originally developed for the Arctic. Our estimation of MYI concentration actually is a two-step procedure that first uses ECICE and then applies two correction schemes to the output MYI concentration in order to account for anomalies of the ECICE results (Ye et al., 2016a, b, originally developed for the Arctic as well). One anomaly causes misclassification of MYI as FYI (usually observed 110 in autumn) and the other causes misclassification of FYI as MYI (usually observed in spring).

### 2.1 ECICE – Environment Canada's Ice Concentration Extractor

ECICE can take passive and active microwave satellite measurements as input. Possible passive microwave input data are from the satellite microwave radiometers Special Sensor Microwave/Imager (SSM/I, 1987-2009), Special Sensor Microwave/-Sounder and Imager (SSMIS, 2003-2022), Advanced Microwave Scanning Radiometer for EOS (AMSR-E, 2002-2011) and 115 Advanced Microwave Scanning Radiometer 2 (AMSR2, 2012-present). As AMSR-E and AMSR2 have considerably higher spatial resolution than SSM/I and SSMIS, we have used only their data so far. The measured quantities are the brightness temperatures at different frequencies and at vertical (V) as well as horizontal (H) polarisation. Possible active microwave input data include scatterometer measurements from QuikSCAT (1999-2009) and Advanced Scatterometer (ASCAT, 2007-present). The measured quantity is the backscattering coefficient (normalised radar backscattering cross section, NRCS) at one or two 120 polarisations (HH and VV). The number of input parameters to ECICE must be equal or greater than the number of the surface types to be distinguished. Here we use four surface types; namely open water, young ice (YI), first-year ice (FYI), and multiyear ice (MYI). The input parameters are listed in Table 1 and explained in Section 2.3.

Most methods that retrieve the concentration of sea ice or sea ice types from radiometer or radar data use representative values of each input parameter for each surface type (ice or ice types, open water) – these representative values are known as tie 125 points (see, e.g., the 11 sea ice concentration algorithms compared by Ivanova et al. (2015): All but one (ECICE) use tie points, including all "standard" ones like NASA Team or Bootstrap algorithms). In contrast, ECICE uses the statistical distribution

of all possible values of each input parameter for each surface type. Such distributions are obtained by sampling data of the given input parameter from the given surface obtained under different meteorological, dynamic and freezing conditions. The distributions have been established for the application of Arctic sea ice (Shokr et al., 2008; Ye et al., 2016a) and re-established here for the Antarctic application (see details in Section 2.3). Note, however, that under permanent melting conditions in summer, the radiometric and backscattering properties of sea ice change drastically and the differences between the ice types diminish or even vanish (Lindell and Long, 2016). Therefore, sea ice type retrieval with ECICE is not possible in the melt season.

With the input parameter distributions for the different surface types, which can be interpreted as probability densities, a number $n$ of possible realisations of tie points for all surfaces is selected using a random number generator. Here, we use 1000 realisations. For any given observation in remote sensing data, the observation is considered to represent a linear mixture of typical values (tie points) for each surface type, weighted by the area fraction of that surface type in the observation cell. Therefore, a set of linear equations (equal to the number of observations) in which each equation represents decompositions of each input observation into its components from the given surface types is then constructed. The equations are solved simultaneously for the area fraction of each surface type under the constraint that all fractions must add up to one (equality constraint) and that each area fraction must be between 0 and 1 (inequality constraint). This formulates the problem into an inequality constrained optimisation problem, which is solved to find the ice concentrations that minimise a given cost function. Then, the median of the $n$ solutions for the area fractions (i.e., concentrations) is produced from which the final answer is generated: a set of concentrations of the given ice types, each of them between 0% and 100%, adding up to the total ice concentration, and always adding up to 100% when the open water fraction is included. In addition to the ice type concentrations, the spread of the $n$ solutions around the median is used as a measure of confidence of the result for each surface type (see details in Shokr et al., 2008), and is saved along with the output. To summarise briefly: First we calculate, for each surface type, the mean absolute deviation, $MAD$, of the $n$ solutions from the median. Then the confidence level is

$$CL = 1 - \frac{MAD}{AD_{max}} \tag{1}$$

where $AD_{max}$ is the maximum absolute deviation of the $n$ results from the median. The confidence level is between 0.0 (meaning: almost all results have large deviation from the median) and 1.0 (meaning: all results are identical).

## 2.2 Correction schemes

The determination of the ice type concentrations (i.e., the area fractions of the three ice types) is only based on their radiometric and backscattering properties. During atmospheric warm spells, commonly occurring in the transition seasons, snow wetness develops. The return of cold temperatures may cause snow metamorphism, and even without warm spells, snow metamorphism can occur over time. Both effects, snow wetness and snow metamorphism, cause anomalous microwave observations that make MYI appear as FYI and vice versa. Another process that causes anomalous observations is ice surface deformation at floe edges (e.g., pancake ice), which makes brightness temperature and backscattering from FYI look similar to those from MYI. Such errors are reduced by the two corrections schemes described in the following sections.

### 2.2.1 Temperature Correction

Warm air advection can occur during fall and spring seasons. When air temperature rises to near melting or beyond melting conditions, snow wetness develops and MYI will have lower backscatter and higher brightness temperature, typical of FYI. Therefore, it is misclassified by ECICE as FYI (more precisely: the retrieved MYI concentration is too low and the FYI concentration too high). After the end of the warm event, which typically takes between one to a few days, the correct classification is resumed. In order to account for this error, the so-called temperature correction scheme (Ye et al., 2016a) examines the 2-m air temperature (T2m) data to identify warm episodes of up to $N$ days. Each episode starts with temperatures rising above a threshold $T_1$ (near freezing temperature) and ends with temperatures falling below a threshold $T_2$. If the MYI concentration drops at any location affected by the warm spell by more than a specified threshold $\Delta C_t$ and later rises again, such MYI concentrations are replaced by values linearly interpolated from before and after the warm episode. The values of the "tuning" parameters $N$, $T_1$, $T_2$ and $\Delta C_t$ used here are explained and specified in Section 2.3 (Table 2).

### 2.2.2 Drift Correction

Since the warm temperatures in the spring may progress for the rest of the season, the above temperature correction may not hold because it depends on the returning to the normal cold winter temperatures. For this reason, another correction was developed to identify locations of estimated MYI which are not realistic. This correction is based on the fact that MYI starts by definition as all remaining ice at the end of the melting season, i.e., at the onset of freeze-up. After that, during the cold season, no new MYI can be generated. MYI can then only drift, and its concentration can only be changed by divergence, convergence, and melting. Therefore, MYI is unrealistic if it appears at locations to which it cannot have drifted. To identify such locations, daily ice drift data are used to implement what is called "drift correction" (Ye et al., 2016b). This correction starts with defining the boundary of the MYI cover from the map of a given day, using a threshold of 20% MYI concentration. The boundary is then adjusted according to the ice drift map of the same day to predict its contour on the next day. This is done by applying ice motion vectors (obtained from the sources mentioned below) to all pixels inside the boundary. This domain is then further extended by one grid cell to the outside in order to account for uncertainty of the ice drift product. Any MYI that has been retrieved outside of this domain cannot be multiyear ice but is FYI or YI that has radiometric and scattering properties that resemble MYI because of deformation, snow metamorphosis, other "ageing" processes of the ice, or because of a thick snow layer. Therefore, all non-zero MYI concentrations in grid cells outside the mentioned domain are set to zero. However, this spurious MYI is in fact FYI or YI that has anomalous radiometric and scattering properties (so that ECICE has classified it as MYI), but there is no way of telling which of he two (YI or FYI) it actually is. Therefore, we keep as a new pseudo-ice-type which we call non-MYI[1]. The corrected MYI concentration is then used to construct, together with ice motion vectors, the domain for the correction of the next day. Note that on the first day of the drift correction, there is no corrected MYI from the day before, so we have to use the MYI concentration without drift correction, but with temperature correction. Therefore the drift correction should start after freeze-up, early in the freezing season: At that time, the mentioned processes that make FYI

---

[1]If we just had two ice types, MYI and seasonal ice, we could simply add the non-MYI to the seasonal ice.

**Table 1.** Input parameters ("channels") used in Antarctic sea ice type retrieval with ECICE. Note: $T_B$ is brightness temperature, $GR$ is gradient ratio (see Equation (2)), $\sigma^\circ$ is normalised radar backscattering cross section (NRCS).

| Instrument | Parameter | Freq. [GHz] | Polarisation |
|------------|-----------|-------------|--------------|
| AMSR2 | $T_B$ | 37 | V |
| AMSR2 | $T_B$ | 37 | H |
| AMSR2 | $GR$ | 19, 37 | V |
| ASCAT | $\sigma^\circ$ | 5.3 | VV |

look like MYI permanently can be expected to be still weak, therefore, the correction for these processes, which is the drift correction, is not very important yet. This means the result (the corrected MYI concentration) should not depend strongly on the exact beginning date for the drift correction.

In addition, this correction scheme includes correction for some effects of snow metamorphosis for pixels inside the MYI contour of the given day which would make FYI radiometrically similar to MYI. It looks for sudden (within one day) rises, $\Delta C_d$, of MYI concentration concurrent with sudden reductions, $\Delta T_{37}$, of $T_{B,37V}$ (brightness temperature at 37 GHz, vertical polarisation) or reductions, $\Delta T_{19-37}$, of $T_{B,19H} - T_{B,37H}$ (difference of the brightness temperatures at 19 GHz and 37 GHz, horizontal polarisation). The latter difference is also called horizontal range, HR, and is used by Drobot and Anderson (2001)

to identify the onset of snow melt. The use of this parameter in the drift correction is explained in detail in Ye et al. (2016b). In cases of such anomalies, the MYI concentration at the given pixel is replaced by the value of the previous day. The values of the "tuning" parameters $\Delta C_d$, $\Delta T_{37}$, and $\Delta T_{19-37}$ used here are explained and specified in Section 2.3 (Table 3). A final note on the name of this correction scheme: we call it drift correction here because it uses sea ice drift data to correct for the effect of snow metamorphism and ice deformation and other processes that make YI and FYI resemble MYI for the ECICE retrieval.

It does not correct for drift effects.

A flow chart showing ECICE, the two correction schemes and the various types of input data is shown in Figure 1.

## 2.3   Adapting ECICE algorithm to the Antarctic sea ice

In order to use ECICE and the correction schemes for the Antarctic, the algorithms themselves do not need to be adapted. Instead, for ECICE, we need input parameter distributions derived from Antarctic data because Antarctic sea ice is different

from Arctic sea ice (see Section 1). For the correction schemes, we might need to adapt the tuning parameters. The Antarctic implementation of ECICE at the IUP, University of Bremen, uses as input microwave radiometer data of the sensors AMSR-E (Advanced Microwave Scanning Radiometer for EOS) on the NASA satellite Aqua (2002–2011), or AMSR2 (Advanced Microwave Scanning Radiometer 2) on the JAXA satellite GCOM-W1 (since 2012), and scatterometer data from ASCAT (Advanced Scatterometer) on the European polar-orbiting satellites MetOp-A, MetOp-B, and MetOp-C. The input parameters

("channels") we use are the same as for the Arctic sea ice type retrieval (Ye et al., 2016a, b), they are listed in Table 1. The

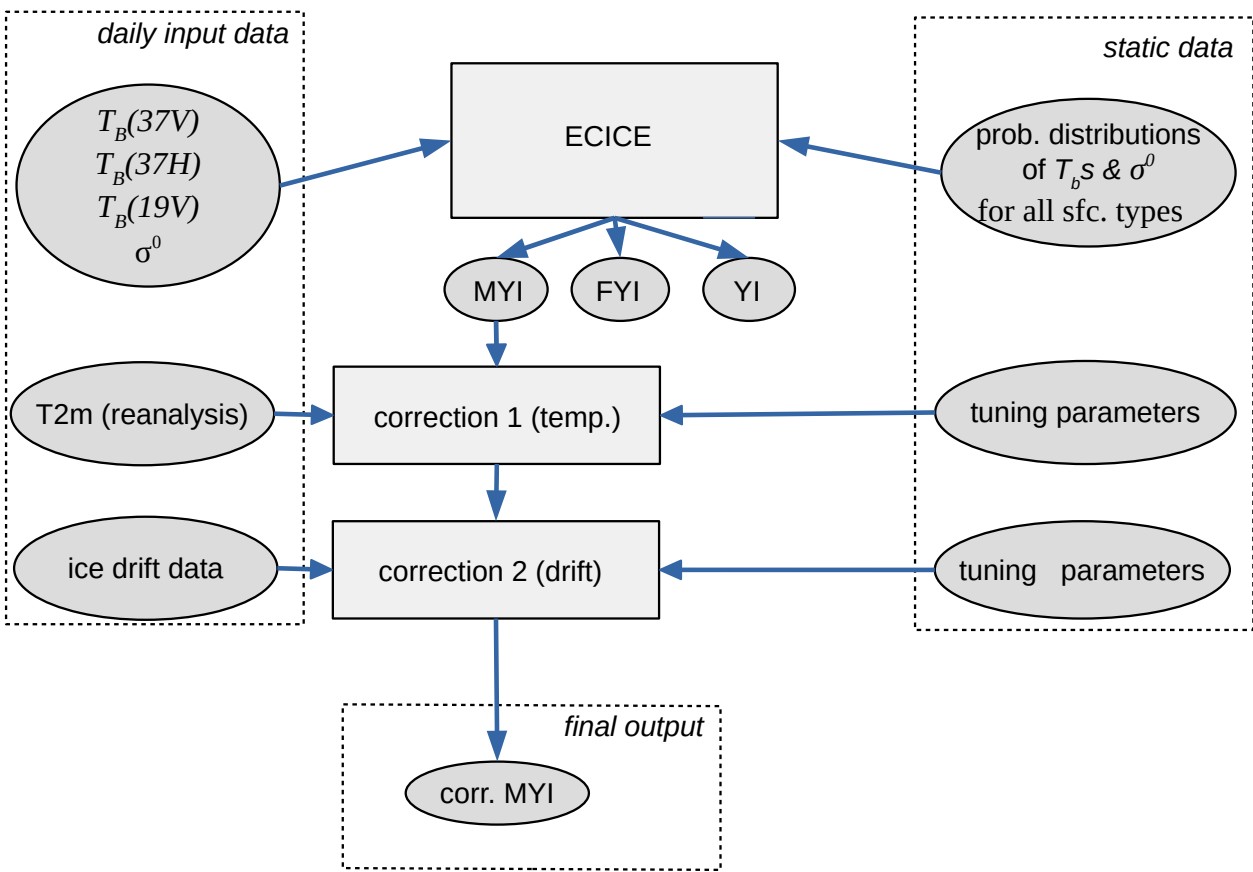

**Figure 1.** Flowchart of ECICE and the two correction schemes as well as the different types of input data. Note: prob. = probability; sfc. = surface

gradient ratio at 19 and 37 GHz, V polarisation is defined as

$$GR(37V, 19V) = \frac{T_{B,37V} - T_{B,19V}}{T_{B,37V} + T_{B,19V}} \qquad (2)$$

For all input parameters, we use daily gridded data, projected on a polar stereographic grid with a nominal resolution of 12.5 km. This is the common grid used by the National Snow and Ice Data Center, NSIDC (see more details in Melsheimer and Spreen, 2019). The grid spacing is close to the resolution and the sampling interval of the original swath data of the used AMSR2 channels, which is 10 km (note that the footprint size is 7 km by 12 km at 36 GHz and 14 km by 22 km at 19 GHz). For the gridding, we combine all swaths of one day and then interpolate to the target grid (NSIDC polar stereographic) using a distance-weighted near-neighbour approach with four sectors, from the Generic Mapping Tools (GMT, Wessel et al., 2013). Before the gridding, the ASCAT data (normalised radar backscatter cross section, $\sigma°$) are converted to a common incidence

angle of 40°, using a simple linear approach for the dependence of $\sigma^\circ$ on the incidence angle, based on a regression analysis of $\sigma^\circ$ values over sea ice (for the details see Appendix C).

In order to derive the probability distributions of the input parameters needed by ECICE, we have to find appropriate sample areas for three ice types, i.e., MYI, first-year ice (FYI), and young ice (YI). The sample areas need to be pure ice types, i.e., 100% MYI, FYI, or YI, respectively, and need to cover more than a few grid cells of 12.5 km by 12.5 km, so their sidelength

or diameter should rather be several tens of kilometres. As such information is almost lacking for FYI and MYI, we have resorted to the time evolution of total sea ice from the operational ASI sea ice concentration retrieval[2] in the year 2018: At the beginning of the cold season, all remaining ice (mainly in the Weddell Sea) is MYI by definition. For this purpose, we have observed the daily ASI sea ice concentration maps in February and March 2018 and define the beginning of the cold season, regionally, as the time when the sea ice extent starts to grow and the ice concentration stays high. It is not possible

to define one day as the exact day of the minimum extent in one region, but this is not needed here. After a few days, the signal of regrowth of ice, i.e., increasing extent at high ice concentration is very clear (to a human observer), so for a short period, before possible drift and divergence with new ice formation blur the picture, areas of MYI can be identified. This means, however, that with this approach MYI sample areas can only be found at the beginning of the freezing season, so, for the time being, we do not have samples throughout the season. Later in the season, sea ice that has formed in the season

"away" from the MYI is FYI – taking into account, of course, that MYI can drift about 10 km per day. For reliable samples of YI, we have used a satellite-based polynya data set from 2018 which is based on the Polynya Signature Simulation Method (PSSM, Markus and Burns, 1995) in the implementation by Kern et al. (2007), using combinations of the 36 GHz and 91 GHz, H and V polarisation channels of the Special Sensor Microwave Imager/Sounder (SSMIS). We use the NRT data provided by ICDC/CEN, University of Hamburg (see details at https://www.cen.uni-hamburg.de/en/icdc/data/cryosphere/

polynya-antarctic.html), region "Western Ross Sea". The data contain a surface class "thin ice", up to 20 cm thick, which was used here for YI (note, however, that it could be thicker ice but lower ice concentration). For the surface type "open water", sample areas in the Southern Ocean in August 2018 and in the ice-free part of the Ross Sea in March 2018 were taken. They were large enough (about 600 km and 250 km sidelength) and taken during a period of several days that they cover a wide range of atmospheric (e.g., wind, precipitation) and oceanic (e.g., surface waves) conditions. Details of all the sample areas are listed

in Appendix A. The resulting distribution functions of the four input parameters for the surface types YI, FYI, MYI and open water are shown in Figure 2. It is obvious that the four surface types cannot be distinguished using a single input channel, as the distributions generally overlap. However, in some channels one specific surface type has a peak separated from the others, e.g., the backscatter ($\sigma^\circ$) of MYI (Figure 2(a), red curve), or $GR$(37V,19V) of OW (panel (c), blue curve). In addition, there are pairs of mutually non-overlapping distributions, such as OW and YI for $T_B$(37V) (panel (b), blue and cyan curves), or OW

and FYI for $T_B$(37H) (panel (d), blue and green curves). Comparing the distributions for brightness temperatures at 37 GHz, V and H polarisation ($T_B$(37V) and $T_B$(37H), respectively) to corresponding Southern Hemisphere tie points for the NASA Team and the Bristol algorithms (Table IV in Ivanova et al., 2014) shows that the open water tie points, about 205 K (37V) and 140 K (37H) are contained in the distributions for open water (dark blue curve in Figure 2(b) and (d)), the same is true

---

[2]ARTIST Sea Ice (ASI) algorithm (Spreen et al., 2008) https://seaice.uni-bremen.de/sea-ice-concentration/, Melsheimer and Spreen (2019)

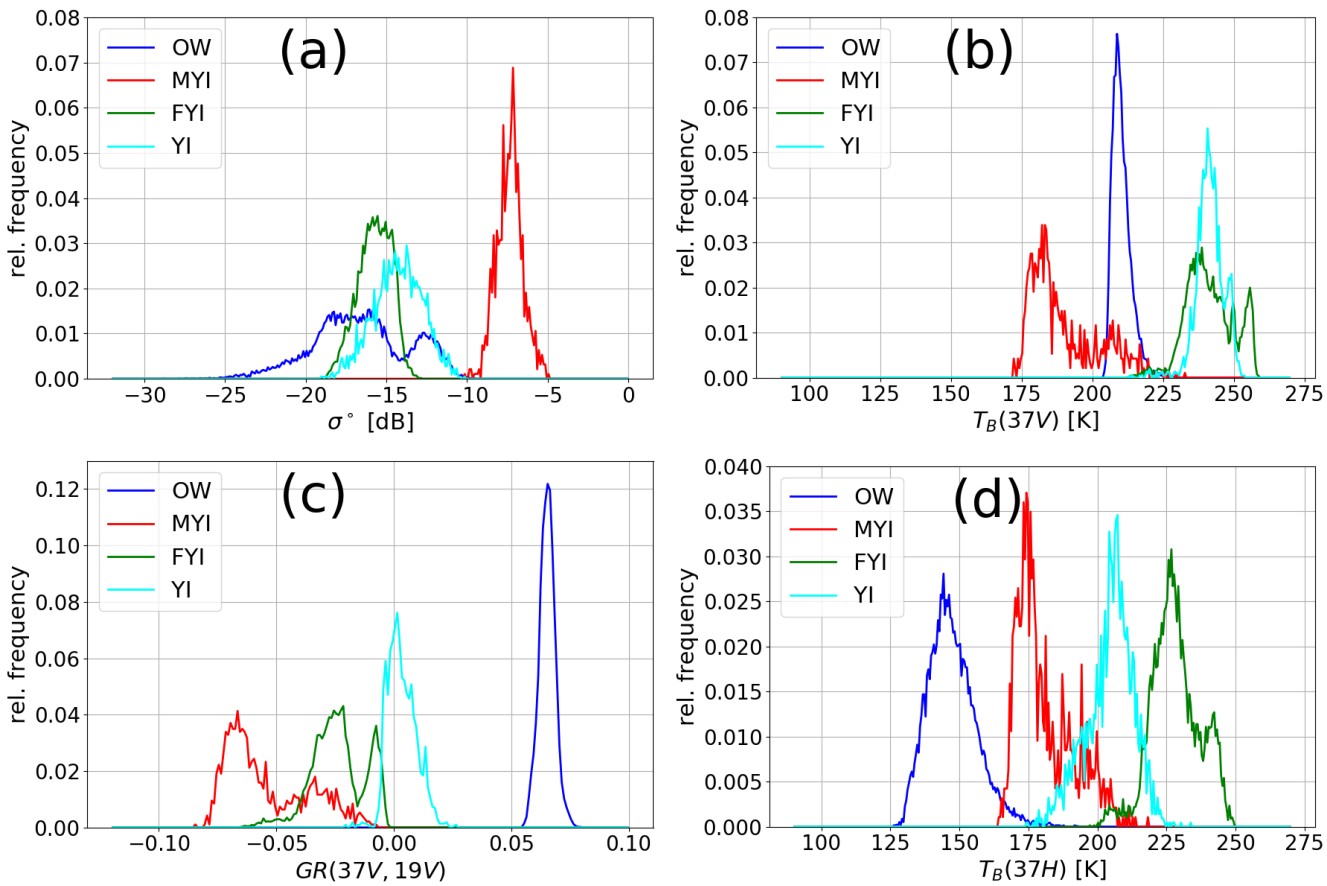

**Figure 2.** Distribution of the four input parameters to ECICE for the Antarctic surface types of MYI (red), FYI (green), YI (cyan) and open water (blue) for the sample areas specified in Appendix A: (a) ASCAT $\sigma°$ (top left); (b) AMSR2 $T_B$ at 37 GHz, V polarisation (top right); (c) AMSR2 $T_B$ at 37 GHz, H polarisation (bottom right); (d) and AMSR2 $GR(37V,19V)$ (bottom left). The total number of samples for OW: 13742, MYI: 964, FYI: 6852, YI: 2168.

for FYI: about 240 K (37V) and 230 K (37H), cf. the green curve in Figure 2(b) and (d). The MYI tie points however, about
190 K (37V) and 180 K (37H), are about 5 to 10 K above the mode of the corresponding distributions (red curves). The MYI
tie points from the Antarctic "roundrobin data package" (see Table A1 in Ivanova et al., 2015) are even higher, 227 K (37V)
and 205 K (37H) and thus at the very top end of our distributions. Note that the distributions are only from samples from the
beginning of the freezing season, so they might not be representative for the whole season (see below, end of Section 3.2). As
to the C-band NRCS (Figure 2(a)), MYI has values between about -10dB and -5dB, which is higher than values reported for
MYI by, e.g., Arndt and Haas (2019). The time series of the NRCS of MYI in that study also shows that the highest values
occur in autumn and early winter (roughly February to April), which is the time for which our MYI samples were taken.

**Table 2.** Tuning parameter setting for the **temperature correction** scheme

| Parameter | Meaning | Value |
|---|---|---|
| $T_1$ | thresh. temp., start of warm episode | -2°C |
| $T_2$ | thresh. temp., end of warm episode | 1°C |
| $N$ | max. duration of warm episode | 10 days |
| $\Delta C_t$ | min. drop of MYI conc. | 10% |

**Table 3.** Tuning parameter setting for the **drift correction** scheme

| Parameter | Meaning | Value |
|---|---|---|
| $\Delta C_d$ | min. rise of MYI conc. | 20% |
| $\Delta T_{37}$ | min. drop of $T_{B,37H}$ | 20 K |
| $\Delta T_{19-37}$ | min. drop of $T_{B,19H} - T_{B,37H}$ | 10 K |

For the correction schemes, we use lowest-level (2 m) air temperature data from meteorological reanalysis of the European Centre for Medium-Range Weather Forecast (ECMWF), namely, from the ERA Interim data set (Dee, 2011). Note that such reanalysis data are the only source of comprehensive and consistent temperature data for the Antarctic sea ice areas (there are several different reanalysis products). We further use the low resolution sea ice drift product of the EUMETSAT Ocean and Sea Ice Satellite Application Facility (OSI SAF, www.osi-saf.org, Lavergne et al., 2010). Note that sea ice motion data from the National Snow and Ice Data Center (NSIDC) (Tschudi et al., 2016, 2020) can also be used, and have actually already been used for the MYI retrieval in the Arctic (Ye et al., 2016b). At the time this study was started, the OSI-SAF drift data seemed the best choice, but the new NSIDC version 4 ice drift product might be an alternative. Sea ice drift data for the Antarctic are, however, less reliable than for the Arctic (Lavergne et al. (2021) found larger standard deviation in the Antarctic than in the Arctic when comparing ice drift from passive microwave data with buoy data).

The used values of the tuning parameters for the temperature correction, $N$, $T_1$, $T_2$, $\Delta C_t$ (Section 2.2.1) are listed in Table 2, and the used values of the tuning parameters of the drift correction, $\Delta C_d$, $\Delta T_{19-37}$, and $T_{B,37H}$ (see Section 2.2.2) are listed in Table 3. For the time being, we have used the values that were used for the Arctic and fine-tuning might still improve the results, but for a "proof of concept" as which this study is intended this is sufficient. Besides, other issues might need attention first (see below, Section 3.2, "Discussion").

The final result of the two-step retrieval method (i.e., ECICE and the corrections schemes) are MYI map data. In addition, as an intermediate result, a preliminary distinction of the sea ice into three types, i.e., MYI, FYI and YI (i.e., the "pure" ECICE output, without applying any correction) is produced as well. Hereinafter it is called "uncorrected YI, FYI, MYI concentration".

To date, we have retrieved data for the cold seasons 2013-2021. AMSR2 has been operational since August 2012, but since we need the start of the cold season for applying the drift correction, our retrieval starts with the year 2013.

For each year, we have retrieved the ice type concentrations for the months of February to November (autumn, winter and spring). As the correction schemes need extra time to check for transient changes in MYI concentration and temporary warming events, the corrected MYI concentration is available from 22 February to 8 November each year (roughly the Antarctic freezing season) – see the example for YI, FYI and corrected MYI in Section 3.1.2 (Figure 4). Note that also our MYI at the start of the season is always based on the MYI that is output by ECICE. We do not use the total ice extent of a certain day when freeze-up starts as the starting MYI domain – which would be problematic as there is no single freeze-up day for the Antarctic seas. Starting with ECICE output is more robust, as explained in Section 2.2.2. The AMSR-2 radiometer data as well as the ASCAT data are available within about 1 day of acquisition, thus, the uncorrected ice type concentrations can be retrieved in near real time with a time lag of one to two days. The corrected MYI concentration can be produced with an additional time lag of two weeks because of the corrections schemes, as just mentioned.

Note that retrieving Antarctic ice types in the freezing seasons of 2002-2011 is possible, using AMSR-E instead of AMSR2, and QuikSCAT instead of ASCAT before 2008. QuikSCAT uses a frequency of 13.4 GHz ($K_u$ band) while ASCAT uses 5.4 GHz, so it is not trivial to switch. QuikSCAT, however, has already been used by us for the retrieval of Arctic MYI (see Ye et al., 2016a). For the years before 2013 (start of OSISAF drift data record for Antarctic), the sea ice drift data needed for the correction scheme of the MYI concentration are available from NSIDC (also used by Ye et al. (2016a)).

## 3   Results

### 3.1   Comparison with other data

Validation of sea ice type concentration is a challenging task not only because there are few in-situ data sets of ice types available for the Antarctic region but also because point observations do not represent the large footprints of the satellite data. Validation studies using data from research cruises in the Antarctic have just started recently.

In this study we have compared the sea ice type concentration output from ECICE against three other satellite-based data sets: The new Antarctic uncorrected FYI and YI data and the corrected MYI data have been compared to data from (1) synthetic aperture radar (SAR) images, (2) charts showing the stage of development (SoD) of the ice, (3) maps of thin ice from the polynya data set mentioned above (but from a different year than those used for algorithm tuning). Examples are presented in the following.

### 3.1.1   Synthetic Aperture Radar (SAR) Images

As an initial "sanity check", we have compared the MYI concentration from ECICE with high resolution SAR images acquired by Sentinel-1A/B at 40 m grid resolution (extra wide swath mode at HH polarisation; obtained from https://scihub.copernicus.eu). The focus was on the rather prominent FYI-MYI boundary in the Weddell Sea. In SAR images, MYI usually looks considerably brighter than FYI and YI because, e.g., over time the surface has become more deformed (i.e., rougher) and the ice is more porous because of brine drainage so there is more volume scattering. For the comparison we manually scaled the

radar backscatter to show good contrast between ice classes. We have examined Sentinel-1 SAR scenes from the two days 3 March (5 scenes) and 11 August, 2016 (16 scenes) in the Weddell Sea and near the Antarctic Peninsula. Since the ECICE data represent a daily average (all AMSR and ASCAT swaths of one day are included), no correction for ice drift has been applied between the two data sets. An overview of all scenes and a zoom into an area in the Weddell Sea on 11 August, 2016 is presented in Figure 3. The Sentinel-1 SAR image (grey shade) is overlaid on the MYI concentration (colour scale). In addition, the dominant ice type is indicated by the coloured contours; white/cyan: YI=50%; purple: FYI=50% or 70%; black: MYI=70%.

In the zoom image from 11 August the black and purple contours, which mark the transition from MYI to FYI, coincide well with a clear boundary between bright and dark radar backscatter which marks the boundary between dominating MYI and dominating FYI. Note also that ECICE correctly identifies an area of YI in the lee of the grounded iceberg A23A. The iceberg itself is retrieved as mixed type sea ice as it is outside the 70% of FYI or MYI and outside the 50% contour of YI. Qualitatively, the comparison to the other Sentinel-1 SAR scenes shown in the overviews at the top of Figure 3 is similarly good.

### 3.1.2 Stage of Development (SoD) Charts

Weekly charts of the stage of development (SoD) of the Antarctic sea ice have been jointly produced by the U.S. National Ice Center (NIC) and the Russian Arctic and Antarctic Research Institute (AARI) since June 2015 (*AARI-NIC-NMI pilot project on integrated sea ice analysis for Antarctic waters*, http://ice.aari.aq/antice/). The charts, based on analysis of visible/infrared and microwave satellite imagery, temperature and wind data by experienced specialists (see details in https://usicecenter.gov/Resources/AnalystProcedures ), show various ice types, grouped roughly into YI, FYI, MYI. In this classification, the surface is divided into polygons each of which is assigned a single ice type. Purple tones indicate various types of YI, including nilas and grey ice (thickness up to 30 cm); yellow and green tones indicate various types of FYI (thickness above 30 cm); brown, orange and red indicate MYI. However, no distinction is made between *thick first-year ice* and *residual ice*, i.e., FYI that has survived the summer melt and has started a new cycle of growth. In the Antarctic, it is relabelled *second-year ice* only after July 1 (JCOMM Expert Team on Sea Ice, 2015; WMO, 2014).

We have compared the weekly SoD charts of the cold seasons (March to October) of 2017 and 2018 with our ice type data, and also done sporadic comparisons with data from other years. Some examples are presented below. There is an overall correspondence of the ice types from the charts and the dominant ice type in our data. An example is presented in Figure 4, showing (left to right, top to bottom) the Antarctic SoD chart from AARI (for the week that ends on 30 March, 2017), along with the YI and FYI concentrations from ECICE, and the corrected MYI concentration from ECICE plus correction schemes for 30 March, 2017. The SoD chart has been cropped to save space, the colour legend can be found in the Appendix B (Table B1). Note the YI (purple tones) in the Eastern Weddell Sea, in the Ross Sea and mainly along the Eastern Antarctic coast. The areas of MYI are in the Western Weddell Sea, in the Amundsen Sea and at the coast of Wilkes Land (130°–170°E).

At this early stage of the freezing season, the charts from AARI and from NIC often disagree on the MYI-FYI distinction. As an example Figure 5 shows the Weddell Sea on the same date of 30 March, 2017. The NIC chart (on the right) shows large strips of FYI (yellow) and MYI ("old ice", brown), while the AARI chart (on the left) just shows MYI (and is much closer to the results of our retrieval, see previous figure). From April 2017 on, however, both the AARI and the NIC charts show a large

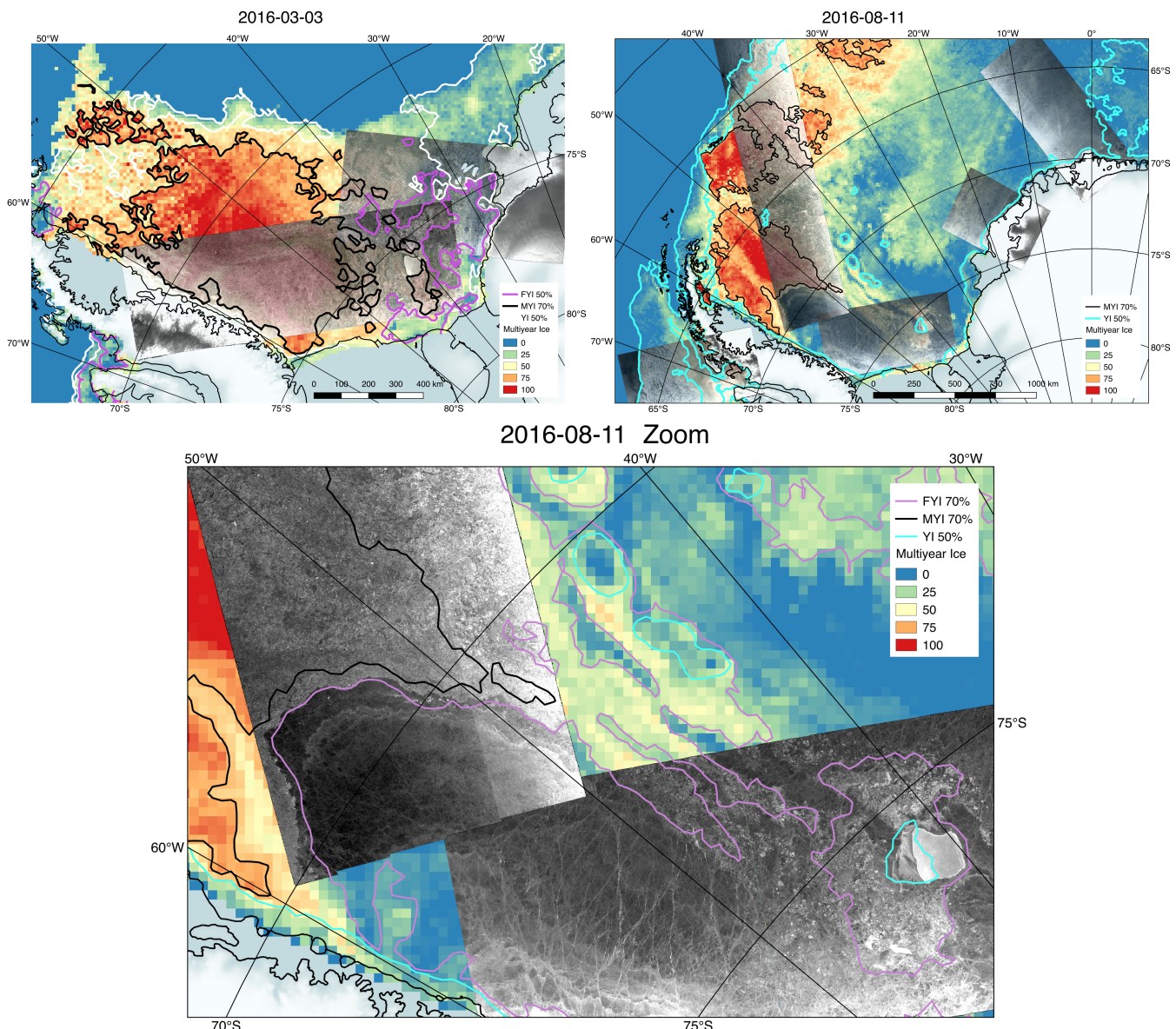

**Figure 3.** Sentinel-1 SAR image (grey shade) acquired on 3 March and 11 August, 2016, overlaid on MYI concentration (colour scale) of the same days from ECICE. At the top an overview of the Weddell and adjacent seas for the two days is shown. the bottom map shows a zoom in to an area is the inner (Southeastern) Weddell Sea, bordering the Antarctic Peninsula (lower left corner of the Figure). Dominant ice type from ECICE are indicated by contour lines. The MYI color legend shows only the most important colours for multiples of 25% concentration.

and compact area of MYI in the Weddell Sea (so the strips of FYI marked in yellow in the NIC charts have been re-labelled MYI in brown).

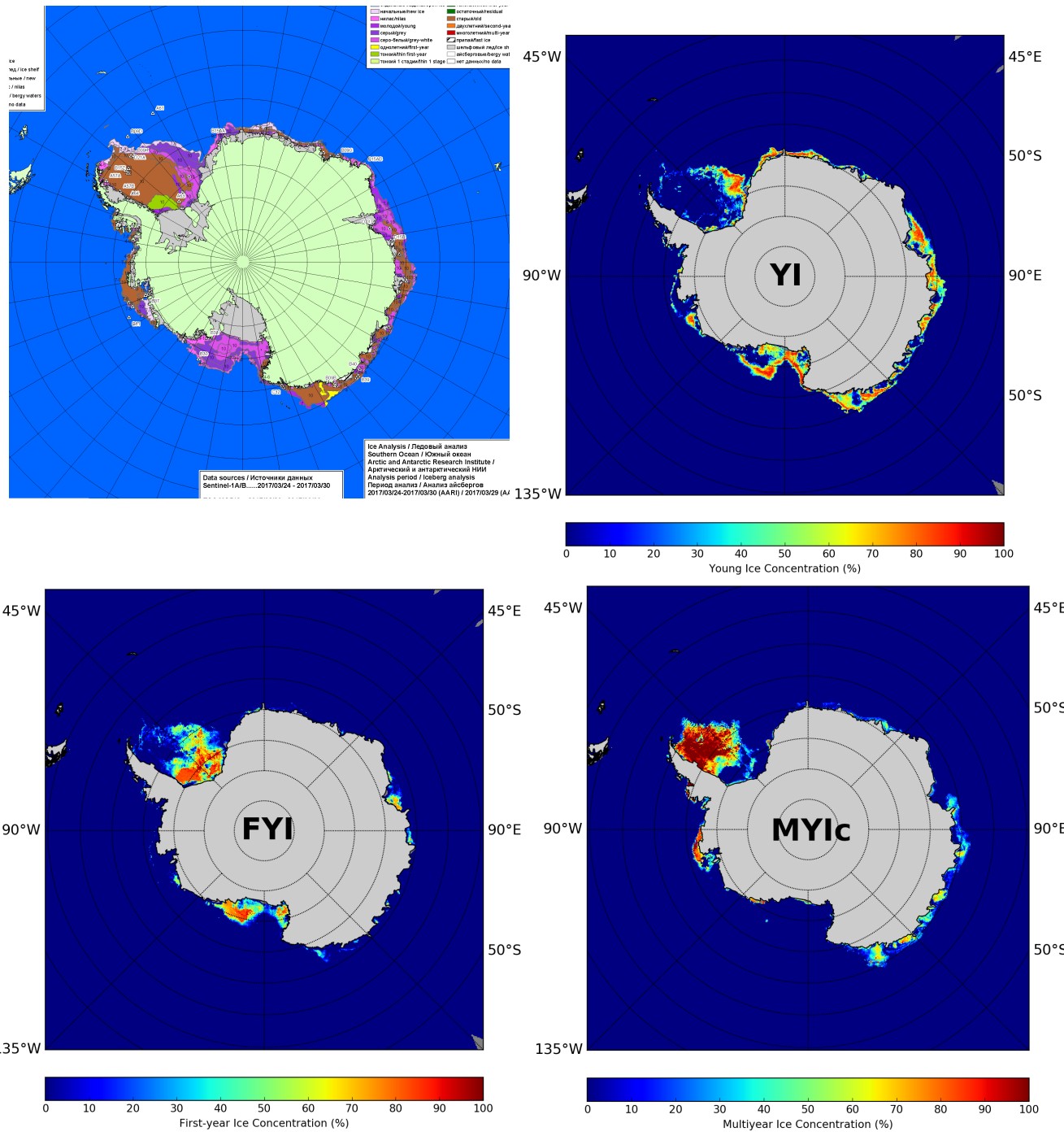

**Figure 4.** Top left: Stage of Development (SoD) chart by AARI, 30 Mar, 2017; top right: YI concentration (ECICE), bottom left, FYI concentration (ECICE); bottom right: corrected MYI concentration (MYIc, ECICE and correction schemes). In the SoD chart, purple shades are YI (see colour legend in the appendix, Table B1), yellow and green shades are FYI, and brown and orange/red shades are MYI.

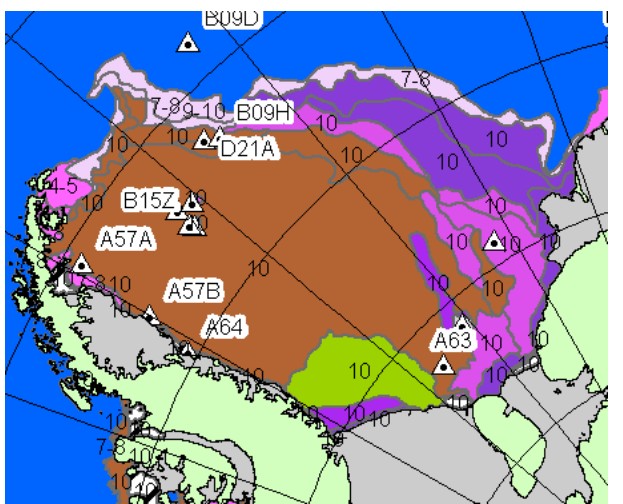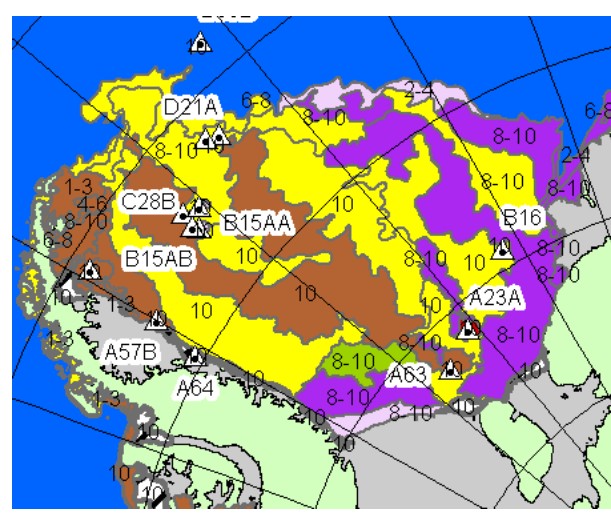

**Figure 5.** Stage of Development (SoD) charts of the Weddell Sea, 30 March, 2017, from AARI (left) and NIC (right). FYI is yellow, MYI ("old ice") brown (see also colour legend in Appendix B,Table B1). The triangle symbols denote large icebergs.

A feature that is very well reproduced by the ice type maps in most years is the drifting of the MYI, adjacent to the Ronne
355 Ice Shelf in the inner Weddell Sea at the beginning of the freezing season, towards the North and North-East in the subsequent
months, shown for 2017 in Figure 6: The columns show charts and maps of the inner Weddell Sea for 16 March, 27 April, 25
May, and 6 July, 2017; namely (top to bottom), SoD, corrected MYI, YI and FYI. The corrected MYI concentration (second
row) has a sharp inner (Southern) boundary that agrees well with the boundary between MYI (brown) and YI (purple) or FYI
(green and yellow) classes in the SoD charts (top row). The MYI and its inner boundary move towards North and North-East
360 from March to July (left to right). Note the similarity to the FYI-MYI boundary in the previous Section 3.1.1 (Figure 3), and the
two strips of MYI separated by FYI in the right part of the rightmost two SoD charts: The MYI strips correspond to similarly
shaped areas with MYI concentration above about 20% to 40% in the MYIc maps (second row). However, the MYI strips of the
SoD charts and the strips in the MYIc maps are usually not fully congruent. This is not surprising as the SoD charts are weekly
charts whereas the retrieved ice type maps are daily averages. The FYI areas in the SoD maps, in turn, correspond rather to FYI
365 concentrations (fourth row) above about 60% to 70%. This hints at the difficulty of comparing a sea ice classification where
each point is assigned exactly one sea ice type class (such as in the SoD charts) with a sea ice type fraction retrieval such as
ECICE where in each grid cell, more than one surface type can coexist with the summation of their fractions (including the
open-water fraction) equal to 100%. The YI and FYI concentration in Figure 6 (third and fourth row) also match reasonably
well with the SoD charts, noting that here as well, areas with YI concentration above about 30% correspond to the class YI
370 in the SoD charts, and only FYI concentrations above about 70% correspond to the FYI classes in the SoD charts. This is
best visible on the first two maps from the left. Another noteworthy detail are the small patches of YI apparently in the lee of
grounded iceberg A23A (third and fourth column). The iceberg itself seems to be marked by almost zero FYI concentration
(see the "holes" in the fourth row, FYI, last two panels).

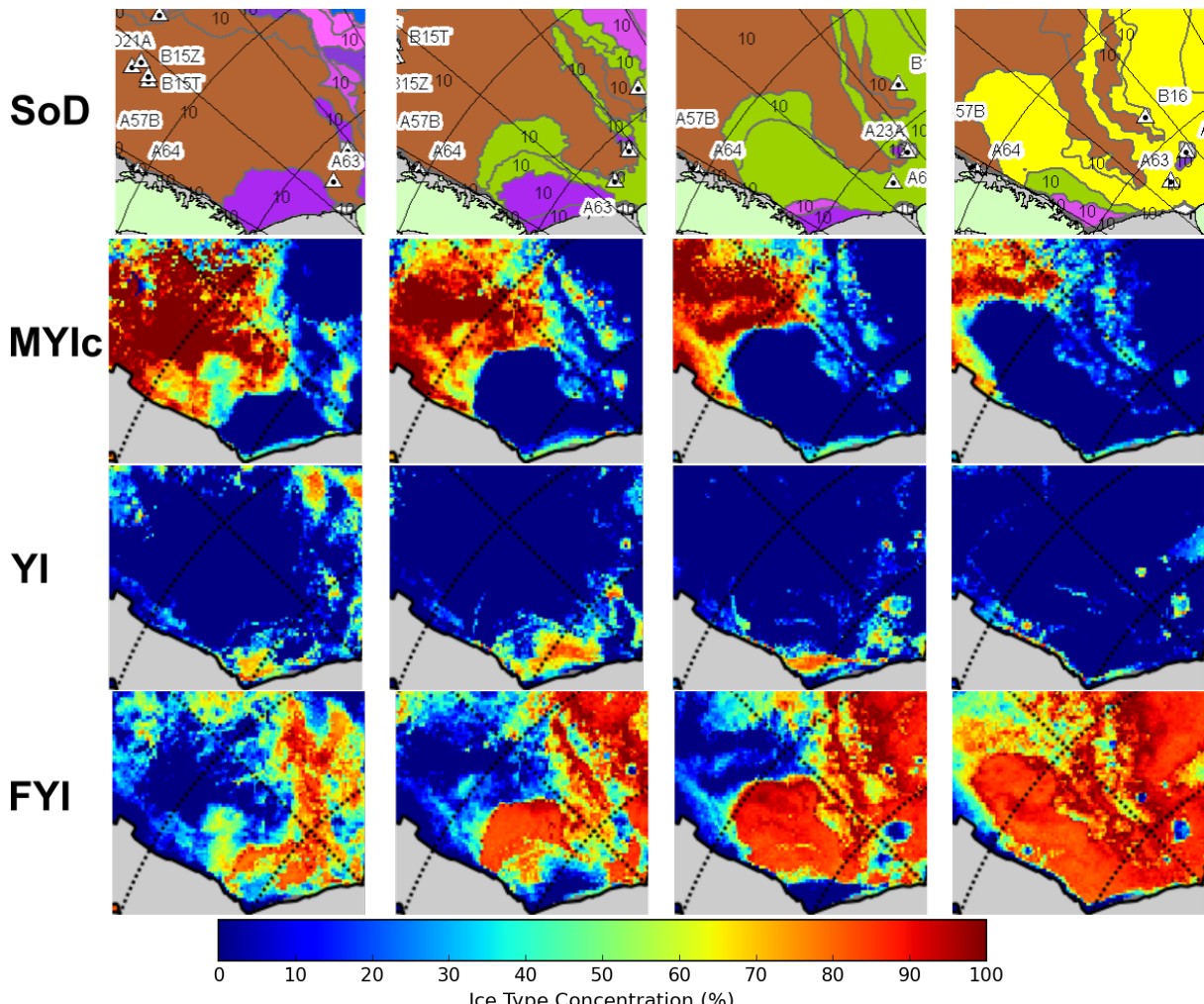

**Figure 6.** Southwestern Weddell Sea, March to July 2017, top to bottom: SoD chart from AARI, corrected MYI (MYIc), YI, FYI concentration; left to right: 16 March, 27 April, 25 May, 6 July, 2017.

We notice that our total ice concentration, which is the sum of YI, FYI and uncorrected MYI concentrations (not shown here), for the cases shown in Figure 6, is about 80–95% in the YI and FYI area, and thus lower than the 10 tenths indicated in the SoD ice charts on the respective day. However, two weeks after the 16 March, 2017 (left row of Figure 6) there are concurrent AARI and NIC SoD charts where the former shows 8-10 tenths while the latter shows 10 tenths, which gives an idea about the uncertainty. So while our total ice concentration might be biased slightly (about 10%) low we still consider the result to be within the expected uncertainty range.

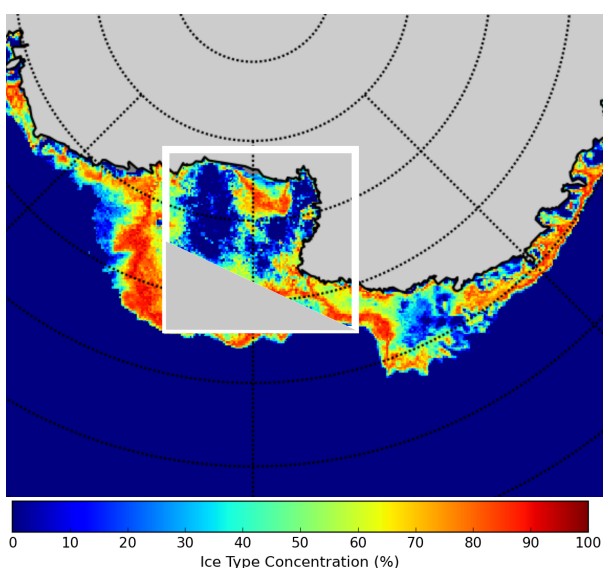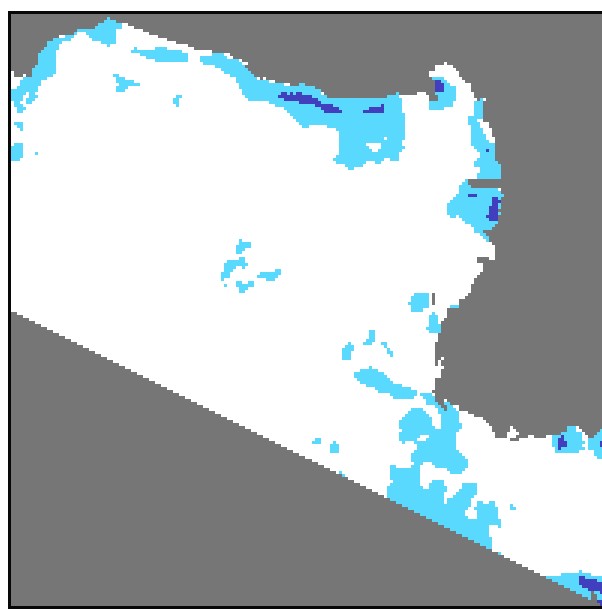

**Figure 7.** Left: map of YI, 3 May, 2017, right: PSSM polynya map in the area indicated by the white rectangle in the YI map (dark blue: open water, light blue: thin ice, white: other ice, grey: masked-out area).

### 3.1.3 Polynya Data Set PSSM

A further comparison was made with the already mentioned polynya data based on microwave brightness temperatures from SSMIS (see Section 2.3), from the year 2017, thus avoiding, of course, the data used for the probability distribution extraction which were from 2018 (Section 2.3). Figures 7 and 8 are examples of an area in the Ross Sea, near the Ross Ice Shelf, on 3 May and 7 September, 2018, respectively: the left panels of both figures show the map of YI concentration from ECICE; the right panels show PSSM maps with the surface type classes open water (dark blue), thin ice (light blue) and other ice (white). Masked out areas (land, non-coastal sea) are shown in grey. The thin ice areas in the polynya data set match the dominantly YI areas from the ECICE retrieval.

Figures 9 and 10 show similar examples from the Weddell Sea, on 2 May, and 19 May, 2017, respectively, demonstrating the variability of coastal polynyas and the young ice associated with them: The thin ice (light blue) in the PSSM maps on the right corresponds roughly to YI concentrations above about 50% (green-yellow-orange) in the retrieved YI maps on the left. On 2 May (Figure 9), there is a large polynya covered with thin ice in the Southwestern Weddell Sea (light blue at the bottom centre of PSSM map on the right) and also coastal polynyas at the Brunt and the Riiser-Larsen Ice Shelves (top right of the PSSM map). Both are matched by high concentrations (orange hues) of YI in the map on the left. On May 19, however (Figure 10), there is much less polynya activity according to the PSSM map, and also no areas of more than about 40% YI in the map on the left, with the exception of some narrow areas on the coast/shelf in the Northeastern part (top right) of the box.

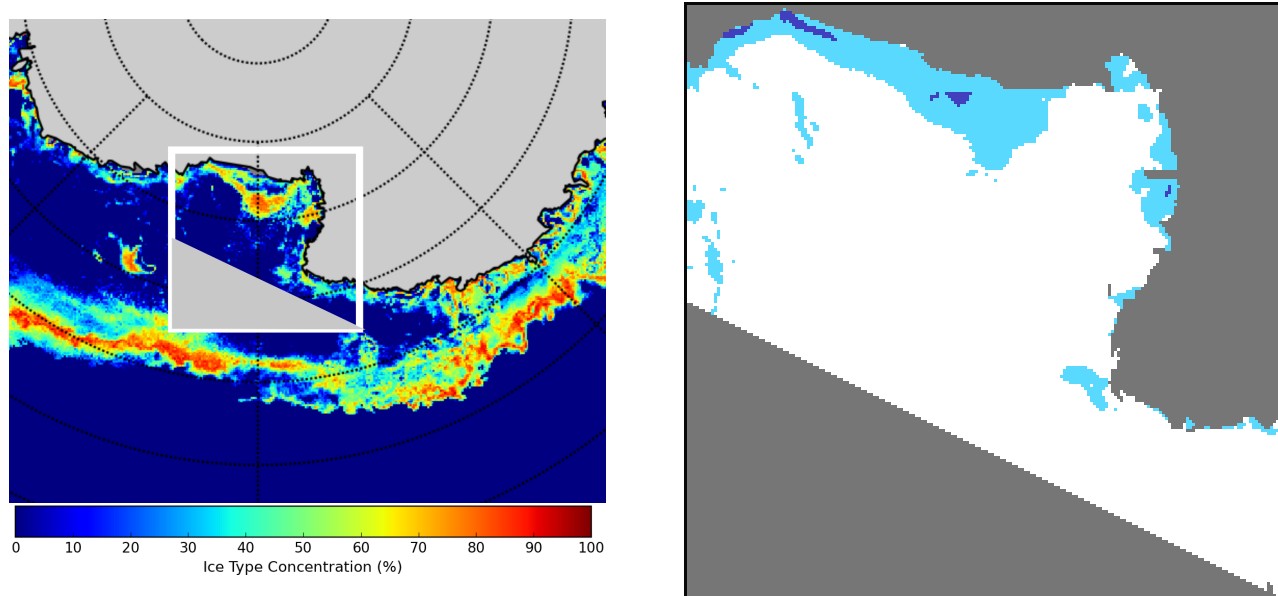

**Figure 8.** Left: map of YI, 7 Sep., 2017, right: PSSM polynya map in the area indicated by the white rectangle in the YI map (dark blue: open water, light blue: thin ice, white: other ice, grey: masked-out area).

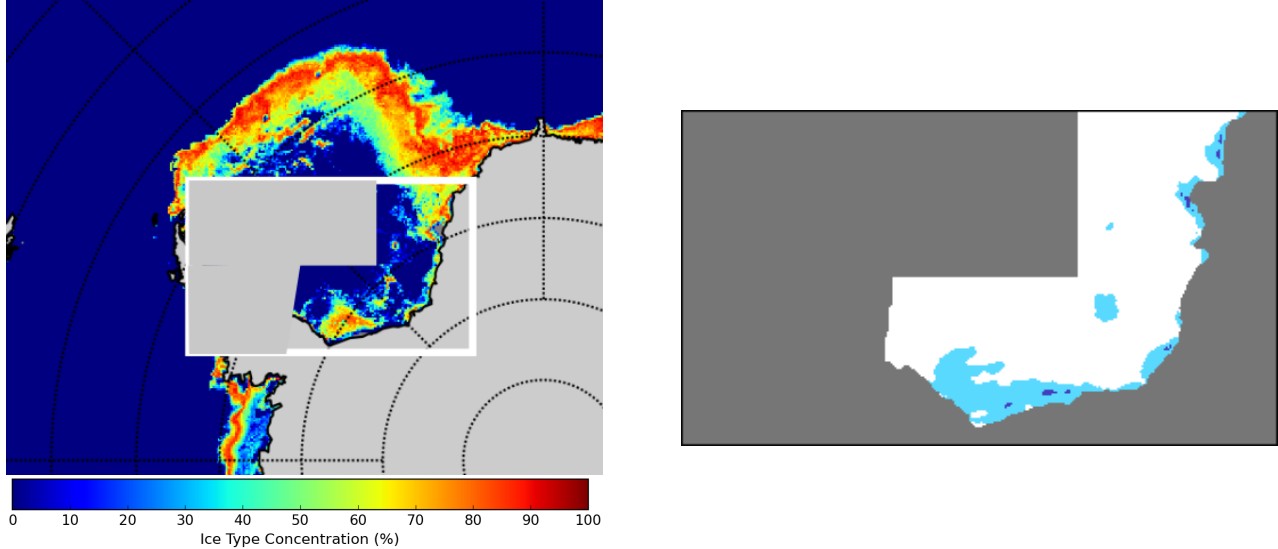

**Figure 9.** Left: map of YI, 2 May, 2017, right: PSSM polynya map in the area indicated by the white rectangle in the YI map (dark blue: open water, light blue: thin ice, white: other ice, grey: masked-out area).

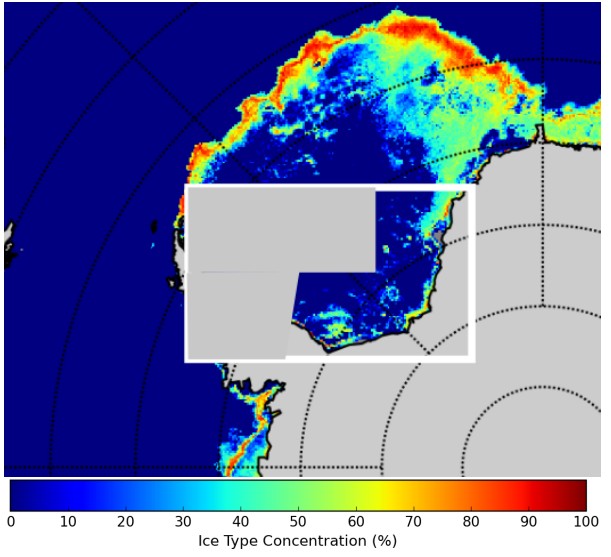 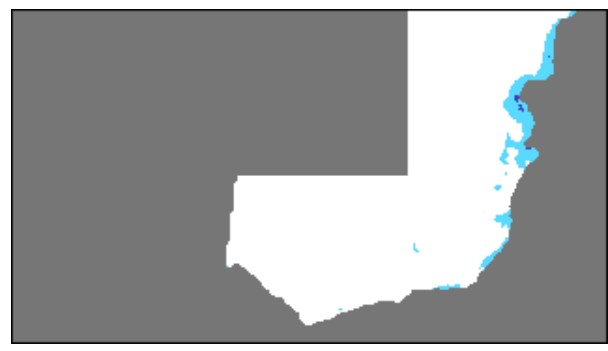

**Figure 10.** Left: map of YI, 19 May, 2017, right: PSSM polynya map in the area indicated by the white rectangle in the YI map (dark blue: open water, light blue: thin ice, white: other ice, grey: masked-out area).

## 3.2 Discussion

As shown in the previous section, the retrieved corrected MYI (MYIc), and also the uncorrected FYI and YI concentrations, compare mostly well with other data. Large icebergs seem to be retrieved as very low FYI, but there is no clear picture, and we expect their radiometric signature to change with the season similar to what happens to sea ice. We have not looked further into that topic as the position and extent of such icebergs is usually well known and monitored, so they can be masked out using ancillary iceberg data (e.g., https://usicecenter.gov/Products/AntarcIcebergs).

The total area of MYI in the entire Antarctic should not increase during one cold season because MYI originates as the remaining ice at the end of the melting season and hence cannot be generated after freeze-up. However, the total MYI area derived from our data shows large fluctuations, and in most years even an increase around July (see also Section 3.3). An example can be seen in Figure 11 which shows the total MYI area for all Antarctic seas for the cold season 2018 (here, a slight increase already starts in May/June). The total MYI area is calculated by multiplying the MYI concentration (0%–100%) of each grid cell with the area of the grid cell (taking into account that in a polar stereographic grid the grid cell area depends on the latitude).

The main reason for the increase in July seems to be a large offshore area of MYI in the outer Ross sea and off Wilkes Land, East of the Ross Sea (roughly between 160°E and 140°W, and between 65° and 70°S) that often seems to grow during the cold season and is not eliminated by the drift correction. Figure 12 shows such an area of MYI that appears in late July, 2018, and disappears in September. While, according to the NIC/AARI ice type (SoD) charts, a considerable area of MYI persists far offshore in the outer Ross Sea in many years (though not in the year 2018 shown here), the retrieved offshore MYI areas

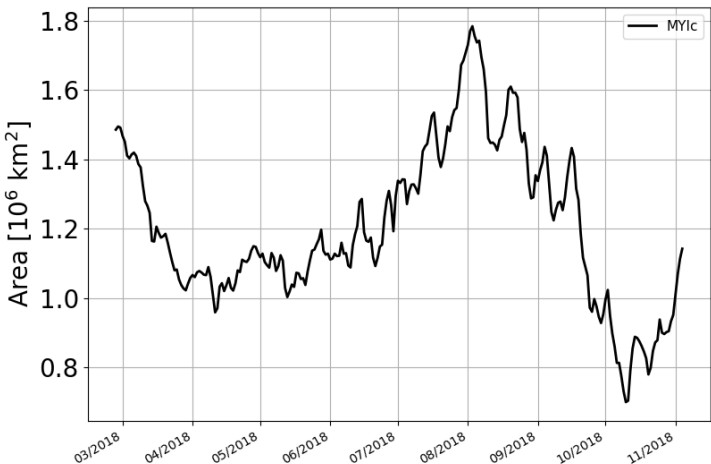

**Figure 11.** Time series of the total area of corrected MYI in the entire Antarctic, March to November 2018.

are larger, in different locations, and sometimes grow quickly within days, which cannot be correct and is not shown on the
SoD charts. As MYI cannot be generated during the freezing season, this is clearly spurious identification of MYI. This area
of spurious MYI is in the marginal ice zone. A similar phenomenon can be observed at around the same time in that region in
all years of the current data record 2013 to 2021 though it is less pronounced in 2013 and 2017. The most likely reason for this
is that in the course of the cold season, the snow layer on FYI in particular in the marginal ice zone changes. Usually, snow
backscatter increases with time and emissivity decreases, making it resemble MYI in that respect (see also  Willmes et al.,
2011; Arndt et al., 2016; Arndt and Haas, 2019). In addition, pancake ice has higher backscatter than FYI and might also be
mistaken for MYI. Elsewhere in the Antarctic seas, such areas of spurious MYI ice can also occur but are generally much
smaller. In principle, spurious MYI showing up during the cold season should be removed by the drift correction. The fact that
this fails can have two possible reasons: (1) problems in the ice drift data used by the drift correction, such as wrong direction
or wrong speed - a single drift vector that is much too large can initiate a growing area of spurious MYI (we also add one pixel
of uncertainty margin to the drift which might also cause the MYI domain to wrongly extend - cf. Section 2.2.2); (2) single
fixed points that permanently, but wrongly show MYI at the beginning of the freezing season: spurious MYI showing up within
one day's drift of such a wrong MYI point will then not be removed. On the next day, spurious MYI next to that point will
not be removed either. In this way, what started as one wrong MYI pixel ("seeding pixel", so to say) can grow into a large
region of wrong MYI within weeks. Such seeding pixels can be caused by "land contamination" of the satellite measurement
in footprints directly at the coast or near small islands that are not properly masked. Since removing coastal MYI pixels and
extending the land mask before applying the correction had no effect on the spurious MYI pixels, the latter reason can be
ruled out. Thus, the reason is most likely inaccuracies in the drift data which accumulate over the season (strange drift vectors
have been observed in particular in the Ross sea near the date line; Ted Maksym, Woods Hole Oceanographic Institution, priv.

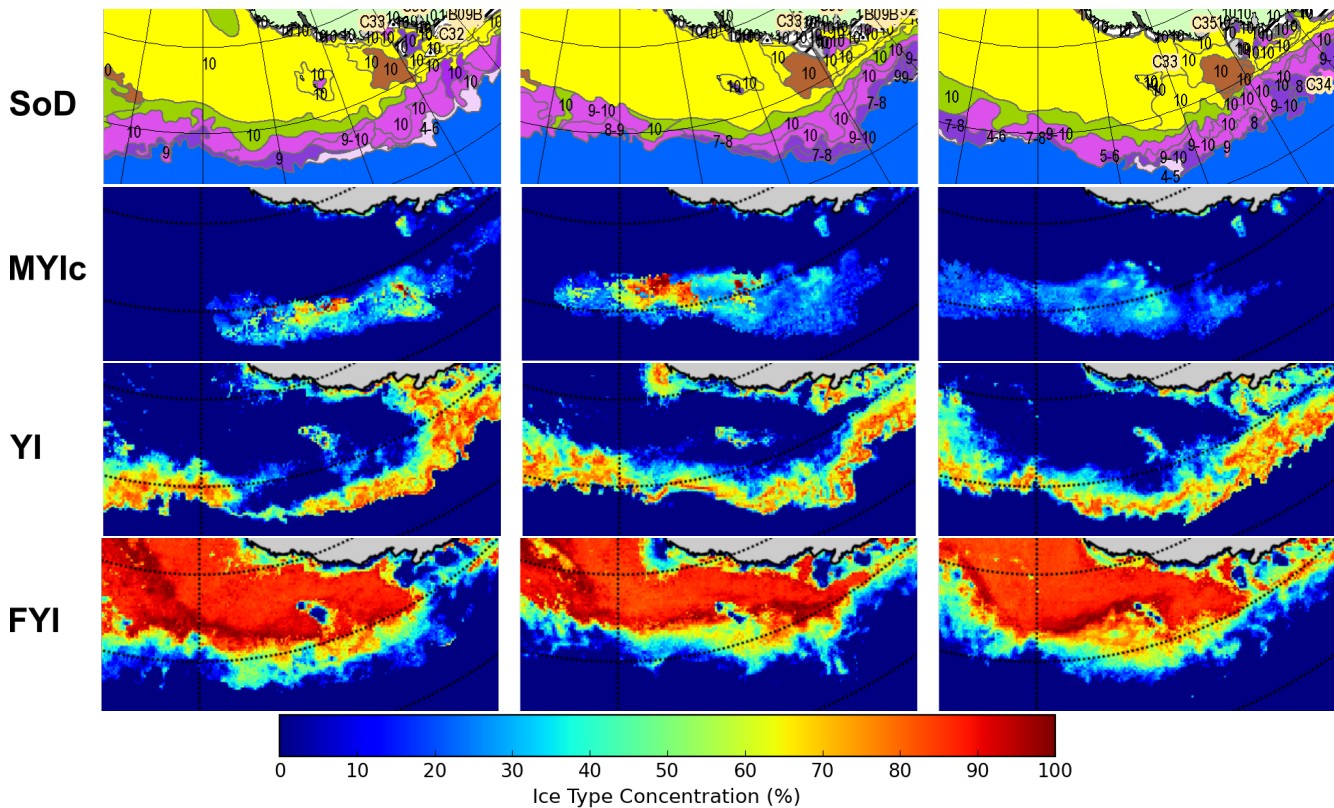

**Figure 12.** Outer Ross Sea, off Wilkes Land, August and September , 2018, top to bottom: SoD chart from AARI, corrected MYI, YI, FYI concentration; left to right: 2 Aug, 30 Aug, 27 Sep, 2018.

comm., 2020). In addition, the possibility cannot be ruled out that there are changes in the Antarctic sea ice that are not captured by the two correction schemes originally developed for the Arctic.

However, before putting effort into finding a flaw in the drift correction that does not properly remove spurious MYI, or devising new correction schemes, it makes more sense to prevent the misclassification that leads to retrieval of spurious MYI by ECICE in the first place. To do this, the distributions of MYI, FYI, and YI in the different channels should be investigated, specifically in the outer Ross Sea and off Wilkes Land. The fact that YI in the marginal ice zone often is pancake ice (see above) has not sufficiently been taken into account by our focusing just on YI in polynyas. If these parameter distributions for the ice types in the Ross Sea differ significantly from those currently used (see Section 2.3), they should be adapted. Looking at the lists of sampling areas and times for the distribution (Table A2 in Appendix A), we note that the MYI samples are all from the beginning of the freezing season. The reason was of course that the sample areas were found by visual analysis of the evolution of daily sea ice maps, starting at the beginning of the freezing season when all remaining ice is MYI (cf. Section 2.3). Since the MYI will drift and possibly suffer break-up from divergence with new ice forming in the gaps, we cannot only derive the needed areas of pure (100%) multiyear ice with this approach in the first weeks after freeze-up. The consequence, however,

is that the MYI samples are probably not representative for the MYI later in the season. Therefore, finding MYI samples later in the season, using external data, might improve the retrieval.

Towards the end of the freezing season, in September and October, the retrieved, corrected MYI concentration declines strongly in most years (see Figure 11), which is not seen in the weekly SoD charts. However, SoD charts seem to become unstable in the sense that the ice type changes between FYI (WMO type 2.5 "first-year ice" (JCOMM Expert Team on Sea Ice, 2015)) and MYI (WMO type 2.6 "old ice") from one week to the next, i.e., between AARI chart and NIC chart as they produce the weekly SoD charts in turns. Occasionally, there are almost simultaneous SoD charts by AARI and NIC, and they also disagree in the FYI-MYI discrimination (which might suggest that FYI and MYI at that time look almost similar even to ice analysts). Note that the FYI samples (Table A2 in Appendix A) are from April, June and August, so they might not contain the conditions in September and October.

The most likely reason for the too strong decline of MYI in our retrieval scheme are temperatures rising to near melting conditions which causes MYI to be misclassified as FYI as described above in Section 2.2.1 about the temperature correction. Where these near-melting or melting conditions are not episodic any more, they cannot be corrected by the temperature correction.

Note that a first analysis of the new OSI-SAF ice type classification data (Aaboe et al., 2021a) likewise does not show the expected decrease of Antarctic MYI area in the course of the season, but instead a steady slow increase followed by a rapid decline toward the end of the freezing season in September/October (Aaboe et al., 2021b, Fig. 14), which is similar to the time series derived from our data (Figure 11).

## 3.3 Time series 2013-2021

Having processed data from 2013 to 2021, we can now look at the longer time series. The problems with wrong MYI in the outer Ross Sea mainly start in the middle of the cold season, so, e.g., the MYI data in the beginning of the cold season should be more robust. Having nine consecutive seasons of ice type data, i.e., 2013–2021, we can have a look at how they evolve, in particular in view of the strong changes of the overall Antarctic sea ice extent since 2014: there was a record large sea ice extent in September 2014, then a record decline resulting in a record low in February 2017, and another record low in February 2022. A first check is to add up the concentration of the three ice types, which results in total sea ice, and compare the time series of its extent with the "standard" time series of sea ice extent based on data that do not distinguish between sea ice types, like the times series based on the ASI algorithm. Here (just for one plot) we use the common definition of extent as the total area of all grid cells with an ice concentration above 15%. Our time series shows the same ranking of the yearly maxima of Antarctic sea ice extent as the "standard" ones (here: ASI), see Figure 13. As the sea ice type retrieval does not work well during melting season, the months November to February are masked out, so the yearly minima are not captured.

Now we can have a look at the total area of the three ice types – here we sum up the areas of all grid cells multiplied with their concentration (i.e., surface fraction, 0% to 100%) of the respective ice type. The result is shown in Figure 14. We see the steep rise of YI (black curve) and FYI (blue) at the beginning of each freezing season, the levelling off of the YI while the FYI continues to grow most of the season. The MYI area (green curve) in each season should not grow, which is not the case

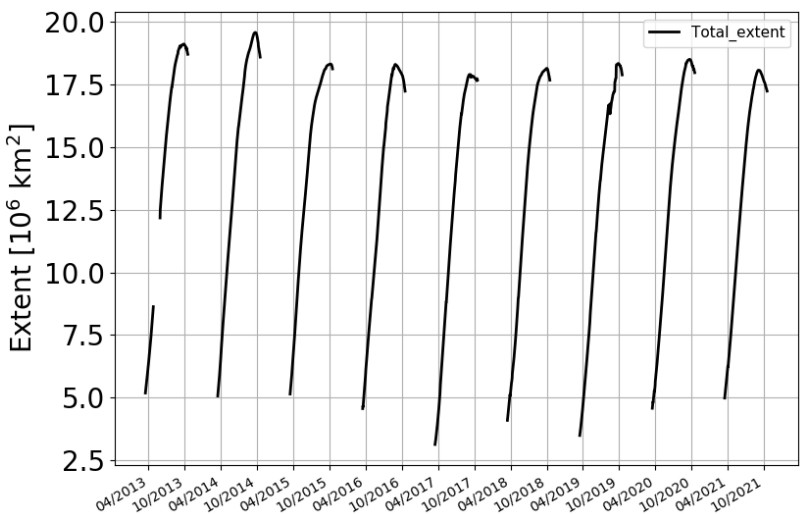

**Figure 13.** Total sea ice extent of the Antarctic sea ice, years 2013–2021, calculated from the sum of YI, FYI and MYI concentrations, smoothed with a 31-day moving average.

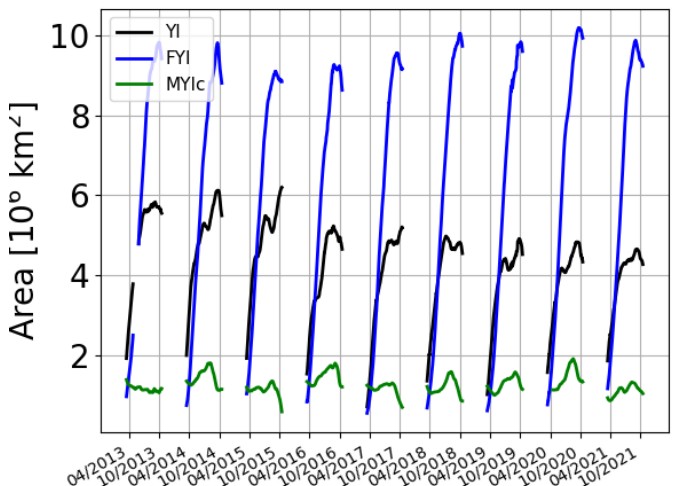

**Figure 14.** Area of the three sea ice types YI (black), FYI (blue) and corrected MYI (green), years 2013–2021, March to October, smoothed with a 31-day moving average.

in most years: except for 2013 and 2017, there is a peak in the middle of the freezing season, which is, as mentioned in the previous section, caused by (1) erroneous MYI retrieval due to insufficient sampling of the parameter distributions for MYI, and then (2) failure of the correction scheme to adequately correct this, most notably in the outer Ross Sea.

Figure 15 zooms in into just the corrected MYI area in 2017 and 2018 shows the extent of this erroneous rise that starts in
July 2018 and amounts to about 50% of the area (from 1.2 million $km^2$ to 1.8 million $km^2$).

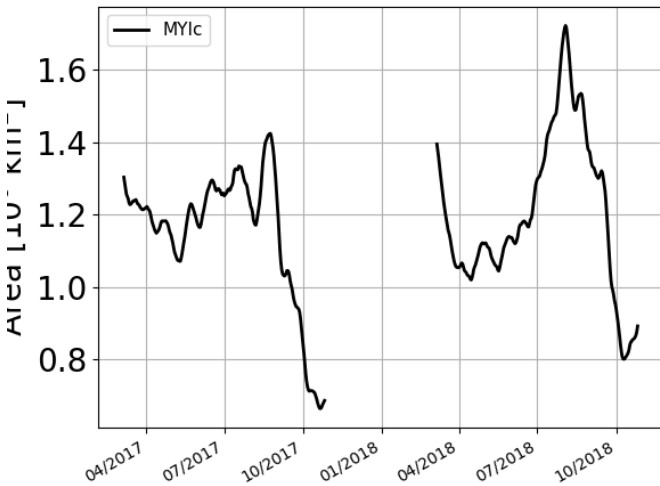

**Figure 15.** Area of corrected MYI, years 2017 and 2018, March to October, smoothed with a 11-day moving average.

The main question, however, is the interannual variability or change. In Figure 14, we see (1) a decline of the YI area maximum, (2) fluctuations of the FYI maximum that almost follow the maxima of the total ice extent (Figure 13) – the extreme years are not entirely the same; but (3) not much interannual change of the MYI area so far. Moreover, the MYI area seems hardly influenced by the strong fluctuations of total ice cover since 2014.

## 4   Summary and Conclusions

The sea ice type retrieval method ECICE (Shokr et al., 2008; Shokr and Agnew, 2013) and the subsequent correction schemes for MYI (Ye et al., 2016a, b) developed for the Arctic can be adapted for the Antarctic, given samples of Antarctic ice types are available. Input satellite data are microwave radiances at 19 and 37 GHz as well as scatterometer backscattering measurements. Daily maps of uncorrected YI, FYI and MYI, and of MYI corrected for effects of melt-refreeze and snow metamorphosis, can be retrieved, during the freezing season, at grid resolution of 12.5 km. The results look reasonable in the sense that they show agreement with SAR images, with remote-sensing-based polynya data, and with weekly charted sea ice stage of development (so far the only source of detailed ice type information in the Antarctic, apart from ship-based observations of the ice conditions). In particular, the general distribution of the Antarctic MYI at the beginning of the freezing season is well captured by our corrected ECICE results. The subsequent time evolution of the MYI concentration in the Weddell Sea, as far as the AARI/NIC stage of the development charts can tell, is captured as well in the months after freeze-up, showing the effects of advection and melt. The retrieved distributions of YI and FYI concentration are reasonable as well. However, comparing our ice type concentrations, i.e., area fractions of ice types, with weekly charts that assign only one ice type to each location is inherently problematic. A more detailed validation study that also includes in-situ observations from research cruises is planned. The most problematic area is the outer Ross Sea and the sea off Wilkes Land, where large and growing areas of spurious MYI are

retrieved in the marginal ice zone in late winter in most years. The data become unstable toward the end of the freezing season in September/October, with MYI probably being underestimated. The new time series, spanning the years 2013 to 2021, is the first comprehensive time series to give insight into the distribution and evolution of Antarctic sea ice types. A first look at it shows that the MYI area in the Antarctic has apparently not followed the strong fluctuations of the Antarctic total ice extent observed since 2014. The current time series of uncorrected YI, FYI, MYI and MYIc can in principle be extended backwards

to 2002, using AMSR-E radiometer and QuikSCAT scatterometer data.

The next steps to establish a consolidated long-term time series of Antarctic MYI are, in order of importance: (1) derive new distributions for MYI, FYI and YI, possibly including pancake ice in FYI or YI; (2) evaluate the ECICE ice type results before the correction schemes to make sure that all ice types are correctly identified under typical freezing conditions; (3) improvement of correction schemes, i.e., evaluate additional sea ice drift data from NSIDC (version 4) in addition to OSI-SAF,

and use ERA-5 reanalysis data instead of ERA-Interim (4) retrieve Antarctic MYI concentration for the years 2002–2011 (the "era" of AMSR-E); (5) comparison with in-situ and ship data where available.

Finally, the time series can be continued for the years to come, as the successor instrument of the radiometer AMSR2, namely, AMSR3, is scheduled for launch in 2023 or 2024 (see, e.g., Maeda et al., 2022), and scatterometers similar to ASCAT will be on the European MetOp Second Generation satellites B1, B2 and B3, from about 2025 . In addition, there is the new

Chinese-French Oceanography Satellite (CFO-Sat) carrying a $K_u$ band scatterometer (Hauser et al., 2016), and the upcoming mission CIMR (Copernicus Imaging Microwave Radiometer, see Kilic et al., 2018) with radiometer channels at an improved spatial resolution from 1.4 to 36.5 GHz, that might even enable sea ice type retrieval without additional scatterometer data: At 1.4 GHz, the radiation emitted by sea ice comes from a much thicker layer than emission at 19 GHz and higher. As MYI is less saline, has more air inclusions, thicker snow, and on average is thicker than other ice types, MYI can be identified by a

combination of the $1.4$ to $36.5$ GHz channels (Scarlat et al., 2020).

*Code availability.*

*Data availability.* The uncorrected sea ice type data of the Antarctic, years 2013 to present, are available at the web site http://seaice. uni-bremen.de; the corrected MYI data of the Antarctic, years 2013-2021, are available on the same web site, and, in addition, as a data set in the PANGAEA archive, https://doi.org/10.1594/PANGAEA.909054

**Appendix A: Details on sample regions**

ECICE needs the distribution of the input parameters, $T_B(37V)$, $T_B(37H)$, $GR(37V, 19V)$, $\sigma°$ for the four different surface types, YI, MYI, FYI, open water. The Tables A1 and A2 list the geographical coordinates (latitude and longitude of lower left and upper right corner of rectangles in the NSIDC Antarctic maps used throughout in this paper), the dates on which the input

**Table A1.** Sample areas for YI in the Western Ross Sea. Note: lat. = latitude, lon. = longitude, no, = number

| no. | lower left lat. | lower left lon. | upper right lat. | upper right lon. | date | no. of samples |
|---|---|---|---|---|---|---|
| 1 | 71.91°S | 179.09°W | 73.56°S | 175.6°E | 28 Mar 2018 | 195 |
| 2 | 76.22°S | 179.05°E | 77.14°S | 171.0°E | 28 Mar 2018 | 160 |
| 3 | 70.59°S | 176.62°E | 72.6°S | 172.06°E | 03 Apr 2018 | 220 |
| 4 | 75.07°S | 178.02°E | 76.01°S | 173.17°E | 03 Apr 2018 | 90 |
| 5 | 76.73°S | 179.26°E | 77.43°S | 172.65°E | 12 May 2018 | 96 |
| 6 | 76.1°S | 178.34°E | 77.38°S | 172.94°E | 25 May 2018 | 120 |
| 7 | 76.67°S | 179.51°W | 77.39°S | 171.61°E | 03 Jun 2018 | 136 |
| 8 | 76.03°S | 177.41°E | 77.19°S | 172.52°E | 19 Jun 2018 | 96 |
| 9 | 74.87°S | 175.87°E | 77.24°S | 170.41°E | 22 Jun 2018 | 176 |
| 10 | 76.26°S | 177.13°E | 77.14°S | 171.0°E | 27 Jul 2018 | 120 |
| 11 | 76.07°S | 176.22°E | 77.38°S | 171.1°E | 28 Jul 2018 | 117 |
| 12 | 76.54°S | 177.06°W | 77.51°S | 173.66°E | 08 Sep 2018 | 162 |
| 13 | 76.24°S | 175.93°W | 77.62°S | 176.54°E | 11 Sep 2018 | 168 |
| 14 | 76.21°S | 177.85°E | 77.07°S | 170.54°E | 19 Sep 2018 | 126 |
| 15 | 76.56°S | 179.02°W | 77.06°S | 171.83°E | 29 Oct 2018 | 90 |
| 16 | 75.92°S | 177.9°E | 76.77°S | 172.01°E | 07 Nov 2018 | 96 |

**Table A2.** Sample areas for MYI, FYI and open water (OW). Note: lat. = latitude, lon. = longitude, no, = number

| surface type | lower left lat. | lower left lon. | upper right lat. | upper right lon. | region | date range | no. of samples |
|---|---|---|---|---|---|---|---|
| MYI | 70.66°S | 99.18°W | 71.66°S | 97.17°W | Bellingshausen Sea | 3 Mar – 4 Mar 2018 | 112 |
| MYI | 70.66°S | 99.18°W | 71.66°S | 97.17°W | Bellingshausen Sea | 8 Mar – 9 Mar 2018 | 112 |
| MYI | 66.14°S | 59.15°W | 66.5°S | 54.46°W | Weddell Sea | 28 Feb – 4-Mar 2018 | 740 |
| FYI | 73.94°S | 43.84°W | 73.14°S | 31.1°W | Weddell Sea | 27 Apr – 30 Apr 2018 | 2100 |
| FYI | 69.31°S | 23.24°W | 67.47°S | 14.95°W | Weddell Sea | 11 Jun – 13 Jun 2018 | 1512 |
| FYI | 71.73°S | 152.79°W | 75.46°S | 158.7°W | Ross Sea | 21-Aug – 25-Aug 2018 | 3240 |
| OW | 51.15°S | 130.48°W | 58.56°S | 129.33°W | Souther Ocean | 21 Aug – 25 Aug 2018 | 12500 |
| OW | 74.28°S | 172.87°W | 76.43°S | 177.09°E | Outer Ross Sea | 1 Mar – 3 Mar, 2018 | 1242 |

parameters were sampled, and the number of samples (one sample is one grid point on one day). Table A1 lists the sampling
areas for YI (from the polynyas in the Western Ross Sea), and Table A2 lists the sampling areas for MYI, FYI and open water
(OW).

| Colour | | RGB colour model | Stage of development (SoD) | Number from WMO Sea Ice Nomenclature |
|---|---|---|---|---|
| alternative | prime | | | |
| | | 000-100-255 | Ice free | 4.2.8 |
| | | 150-200-255 | <1/10 ice of unspecified SoD (open water) | 4.2.6 |
| | | 240-210-250 | New ice | 2.1 |
| | | 255-175-255 | Dark nilas | 2.2.1 |
| | | 255-100-255 | Light nilas | 2.2.2 |
| | | 170-040-240 | Young ice | 2.4 |
| | | 135-060-215 | Grey ice | 2.4.1 |
| | | 220-080-235 | Grey-white ice | 2.4.2 |
| | | 255-255-000 | First-year ice (FY) | 2.5 |
| | | 155-210-000 | FY thin ice (white ice) | 2.5.1 |
| | | 215-250-130 | FY thin ice (white ice) first stage | 2.5.1.1 |
| | | 175-250-000 | FY thin ice (white ice) second stage | 2.5.1.2 |
| | | 000-200-020 | FY medium ice | 2.5.2 |
| | | 000-120-000 | FY thick ice | 2.5.3 |
| | | 180-100-050 | Old ice | 2.6 |
| | | 000-120-000 | Residual ice | 2.6.1 |
| | | 255-120-010 | Second-year ice | 2.6.2 |
| | | 200-000-000 | Multi-year ice | 2.6.3 |

**Table B1.** Colour coding of the sea ice types in the ice charts shown above

## Appendix B: WMO colour coding of sea ice types

The AARI/NIC ice charts showing the stage of development in Figures 4 to 12 use the colour code specified by WMO (2014). For easier reference, the colour codes of the ice types relevant here are shown in Table B1.

## Appendix C: Incidence angle conversion of scatterometer data

The conversion of the ASCAT NRCS to a standard incidence angle is based on the observation that the dependence of the NRCS (in dB) of sea ice on the incidence angle $\theta$ for ASCAT can be well fitted by a straight line (also observed for ERS scatterometer data by Gohin and Cavanié, 1994). The data were taken over regions with near 100% ice cover in January 2016 in the Arctic, and a linear fit with respect to the incidence angle $\theta$ yielded the slope $p = -0.192$ dB/°. Then the NRCS $\sigma°(\theta)$ is converted to the NRCS at $\theta = 40°$ as

$$\sigma°(40°) = \sigma°(\theta) + (40° - \theta)p \tag{C1}$$

While this is a rather simple method, it works well: daily composites of ASCAT NRCS data thus converted have not shown any visible swaths.

*Author contributions.* CM has adapted the algorithm and correction schemes for the Antarctic, has written the initial manuscript and revised

it; YY has developed the original correction schemes for the Arctic and has given feedback and advice for the work; MS has contributed the original ECICE retrieval which is the basis of the whole retrieval, has given important feedback and advice and contributed substantially to the revision; GS has contributed the comparison with SAR data, and has given critical scientific advice in all stages of the work; all co-authors have reviewed the manuscript.

*Competing interests.* The authors declare that they have no competing interests.

*Acknowledgements.* This work was supported by the Deutsche Forschungsgemeinschaft (DFG), project SITAnt (grant 365778379), in the framework of the Antarctic priority programme SPP 1158 "Antarctic Research with comparative investigations in Arctic ice areas", and the European Union's Horizon 2020 research and innovation programme via project CRiceS (Climate Relevant interactions and feedbacks: the key role of sea ice and Snow in the polar and global climate system; grant 101003826). The authors acknowledge the International Space Science Institute (ISSI) in Bern for support and for discussions in the ISSI team "Satellite-Derived Estimates of Antarctic Snow and Ice

Thickness" led by P. Heil, University of Tasmania. The authors are very grateful to S. Kern, University of Hamburg, for producing and providing the polynya data set, and to S. Arndt and C. Haas, Alfred Wegener Institute, Bremerhaven, for important discussions. JAXA, EUMETSAT, and ESA are acknowledged for providing the AMSR2, ASCAT, and Sentinel-1 satellite data, respectively, and AARI and NIC are acknowledged for the stage of development ice charts in the framework of the "AARI-NIC-NMI pilot project on integrated sea ice analysis for Antarctic waters". The authors also want to thank the anonymous reviewers for their constructive criticism, additional hints and

pieces of information which have greatly improved this work.

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
