# Peer review of "First results of Antarctic sea ice type retrieval from active and passive microwave remote sensing data"

_The Cryosphere, 2021_

## Referee Comment (RC2)

**tc-2021-381**

Review of tc-2021-381: Antarctic sea ice types from active and passive microwave remote sensing, by C. Melsheimer and co-authors

The authors present in this manuscript a method based on a combination of scatterometer and passive microwave radiometry data to map the distribution of the Antarctic sea ice types. With a product archive covering years 2013-2019, this is the first time a multiyear record of Antarctic ice type is presented and made publicly available. The Arctic sea ice type (or age) has been monitored for a long time and has shown a clear decreasing trend of the older ice. Similar monitoring of the different ice types for the Antarctic ice has been lacking and therefore the presented work is filling an important gap and the topic is very relevant for TC.

Below I present some of my major comments which should be considered before publishing. This is followed by other comments to consider as well.

**Major comments:**

**Writing style**

The manuscript would benefit from (and strongly needs) a thorough tightening up of the writing style of the whole manuscript, including but not limited to:
- Use a clear structure with separated sections for the data and the methodology. At present, the input data and input products are presented "here and there" within the method section.
- Avoid repetitions. Several repetitions occur and sometimes a simple re-ordering of the sentences would make the reading flow better.
- Make sure that information given across the manuscript is in synergy with itself. For instance, it is very unclear whether the new product covers 2013-2019, 2013-2020, or 2013-present.
- Make use of general spell checking.

The presentation of this work should be clearly separated from possible improvements or potential future works. Future works or possible upgrades should rather be listed and discussed in a discussion section or the Conclusion.

My general comments below is not a full overview of typos, grammatical errors, or repetitions, etc that I found.

**Algorithm presentation vs presentation of long-term time series**

The manuscript seems to have two main goals. On the one hand, it presents a methodology for mapping the Antarctic sea ice type from remote sensing data. And on the other hand, it presents, for the first time, a longer time series of Antarctic sea ice types. Both these topics

are very relevant, however, at the present stage, the manuscript covers these in an inadequate manner.

In order to be a manuscript presenting a new method, the methodology is only superficially described. Below are some specific points to be considered:

- I suggest naming the full algorithm which is implemented at the University of Bremen. When reading the manuscript, it is unclear if "the new method" is ECICE, modified ECICE, or ECICE + post-processing. Examples of this:
  - L64: "Recently, a method has been developed …". Which method is this? Add a reference.
  - L66-69: "The method is based on ECICE … and a later modification …" Unclear if the ECICE has been modified, or if post-processing includes modifications of the outcome…
  - L72-74: "In this study, we have adapted this method to the Antarctic conditions …". Again unclear what "this method" is.
  - L85-86: "Our estimation of MYI concentration actually is a two-step procedure that first uses ECICE and then applies two correction schemes …" Assuming that "our estimation" is coming from "the new method", here for the first time it seems clear that the method is in fact the ECICE retrieval pluss some post-processing (correction schemes).
  - L193: "The final result of the two-step retrival scheme …".
  - etc.
- L114-116: it is mentioned that "the median is used as a measure of confidence of the result for each surface type". However, this confidence field is never presented and it is not clear if this information is provided together with the ice type product.
- Section 2.1 is describing the core/backbone of the classification algorithm. I would have liked to see a few equations or illustrations (e.g. flow diagram) of the ECICE methodology. Especially, the paragraph relevant for the Antarctic adaption (L105-117) is hard to read as it is and would benefit from supplementing equations/illustrations.
- L166: "For all input parameters, we use daily gridded data". How did you arrive at these gridded data? Especially, for scatteromter data, a sentence on how the angle-dependent swath data are gridded would be relevant.
- L175: "Later in the season, sea ice that has formed … away from MYI is FYI". How do you account for the changing position of MYI during the season and thereby collect only FYI data?
- L196-197: "… retrieved the ice type concentrations for the months of Feb to Nov …". I did not find any comment on why summer months are omitted, or why exactly this period has been chosen for the Antarctic product.
- The final output is "corrected MYI" (L193), however the "uncorrected FYI, MYI, and YI" are also provided.
  - It is not clear if the uncorrected fields are "pure ECICE" outcome?
  - uncorrected MYI (or Ex-MYI) is never presented, and maybe they should be shown?
  - Could you include a comment whether MYIcorrected+FYI+YI or MYI+FYI+YI add opp to 100%? And if not, please comment on this as well.
- It is not clear if a threshold is used for the ice edge? Or are all surfaces with >0% ice concentration classified?

- Several places it is announced that this computation can be extended to present time. What will the latency be for such a retrieval?

In order to be a manuscript presentating a longer time series, it is surprising to see that there is a complete lack of presenting or showing any long-term (seasonal, interannual, regional) behaviour or variability. Only a few hand-picked days are shown and the year 2018 MYI total area time series. Since this is the first time a longer time-series of ice type is presented for the Antarctic, then it would be appropriate to show a full record plot and potentially discuss any trends and variabilities, (or missing trends).

Several places it is mentioned the possibility of a record covering the period 2002 to present. However, the present record covers the period 2013-2019. Can you make this more clear - what defines the period you present and why is this period selected? And hereafter (best fitted in a discussion or conclusion section) mention possibilities for extending the time series and what this would require and what is the timeline for implementing this.

**Other comments:**
General comment for the figures: Could you consider to label the sub-figures and thereby avoid " top-right", etc

Figure 1: Why not simply add the sensor type in the title of each subfigure instead of the "Dist. set: Aq2" which is a title that does not give any sense.

Figure 1: Some of the shown density distributions clearly shows a double-peak. Can you comment on this and could this in fact indicate that more types are represented by the distribution?

Figure 2: I find it difficult to locate this region. Either you could add an overview map or with words explain better where this is. E.g. " the inner part of ..."

Figure 3: the sub-titles should be upgraded, preferably with one sub-title for each subplot. Also the sub-title of the lower-right should be updated/corrected.

Figure 6-9: Here is used four figures for illustrating Young Ice. Potentially, these could be merge into fewer figures. The PSSM maps - the gray color should be defined in the caption. Even better, if the coast/land could be added for better orientation of the sub section.

Figure 10/L286: What is shown in this figure? Is it the extent of all ice pixels that contain some concentration of MYI? Or is the MYI concentration or the full ice concentration taken into account?

General comment for the tables: The layout of the table caption differs between the tables - some times it appears above and sometimes below.

General comments:
L25: you could add a reference to the YI definition (e.g. WMO Sea-Ice Nomenclature 2017)

L33: explain or add a reference to why all MYI is SYI.

L38-39: "Antarctic sea ice has strong region-dependent …" This sentence stands a bit alone. Could you say a few more words on this?

L43: Move "in the Antarctic" to the beginning: "Sea ice cover in the Antarctic…"

L51: repeats that MYI typically is in Weddell Sea (L31)

L62: "Total and partial sea ice concentration …" when first time reading this, it was not clear that "partial sea ice concentration" was referring to the ice types. I suggest re-wording this whole paragraph.

L72: "Ye et al., 2019" is not accepted for publication. Can you find another reference?

L74: replace "regularly" with "operationally"

L77: "brief account of ECICE and its adaption to Antarctic …" Would it be more correct with "brief account of ECICE, implemented correction schemes, and the adaption to Antarctic …"

L78: remove "first"?

L78-79: since the entire record is never presented, I would suggest deleting the period. Simply "In sec 3, the outcome of the Antarctic sea ice type concentration mapping is compared with …"

L91: Any reason why SSM/I can be included but not SSMIS (and other passive microwave radiometers)? Why mention SSM/I if it is not included in the present method/production?

L100: "Most methods" please include references.

L132: "If MYI .. drops at any location during a warm spell …". Are there not any restrictions on this drop to occur in the vicinity of where the warm anomaly appears?

L140: "After that, MYI can only drift …" What exactly do you mean by this? If you mean that no more MYI will be created (per theoretical definition) after this point, then say this more clearly. In the same sentence is used the word "melting" which is not a part of "drifting"... Please re-phrase this sentence.

L144: "... boundary of MYI cover …" Please define the boundary of MYI cover. Is this where MYI conc = 0%?

L153: "... sudden reductions …", sudden reductions in time, I assume?

L156: "The values of the parameters …". Please indicate from where these values are taken, e.g. include reference or discussion on how they have been chosen.

L160: Why is AMSR-E presented here when the ice type record covers 2013-2019?

L167-168: Is the Melsheimer reference the right reference to add just after NSIDC?

L168-169: Could you please elaborate a bit more on this, e.g. by simply presenting the approximate spatial resolution of the used input data?

L174: As "ASI" is used only once, I suggest to just fully write the full name here, for easier readabilty.

L174-175: It is a bit unclear from what seasons the training data is collected. Is MYI data collected from only beginning of the freezing period. Please give a bit more details.

L188: Are there any reasons for using ERA Interim instead of the newer ERA5?

L190: When mentioning the potential NSIDC ice drift data, please include a comment on why OSI SAF ice drift data are chosen to be used, and whether NSIDC data have been tested out in the ice type retrieval. Also, please note that an ice drift climate data record from OSI SAF is in the pipeline for this spring 2022 (regarding L202-203).

L191-192: this is a repetion

L218: Are these threshold procentages randomely chosen, or can you comment on why 50% is used for YI and 70% for the others.

L225: To my knowledge, NIS do not produce Stage of Development maps. Plese check this out.

L225: To my knowledge, no SoD maps are available on the webpage in 2014. Please chech this out.

L231-233: Is this information relevant (and is it true in practice?)?

L235-236: when you write "an overall correspondence" - is this referring to results shown here, or did you check all available charts against the product. Please give more information on what is in overall correspondence and how this has been concluded.

L279-280: The contribution from Ex-MYI is not shown or taken into account here. Please include this contribution to the discussion of FYI and YI.

L310: Please add affiliation or title for Ted.

L312-314: This paragraph is unclear. Please re-phrase.

L315-316: I suggest that you include some mapping examples from September/October to better visualize for the reader why you see an increase in the MYI area.

L332: "spatial resolution" or grid resolution?

L334: "(so far the only source of ice type information in the Antarctic)". This should be reworded. I assume that in situ data exists to some extent. Also, OSISAF ice type product exists and is processed on an operational basis.

L341-342: "The data become unstable toward the end of the freezing season in September/October, with MYI being underestimated" I think this has not been shown clearly in the result section. And how do you conclude that MYI is being underestimated?

L344: I would remove the sentence "outweighs the shortcomings". This only put your product in a bad light I would say.

L346-350: Give better and more correct references to upcoming satellites/sensors. And please add a comment on why 1.4 - 36.5 GHz is assumed to make scatterometer less important.

---

## Author Comment (AC2)

**Response to RC2:**

Note that we quote the reviewer's comments and suggestions in red.

**Major comments:**

**Writing style**

The manuscript would benefit from (and strongly needs) a thorough tightening up of the writing style of the whole manuscript, including but not limited to:

● Use a clear structure with separated sections for the data and the methodology. At present, the input data and input products are presented "here and there" within the method section.

We will try to make a clearer separation between data and methodology. Having to separate sections "Data" and "Methods", however, does not seem very useful to us, as mentioning all data *before* explaining the method is much information that cannot be used yet, and mentioning the data *after* the describing the method would be too late. We rather suggest a table listing the input data and auxiliary data at the end of section 2

● Avoid repetitions. Several repetitions occur and sometimes a simple re-ordering of the sentences would make the reading flow better

● Make sure that information given across the manuscript is in synergy with itself. For instance, it is very unclear whether the new product covers 2013-2019, 2013-2020, or 2013-present.

● Make use of general spell checking.

We will try to straighten the text, eliminate inconsistencies like the one mentioned, and, of course, apply a spell checker again.

The presentation of this work should be clearly separated from possible improvements or potential future works. Future works or possible upgrades should rather be listed and discussed in a discussion section or the Conclusion.

We think that mentioning some possible improvements/potential future works in the course of the paper is unavoidable: For example. when discussing the wrong multiyear ice in the Ross Sea, we of course mention possible ways to mitigate that (L311 ff.). However, we agree that it makes sense to list all possible improvements and future work in one place towards the end. We will either do that at the end of the discussion or in the final summary/outlook chapter.

**Algorithm presentation vs presentation of long-term time series**

The manuscript seems to have two main goals. On the one hand, it presents a methodology for mapping the Antarctic sea ice type from remote sensing data. And on the other hand, it presents, for the first time, a longer time series of Antarctic sea ice types. Both these topics are very relevant, however, at the present stage, the manuscript covers these in an inadequate manner.
In order to be a manuscript presenting a new method, the methodology is only superficially described. Below are some specific points to be considered:

We did not want to repeat the description of the ECICE algorithm and the two correction schemes here, as they are described in the respective publications, but we will try to follow the suggestions below.

● I suggest naming the full algorithm which is implemented at the University of Bremen. When reading the manuscript, it is unclear if "the new method" is ECICE, modified ECICE, or ECICE + post-processing. Examples of this:

○ L64: "Recently, a method has been developed ...". Which method is this? Add a reference.

(The reference is already at the end of that sentence.)

○ L66-69: "The method is based on ECICE ... and a later modification ..." Unclear if the ECICE has been modified, or if post-processing includes modifications of the outcome...

○ L72-74: "In this study, we have adapted this method to the Antarctic conditions ...". Again unclear what "this method" is.

○ L85-86: "Our estimation of MYI concentration actually is a two-step procedure that first uses ECICE and then applies two correction schemes ..." Assuming that "our estimation" is coming from "the new method", here for the first time it seems clear that the method is in fact the ECICE retrieval pluss some post-processing (correction schemes).

○ L193: "The final result of the two-step retrival scheme ...".

○ etc.

Thank you for pointing this out. We agree that we must make the naming more consistent throughout the manuscript and will do that.

● L114-116: it is mentioned that "the median is used as a measure of confidence of the result for each surface type". However, this confidence field is never presented and it is not clear if this information is provided together with the ice type product.

This information is saved along with the resultss of ECICE. We will insert a brief discussion of that.

● Section 2.1 is describing the core/backbone of the classification algorithm. I would have liked to see a few equations or illustrations (e.g. flow diagram) of the ECICE methodology. Especially, the paragraph relevant for the Antarctic adaption (L105-117) is hard to read as it is and would benefit from supplementing equations/illustrations.

We consider including a flow diagram of the methodology but are unsure about the degree of detail needed here. The ECICE algorithm is described in detail in the cited publications (Shokr et al., 2008 ; Shokr and Agnew, 2013) and we do not want to repeat too much of that here.

● L166: "For all input parameters, we use daily gridded data". How did you arrive at these gridded data? Especially, for scatteromter data, a sentence on how the angle-dependent swath data are gridded would be relevant.

We combine all AMSR swaths of one day and then interpolate to the grid using a distance-weighted near-neighbor approach (the one from Generic Mapping Tools with four sectors). The ASCAT swath data are converted to common incidence angle of 40° and then interpolated to the grid. We will either add these details in this section or put this into a small section in the appendix.

● L175: "Later in the season, sea ice that has formed ... away from MYI is FYI". How do you account for the changing position of MYI during the season and thereby collect only FYI data?

We made sure the used FYI areas are so far away from the start-of-the-season MYI that the latter cannot have drifted there. If we assume 15 km of maximum daily drift (rough estimate from 1 year of OSISAF drift data), this would mean, e.g. 1500 km in June. We will add such an estimate to the description.

● L196-197: "... retrieved the ice type concentrations for the months of Feb to Nov ...". I did not find any comment on why summer months are omitted, or why exactly this period has been chosen for the Antarctic product.

In general, under permanent melting conditions, the radiometric/backscattering properties of sea ice change considerably and differences between the ice type diminish or even vanish. Therefore using

ECICE in summer does not yield reasonable ice types. We apologise for not having stated this important fact and will add a similar explanation to section 2.1 on ECICE.

● The final output is "corrected MYI" (L193), however the "uncorrected FYI, MYI, and YI" are also provided.
○ It is not clear if the uncorrected fields are "pure ECICE" outcome?

Yes, the uncorrected fields are the "pure ECICE" outcome. We will add this were we mention the preliminary ice types (L194, "without applying and correction")

○ uncorrected MYI (or Ex-MYI) is never presented, and maybe they should be shown?

We have actually already considered that and decided against it as ExMYI should first be investigated in more detail. Please note that other ice concentration algorithms that produce MYI do not apply corrections to account for anomalies in the locations of the MYI. The correction scheme used in ECICE can be used in any algorithm.

○ Could you include a comment whether MYIcorrected+FYI+YI or MYI+FYI+YI add opp to 100%? And if not, please comment on this as well.

The uncorrected ice type concentrations add up to the total ice concentration (which can be between 0% and 100%, of course), When correcting MYI, the amount added or subtracted is not subtracted or added from FYI or YI as we cannot say to which of the latter two it "belongs". Hence, MYIcorrected+FYI+YI cannot add up to the total ice concentration. We will add this explanation to the text.

● It is not clear if a threshold is used for the ice edge? Or are all surfaces with >0% ice concentration classified?

We do not use any threshold, but directly retrieve the concentration of the three ice types everywhere. In areas of open water (100%), all ice types have 0% of course. We consider mentioning this in an appropriate place.

● Several places it is announced that this computation can be extended to present time. What will the latency be for such a retrieval?

We have since implemented daily retrieval. The ECICE output is in near real time, within 1 day of receiving AMSR2 and ASCAT data. The latency of the corrected MYI is 16 days. We will mention this important information at an appropriate place.

In order to be a manuscript presentating a longer time series, it is surprising to see that there is a complete lack of presenting or showing any long-term (seasonal, interannual, regional) behaviour or variability. Only a few hand-picked days are shown and the year 2018 MYI total area time series. Since this is the first time a longer time-series of ice type is presented for the Antarctic, then it would be appropriate to show a full record plot and potentially discuss any trends and variabilities, (or missing trends).

We will add a section showing complete (2013-2021) time series and discuss them.

Several places it is mentioned the possibility of a record covering the period 2002 to present. However, the present record covers the period 2013-2019. Can you make this more clear - what defines the period you present and why is this period selected? And hereafter (best fitted in a discussion or conclusion section) mention possibilities for extending the time series and what this would require and what is the timeline for implementing this.

We will state this more clearly: The period starts with the availability of AMSR2 data, in principle in July 2012. As the drift correction scheme needs to starts at the beginning of the cold season, we start to retrieve in 2013. The end of the period is now 2021 – we will change the text accordingly.

**Other comments**

General comment for the figures: Could you consider to label the sub-figures and thereby avoid " top-right", etc

The use of "top right" etc. seems common practice in many publications, but we will consider the sub-figure approach.

Figure 1: Why not simply add the sensor type in the title of each subfigure instead of the "Dist. set: Aq2" which is a title that does not give any sense.

We will modify the figure along the suggestions given.

Figure 1: Some of the shown density distributions clearly shows a double-peak. Can you comment on this and could this in fact indicate that more types are represented by the distribution?

Yes, e.g. the distribution of MYI in GR17,37V, the red curve in the lower left plot, shows to almost distinct peaks. This might point at to subtypes. However, in order to retrieve one more ice type (by splitting "MYI" into two subtypes), ECICE would also need one more input channels as the number of input channels must be equal or larger than the number of retrieved surface types. We are not sure if is makes sense to add this discussion into the manuscript, though.

Figure 2: I find it difficult to locate this region. Either you could add an overview map or with words explain better where this is. E.g. " the inner part of ..."

We will do that (overview map or text).

Figure 3: the sub-titles should be upgraded, preferably with one sub-title for each subplot. Also the sub-title of the lower-right should be updated/corrected.

Usually there is just one caption under a figure. We will consider readability and try to improve on that.

Figure 6-9: Here is used four figures for illustrating Young Ice. Potentially, these could be merge into fewer figures. The PSSM maps - the gray color should be defined in the caption. Even better, if the coast/land could be added for better orientation of the sub section.

We consider combining the figures, but are not sure if this makes them better to read. The meaning of the grey areas will be defined in the caption, of course.

Figure 10/L286: What is shown in this figure? Is it the extent of all ice pixels that contain some concentration of MYI? Or is the MYI concentration or the full ice concentration taken into account?

As mentioned in the caption, this is the full MYI area, taking the MYI ice concentration of each pixel (0% to 100%) into account. So this is *not* the extent, which is commonly defined as the sum of the full area of all pixels with an ice concentration above 15%.

General comment for the tables: The layout of the table caption differs between the tables - some times it appears above and sometimes below.

Will be fixed.

General comments:
L25: you could add a reference to the YI definition (e.g. WMO Sea-Ice Nomenclature 2017)

Yes, we add reference to the WMO Sea Ice Nomenclature when we first introduce the ice types.

L33: explain or add a reference to why all MYI is SYI.

We will add the remark that almost all Antarctic sea ice will drift out into lower latitudes and meld within two years.

L38-39: "Antarctic sea ice has strong region-dependent ..." This sentence stands a bit alone. Could you say a few more words on this?

As this statement is not really relevant here, we rather delete it.

L43: Move "in the Antarctic" to the beginning: "Sea ice cover in the Antarctic..."

Yes.

L51: repeats that MYI typically is in Weddell Sea (L31)

Yes, but we see no problem in repeating this statement here as we here discuss the importance of the Weddell Sea in particular.

L62: "Total and partial sea ice concentration ..." when first time reading this, it was not clear that "partial sea ice concentration" was referring to the ice types. I suggest re-wording this whole paragraph.

We well reword this: "The concentration of total sea ice and of the sea ice types"

L72: "Ye et al., 2019" is not accepted for publication. Can you find another reference?

A revision of the cited study is in progress, for the time being, there is only the discussion paper.

L74: replace "regularly" with "operationally"

Yes

L77: "brief account of ECICE and its adaption to Antarctic ..." Would it be more correct with "brief account of ECICE, implemented correction schemes, and the adaption to Antarctic ..."

Yes, of course – we will correct this.

L78: remove "first"?
L78-79: since the entire record is never presented, I would suggest deleting the period.
Simply "In sec 3, the outcome of the Antarctic sea ice type concentration mapping is compared with ..."

We rewrite the sentence as: "In Section 3, results of the Antarctic sea ice type concentration mapping are compared with results from..."

L91: Any reason why SSM/I can be included but not SSMIS (and other passive microwave radiometers)? Why mention SSM/I if it is not included in the present method/production?

After 2002, the preferred satellite instruments to be used are AMSR-E and AMSR2 as they have higher resolution than SSM/I and SSMIS. SSM/I was mentioned as it can extend the record backwards before the AMSR-E era. SSMIS can, of course also be used, and actually it can close the gap between AMSR-E (until Oct 2011) and AMSR2 (from July 2012). We will mention this here.

L100: "Most methods" please include references.

We will include reference to the ice concentration algorithm comparison paper by Ivanova et al., (2014, doi:10.5194/tc-9-1797-2015), this saves specifying 10 extra references.

L132: "If MYI .. drops at any location during a warm spell ...". Are there not any restrictions on this drop to occur in the vicinity of where the warm anomaly appears?

We actually meant "at any location affected by the warm spell" and will correct the text accordingly.

L140: "After that, MYI can only drift ..." What exactly do you mean by this? If you mean that no more MYI will be created (per theoretical definition) after this point, then say this more clearly. In the same sentence is used the word "melting" which is not a part of "drifting"... Please re-phrase this sentence.

We will slightly reword the sentence: "After that, during the cold season, no new MYI can be generated. MYI can then only drift, and its concentration can only be changed by divergence, convergence, and melting."

L144: "... boundary of MYI cover ..." Please define the boundary of MYI cover. Is this where
MYI conc = 0%?
We use a threshold of 20% MYI concentration to define the MYI boundary. We will insert this information in the manuscript

L153: "... sudden reductions ...", sudden reductions in time, I assume?
Yes, we use "sudden" in the standard, temporal, meaning, as the specification "(within one day)" suggests.

L156: "The values of the parameters ...". Please indicate from where these values are taken,
e.g. include reference or discussion on how they have been chosen.
The values were empirically determined for the correction schemes in the Arctic and have been kept here. We will insert this information.

L160: Why is AMSR-E presented here when the ice type record covers 2013-2019?
AMSR-E data will be used in the next step. However, using them has already been implemented.

L167-168: Is the Melsheimer reference the right reference to add just after NSIDC?
Yes, it is, as the NSIDC grid in the context of sea ice type retrieval is described in more detail there.

L168-169: Could you please elaborate a bit more on this, e.g. by simply presenting the
approximate spatial resolution of the used input data?
Yes, the actual numbers will be inserted.

L174: As "ASI" is used only once, I suggest to just fully write the full name here, for easier
readabilty.
This is why we have put the full name and the link and reference into a footnote.

L174-175: It is a bit unclear from what seasons the training data is collected. Is MYI data
collected from only beginning of the freezing period. Please give a bit more details.
The MYI data are collected from the first months of the freezing season. We will add more details on that to the text.

L188: Are there any reasons for using ERA Interim instead of the newer ERA5?
When most of the presented work was done, ERA-5 was not available yet.

L190: When mentioning the potential NSIDC ice drift data, please include a comment on why
OSI SAF ice drift data are chosen to be used, and whether NSIDC data have been tested
out in the ice type retrieval. Also, please note that an ice drift climate data record from OSI
SAF is in the pipeline for this spring 2022 (regarding L202-203).
(NSIDC drift data have been used already for retrieval in the Arctic (see Ye at al., 2016b). At the time this study was started, the used OSISAF data seemed the best choice, but we are actiually considering switching).

L191-192: this is a repetion
Well, we once more reference Tables 2 and 3 (after referencing them in sections 2.2.1 and 2.2.2) and see no harm in that.

L218: Are these threshold procentages randomely chosen, or can you comment on why 50%
is used for YI and 70% for the others.
They were empirically chosen.

L225: To my knowledge, NIS do not produce Stage of Development maps. Plese check this
out.
NIS of the Norwegian Met. Institute is at least one of the three partners of that project. As far as we know, NIS has contributed some regional ice charts (but not SoD?). -- We will leave out NMI/NIS

dL225: To my knowledge, no SoD maps are available on the webpage in 2014. Please chech this out.

Thank you for pointing this out, concentration maps start in December 2014, but SoD maps only in May 2015. We will correct that.

L231-233: Is this information relevant (and is it true in practice?)?

We considered this information necessary because it makes some MYI (in the first half of the season, until end of June) appear in similar colours as FYI. We will check, however, this is is really needed here.

L235-236: when you write "an overall correspondence" - is this referring to results shown here, or did you check all available charts against the product. Please give more information on what is in overall correspondence and how this has been concluded.

We have compared at least one SoD chart per month (there is one per week) for two entire seasons.We consider publishing these comparisons as supplementary data. We will elaborate on that.

L279-280: The contribution from Ex-MYI is not shown or taken into account here. Please include this contribution to the discussion of FYI and YI.

As already mentioned above, we had decided against showing Ex-MYI as we have not investigated it thoroughly. We can only say that it is actually FYI or YI (but cannot tell which).We consider adding this information to the text.

L310: Please add affiliation or title for Ted.

Yes.

L312-314: This paragraph is unclear. Please re-phrase.

Yes, we will rephrase it to make it clearer.

L315-316: I suggest that you include some mapping examples from September/October to better visualize for the reader why you see an increase in the MYI area.

In L315-316, we speak of a decline of MYI concentration, not an increase. We will consider adding a new figure here.

L332: "spatial resolution" or grid resolution?

Yes. it should be "grid resolution".

L334: "(so far the only source of ice type information in the Antarctic)". This should be reworded. I assume that in situ data exists to some extent. Also, OSISAF ice type product exists and is processed on an operational basis.

We meant detailed and comprehensive ice type information (this OSISAF ice product has only FYI and MYI) that does not only rely on automatic satellite data processing. We will modify the text accordingly.

L341-342: "The data become unstable toward the end of the freezing season in September/October, with MYI being underestimated" I think this has not been shown clearly in the result section. And how do you conclude that MYI is being underestimated?

See L315-322 in the Discussion section: We retrieve much less MYI than the SoD charts which, however, seem a bit "unstable" or inconsistent. In the discussion, we give a possible physical explanation for an underestimation (L319-322). Here, in the summary, we will add a "probably" before "underestimated".

L344: I would remove the sentence "outweighs the shortcomings". This only put your product in a bad light I would say.

Thank you for this encouragement! We will remove these words, but add a "preliminary" earlier in this sentence.

L346-350: Give better and more correct references to upcoming satellites/sensors. And please add a comment on why 1.4 - 36.5 GHz is assumed to make scatterometer less important

We will revise these lines and add references, and mention a study that showed possible retrieval of ice types without scatterometer data.

---

## Author Response (AR1)

**Response to RC1:**

We thank the reviewer for the critical eye and for the very detailed and constructive criticism.

**General Comments**

Here is our response to the four general comments (GC1 to GC4)

**GC1: Physics**

It is true that sea ice physics is different between the two polar regions. We agree with the reviewer on the 3 listed aspects, namely the MYI age and formation, the dynamic environment of the sea ice and the differences in the snow depth and composition. There are more aspects manifested in ice motion patterns, ice kinematics (particularly lead formation) and dynamics of marginal ice zones. However, characterization of those differences is beyond the subject of this study. Differences in ice physics between the 2 regions warrants a separate study, perhaps in the form of a review paper or a book chapter. In this regard, we would like to assert that little is known about sea ice physics in the Antarctic compared to the Arctic. Just a quick look at the literature is enough to prove this point. Most books on Arctic sea ice have an opening chapter on ice physics. This is not the case with the only book on Antarctic sea ice (published in 1998, edited by M Jeffries). Nevertheless, a summary of the major differences between gross properties of sea ice and snow in both polar regimes can be compiled and included in the Introduction section though this would increase the length of the manuscript, which we are hesitant to offer. Instead we have slightly extended the section dealing with the differences between Arctic and Antarctic sea ice (L43-50 of original manuscript) and added references (see item (2.) and (3.) below.

The objective of the study is stated as to utilize the ECICE algorithm to quantify the concentration of each ice type (YI, FYI and MYI) with a focus on MYI in the Antarctic region (similar to what has been done for the Arctic ice). It is true that the physics of ice and snow impacts the observed signal (from passive and active microwave sensors in this case) but we do not have to dig in to address the questions of how and why. All what we need to do is to provide the algorithm with a realistic probability distribution of each observation used in the processing for each ice type (without having to explain the physics behind them). This is already covered in Section 2.3. This is no different than the description of other commonly-used sea ice concentration algorithms that use tie points to represent the used radiometric parameter for a given ice type (e.g., NASA Team, Bootstrap, ASI, etc.). In the original papers of those algorithms there is no description of the physics that engender the tie points. Only sampling from homogeneous area of each ice type was needed. This is what we have done in this study (and in the application of ECICE in the Arctic), namely relying on samples from representative area of the given ice types. ECICE has the advantage of using the probability of occurrence of all possible radiometric values from a given ice type, not just a single tie point.

Finally, yes … the snow cover affects the observed signal in ways that we do not fully understand. That is why using the probability of occurrence of all possible values from a certain ice type when covered by snow under different conditions becomes necessary (the advantage of ECICE as mentioned above).

**GC2: Previous work**

The reviewer has raised a few points under this title. We provide a few more statements in the Introduction to point out the benefits of ice type mapping to the modelling community. However, we see no need to discuss the sea ice type mapping in the Arctic (we already provide reference to the ECICE application to the Arctic ice). Less attention has been paid to the Antarctic sea ice simply because there are no economic or geopolitical benefits attached to it. A case in point is the yet unclear

impact of global warming on the Antarctic sea ice, when so much information (observation and modelling) is readily available about the impacts on the Arctic ice.

As for the current knowledge about Antarctic sea ice type distribution, we now provide more information in the Introduction but not about the typical emissivity, brightness temperature and radar backscattering as suggested because we see this as distraction from the objective of the manuscript. One final point in response to the comments under the GC2 section: ECICE was not modified to apply it to the Antarctic ice. Only the suitable input data had to be used. This is now stated explicitly at the beginnig of section 2.3.

**GC3: Description of the methodology**

We have completed the description. There is no difference in the application of the method between Arctic and Antarctic. The difference is in the input data (see above). We have slightly extended the explanations about the sampling and have also added an appendix with details about the sampling areas.

**GC4: Description and interpretation of the results:**

The reviewer raised the point of background information about the difference in physical properties between Arctic and Antarctic sea ice. We have addressed this point in GC1. The reviewer also suggested that it was mandatory to include an expert on Antarctic ice and snow to appropriately interpret the results.

We would like to offer the following arguments. The algorithm is about identifying and quantifying the 3 ice types in each resolution cell of the data based on the input probability density function of the used radiometric values for each ice type. If the input is wrong the output is wrong. Therefore, what is needed from an expert is help to identify authentic samples of a given ice type to construct the distributions. We will check this point, though not raised by the reviewer.

The actual suggestion of the reviewer is to use the knowledge of the expert to support the conclusions from the results. The reviewer presents 3 good themes as examples: the spurious MYI in the Weddell Sea and Ross Sea, the likelihood of leaking YI signature into FYI, and the handling of the iceberg signature, which confuses the identification of MYI. We have thought about including an expert coauthor to help addressing these issues, could not really think of one, besides the effort it would mean for someone to join in the middle of the work.

As for the situations where the sum of the 3 ice types do not sum up to 100%: Note that the (uncorrected) concentrations of the three ice types *plus* the open water fraction add up to 100% (which is guaranteed by the equality constraint in ECICE (see L111-113 of original manuscript) whereas the sum of the three ice type concentration adds up to the total ice concentration which can be 0% to 100%. See also the specific comment on that, item (11.), below.

Finally, yes, in this study we aimed at qualitative evaluation of the results because we thought that this would be appropriate for a first study to apply a new technique to the Antarctic sea ice. The purpose is to "prove the concept" rather than provide a comprehensive data set on ice type distribution for use in models. We nnow state this explicitly at the end of the introduction. We have worked to complete the description of the data used for the evaluation. and try to incorporate some quantitative evaluation in certain areas where more information is readily available in the literature, e.g., MYI in the Weddell and Ross Seas.

**Specific comments**

Note that we quote the reviewer's specific comments and suggestions in red and have numbered them.

**(1.)** Line 33-34 "... but it is ... satellite data" is perhaps an a bit too general statement which i) could be specified better by telling the approaches used of doing so (aka: using instantaneous microwave observations) [note: using multi-annual time series of satellite data would work as well], perhaps by including the work of Comiso et al. (2011?) who figured out the differences in the signature of Arctic SYI vs. MYI, and which ii) could be amended by the fact that sea-ice age data retrieved for the Arctic (but not the Antarctic) are based on ice motion data which are in fact derived using satellite data. Hence it IS possible but nobody looked into it yet.

We now say "...using satellite data directly (i.e., not using multi-annual satellite observation or drift data based on multi-temporal satellite imagery)"

**(2.)** L43-50: This paragraph is meant to provide the fundament for why ECICE needs some form of adaptation when applied to Antarctic sea ice. In that respect and given that this paper is the first attempting to derive partial ice-type concentrations for Antarctic sea ice, it would make a lot of sense to provide an adequate review of the difference in the sea ice AND snow properties year-round between the Antarctic and the Arctic that is back-up very well by a convincing set of references. This paragraph does not fulfil that role and should be re-written. --> GC1 / GC2

See our answers to GC1 and GC2

**(3.)** L47-49: "The ice cover ... The turbulent ..." --> I encourage you to provide 1-2 references each that underline these statements - particularly the notion that Antarctic sea ice is rougher - but also the evidence that the sea-ice structure is often different in the Antarctic compared to the Arctic.

We have underlined the fact that is is rougher and that is has a different structure by three references.

**(4.)** L53/54: "Beside MYI ..." --> It would be very important to underline that in fact a substantial amount of the MYI along the East Antarctic coast is actually fast ice. This is often true (older than 2 years old) multi-year ice and is of even larger importance for the ecosystem and has effects on buttressing the ice shelves.

We have inserted this information and added a reference (Massom et al., 2010).

**(5.)** L55-56: "pancake ice can form" --> Isn't this underestimating the fact that a lot of the Antarctic seasonal sea ice is actually formed via the so-called pancake ice cycle first published (in the 1990ties or late 1980ties?) by Lange et al. ?

We agree with the reviewer. Sea ice is formed in the Antarctic under turbulent atmospheric and weather conditions. Hence, pancake is common. Once again, it is possible to include pancake ice as a separate entity in ECICE using samples from authentic data. But this goes beyond the purpose of the manuscript. We admit, however, that pancake ice can be confused with MYI if radar data is used alone. But the combination with passive microwave can help. We have not done work to confirm this matter. We now mention the importance of pancake ice and refer to the Lange et al (1989) paper, and raise the topic again in section 3.2 (now L397)

**(6.)** L60/61: Please check whether it is really the sea ice type that is required or whether these models wouldn't primarily be happy with using improved data of the sea-ice thickness (distribution), the degree of deformation and the snow load. Also, when it comes to validate a climate model I suspect that there are very few that already provide "sea-ice type" as a variable. They might provide ice age though.

We have checked that. We have reworded the sentence, saying that detailed information, among the ice type and thickness, are needed to better validate and improve models. If a model provides, e.g. ice age, this can be set into relation with the ice types FYI and MYI.

We think that mapping ice types that are characterized by different thermodynamic and emissivity is what models really need. ECICE is a generic method, which is capable of producing this information. We have not tried it because we follow the traditional WMO age-based ice type so far.

**(7.)** L64++: "Recently, ..." --> While it is ok to already mention ECICE here, I ask you to provide a bit more background about algorithms that have been developed in the Arctic to separate FYI from MYI and to provide MYI concentration - first and foremost the NASA Team algorithm.

In addition to that, in order to put the value of your work into a wider context, I also ask the authors to provide more background about other attempts to discriminate between Arctic ice types. It is important that the reader understands that there is almost a full zoo of methods focusing on discriminating between different ice types in the Arctic - in addition to the NASA Team algorithm. To mention in addition to your and Ye's work is the work at met.no, at IFREMER, at BYU (David Long and his group) (and possibly others) that use coarse resolution satellite observations followed by the uncountable attempts to discriminate ice types using SAR. In contrast, activities in the Antarctic are very sparse.

You might argue that you are looking for ice type CONCENTRATION and not a simple discrimination. That is true, but even here your work is more upfront than any other work and this needs to be (implicitly) stressed.

You might also argue that ice type CONCENTRATION is the more important parameter, but if I understood your introduction so far correctly, then we are in need of ANY information about the ice-type distribution of Antarctic sea ice (other than land-fast sea ice), no matter whether this is a binary classification result or whether it is (already) an ice type concentration.

Because of this I ask you to one more time dig into the literature and try to find out what others did in this sector. If we omit polynyas / fast ice - for which a lot of studies exist - then there is not too many, perhaps add: Lythe et al., Classification of sea ice types in the Ross Sea, Antarctica from SAR and AVHRR imagery, International J. Remote Sensing, 20(15), 3073-3085, 1999, http://dx.doi.org/10.1080/014311699211624

and Ozsoy-Cicek et al., Intercomparisons of Antarctic sea ice types from visual ship, RADARSAT-1 SAR, Envisat ASAR, QuikSCAT, and AMSR-E satellite observations in the Bellingshausen Sea , Deep Sea Res. II, 58(9-10), 1092-1111, 2011, https://doi.org/10.1016/j.dsr2.2010.10.031    --> GC2

We have included a brief discussion on other attempts of ice type discrimination (but would refer to the introduction of Ye at al. 2016a and b for details about that in the Arctic). We have included the above references concerning  sea ice type discrimination in the Antarctic.

**(8.)** L84/85: "It takes input ... any given ice types" --> So, I can input 6.9 GHz AMSR2 TB H-pol and 5 GHz ASCAT observations and can obtain the partial concentration of pancake ice? Or I can input 91.6 GHz SSMIS TB at H- and V-Pol and get the partial concentration of MYI ice and FYI ice? If this is not the case then I suggest to re-write this sentence according to the actual capabilities of ECICE which seems to be oversold a bit here.

The sufficient number of channels is the necessary condition (number of input channels greater or equal number of surface types). ECICE is a generic algorithm, so we can in principle use any input channels. However, if the probability distributions of the different surface types in the channels do not differ enough, the retrieved results have very high uncertainty and might be meaningless.

**(9.)** L87/88: "to account for anomalies ... One anomaly causes ..." --> I suggest to re-phrase these statements. It is not clear what you mean by "observations". To me observations are the data you obtain from the satellites, i.e. brightness temperatures or backscatter coefficients. Hence, I ask myself what anomalies are in this regard? You possibly refer to those cases where ECICE fails to interpret the input satellite data into the correct total and/or partial ice concentrations, creating anomalous high or low concentrations and/or an anomalous misclassification of MYI as FYI and vice versa. Therefore it might be more correct to state that one set of satellite observations can be the result of several different combinations of physical parameters, causing ambiguous retrieval of total and/or partial concentrations when input into ECICE.

We rather meant "observations" in the sense of "anomalous ECICE results" and have rephrased accordingly.

**(10.)** L100-102: Please provide 2-4 references for publications that could underline your statement for sea-ice concentration and sea-ice type concentration - for both passive and active microwave observations.

We now refer to the overview paper by Ivanova et al. (2015): With the exception of ECICE, all 11 algorithms presented there use tie points, including all ``standard'' ones like NASA Team or Bootstrap algorithms.

**(11.)** L111-113: What happens, during the retrieval, if fractions do not add up to 1 and/or for fractions below 0 or above 1? Are these set to 1 (or 0) before the median of all realizations is computed?

This is avoided by introducing the inequality constraint in the constrained optimisation approach of ECICE, as mentioned in L111-113.

**(12.)** L116: How is the spread around the median computed? How many valid values are required for a median and its spread to be computed (assuming that not all 1000 realizations provide a valid result)?

This is explained in the original paper of ECICE (Shokr et al. 2008) and in the book "Sea ice: physics and remote sensing" (Shokr and Sinha, 2015, chapter 10). We think we do not have to repeat the information in order to not to distract the reader. However, we have introduced a few lines to explain, at the end of section 2.1.

**(13.)** Lines 119-124: I am missing the physics and references in this paragraph. What are the physical properties of the ice types that cause the different radiometric and backscattering properties that allow us to discriminate between the three ice types? Which of these are influenced by which snow physical properties that make MYI to look like FYI? How about the ambiguities between YI and FYI?

You use snow metamorphism only in the context of "return of cold temperatures" albeit snow metamorphism encloses a wide variety of changes of the snows' crystal structure and composition under the action of temperature, humidity and wind. This should be re-phrased. In addition "warm spells" only cause "snow wetness" to develop if the temperatures are high enough; still, even with considerable below freezing (-5 degC) temperatures snow metamorphism (rounding of grains, etc.) is present. --> GC1

These five lines are only the very brief introduction to the correction schemes described in detail in the next two subsections (including the references with more details on the correction schemes). Stating here that melt-refreeze causes snow metamorphism does of course not imply that snow metamorphism can only be caused by melt-refreeze- we have added a brief statement to clarify that.

**(14.)** L126-134: This paragraph describes the temperature correction as developed for Arctic conditions. You appear to adopt it 1-to-1 to Antarctic conditions as is indicated by the last sentence in this paragraph. Without an adequate introduction and review of the physical, radiometric and backscattering properties of Antarctic sea ice and its snow cover compared to the Arctic, this raises my concerns. On the one hand differ Antarctic MYI and partly also FYI physical and microwave properties from Arctic ones. On the other hand differ Antarctic snow properties often fundamentally from those in the Arctic - not to speak of the frequency with which the weather influences the microwave signature of Antarctic sea ice compared to the Arctic. I am wondering whether a close collaboration with specialists in this field would not substantially improve both, set up of the algorithm and interpretation of the results.

Using the same setting as for the Arctic is a preliminary approach. As stated in our response to GC4 (see above) this study is a "proof of concept". As to the underlying physics, see also the response to GC1, as to the choice of parameters, see items (19.4) and (20.).

**(15.)** L150-152: "this correction scheme ... to MYI" --> I suggest to separate this correction from the drift correction because it has nothing in common with it. I further suggest that you make clear which form of snow metamorphism you are refering to here. In L154 you introduce "HR" as being related to the "onset of snow melt" which, at first glimpse, would suggest an increase in snow wetness and hence elevated brightness temperatures, making MYI to look like FYI rather than the other way round as is stated here. You are possibly refering to melt-refreeze cycles or the like and need to specify this here to avoid confusion.

The additional correction referred to here needs the "MYI domain" of the so-called "drift correction" and is done in the same step, therefore we do not consider it a separate correction. Note that the "drift correction" (in spite of the name) also corrects for the effect of snow/ice metamorphism as it eliminates ice that "looks" to the microwave instruments like MYI but is not, which we have clarified now (end of section 2.2.2). In other words, the drift correction uses the ice drift data to make sure that MYI does not appear very far from its expected domain of expansion (it does not correct for drift or something like that). We have clarified this at the end of the section.

**(16.)** L150: To me "Ex-MYI" implies that this sea ice once was MYI and now is a different ice type. How about you name it "artificial MYI"?

We have looked for a more appropriate term and now call it "non-MYI", as all we know is that it is not MYI.

**(17.)** L166-169: I suggest to add the actual resolutions and sampling interval of the AMSR2 channels used.

Please provide information about the native spatial resolution of the ASCAT data and how you gridded these into the NSIDC grid of 12.5 km grid resolution. It appears to me that the statistics is different for these data than for the AMSR2 data because of the different viewing geometry and swath width.

What would also be important to know is whether the sigma_nought values were corrected towards a certain common incidence angle (e.g. 40 degrees)? If this is not the case, please provide a comment why you deemed that as not being necessary.

Yes, we agree this information is missing and have added it here (section 2.3, below equ. (2)).

**(18.)** L170-180: Your description about the choice of sample areas and time periods is not specific enough to my opinion. I have the following questions:

18.1) Apparently you used ASI SIC maps to define your sample areas. What is the requirement regarding the SIC to have a grid cell contributing to the sample?

Near 100% SIC, of course.

18.2) How did you define "beginning of the cold season"?

For the purpose of finding suitable sample areas, the beginning of the cold season is the time when regionally the sea ice concentration and extent start to grow again, this can be directly determined by visual inspection of the daily ASI maps.  We have included this information in a footnote.

18.3) For which time period (just 1 day?) did you select grid cells from the Weddell Sea defining the MYI distribution?

18.4) For which time period(s) and region(s) did you select grid cells defining the FYI distribution?

18.5) From which time period(s) and region(s) did you define the YI distributions based on the PSSM data set?

We have included additional information on that (new Appendix A)

18.6) How did you take into account the YI that develops during the ubiquitous pancake ice cycle in the MIZ that might cover several hundreds of kilometers? Isn't this, not the one growing in the polynyas, the far more relevant YI type in the Antarctic?

Good point. This is discussed in the discussion section. Taking this into account in order to get better distributions is the next thing to be done (after this proof of concept), mentioned in the list of future work at the end of the Summary/Conclusion section.

18.7) Where exactly, with respect to the ice edge, are your open water sample areas located?

18.8) How representative are the open water sample areas in the regions and months (March, Ross Sea; August, everywhere?) chosen for the weather influence?

One way to answer at least some of these questions would be to create a map in which you show the locations of the sample areas and, via color coding, the time-periods and/or frequency with which you used selected the data.

We have specified the details on the sample area selections and locations in Appendix A.

**(19.)** Table 2:

19.1)- I have concerns with two values in this table. Why do you define the END of the warm episode with a positive (2degC) air temperature? Is this a typo? If not it is absolutely not understandable and needs some justification.

There is actually a typo: the T1 must be -2°C and T2 +1°C. Note that the condition for the start of a warm spell is T>T1 *and* a reduction of MYI concentration in one day by 10%, and the condition for the end is T<1°C *and* a rise of MYI concentration by more then 10% in one day. Only in such a case, the MYI concentration is corrected. T1 and T2 have been found empirically, as described in detail by Ye (2016a). Note also that T1 and T2 are 2-metre air temperatures, not ice surface temperatures.

19.2)- What is the motivation for the very long maximum duration of the warm episode? This does not sound overly reasonable to me - neither for the Arctic nor for the Antarctic actually. I can guess that the length of this period is chosen this way because the melt and melt-refreeze processes change the physical and therefore microwave signature of the snow / sea ice system for a considerable number of days; even after freezing conditions have returned the modified microwave signature might still last (e.g. Voss et al.,  2003, in Polar Research and his work related to that).

Even though most warm episodes will last only few days, there is no harm in setting the *maximum* duration to 10 days in order to also catch rare longer events. It is computationally more expensive, but not in a significant way.

19.3)- Apart from that I am wondering how such a long maximum duration does match the comparably high frequency of warm events caused by cyclones passing over the sea ice.  I'd say that such events can be quite short-lived. Therefore, depending on whether you aim for a monthly or a daily ice type product one could recommend to use a considerably shorter maximum duration of such events of just 5 or even 3 days.

Yes, a reduction of the maximum duration will reduce the latency times if we want to do NRT data, but that was not a concern of the present study.

19.4)- In case you comment on the choice of these parameters later in the paper, i.e. in the context of the discussion, please point the reader that already here to increase the credibility of your choices.

See item (20.)

**(20.)** Table 3:

- I note that the choice of the values for these parameters specifically for Antarctic conditions has not been discussed and/or movitated so far. You might want to do that, please.

- In case you comment on the choice of these parameters later in the paper, i.e. in the context of the discussion, please point the reader that already here to increase the credibility of your choices.

We have mentioned that for the time being the same "tuning" parameters have been used as in the Arctic and fine-tuning might improve the results (section 2.3, L251ff.). However, other issues (pancake ice, outer Ross Sea) appear more important to be resolved first

**(21.)** L187-191: Three comments here:

21.1) What kept you from using ERA5 data? Is there are credible argument to stick to ERA-Interim data for surface temperature data in the Antarctic?

When most of the presented work was done, ERA-5 was not available yet.

21.2) Tschudi et al. (2016) appears to be a bit outdated given the fact that there is a version 4.1 of the NSIDC sea-ice motion data set, referenced as Tschudi et al. (2019 or even 2020).

We have updated the reference.

21.3) What is the motivation to use this rather low resolution OSI SAF sea-ice drift product? Doesn't it harmonize with the overall 12.5 km grid resolution you aim for much less than the NSIDC sea-ice drift product?

The source of the drift data was decided at the start of the study, several years ago. At that time, it seemed a good solution (probably,the temporal/seasonal coverage for Antarctic was better at that time). We are actually considering switching to NSIDC drift data for the future (see end of Summary/Conclusion).

**(22.)** L192: In Table 3 you mention the TB at 37 GHz, not 19 GHz; please check.

This is a typo. Thanks for finding it, we have corrected it.

**(23.)** Figure 1: I have a number of comments here; comment #1 and #2 are related directly to the figure content while comments #3 to #5 are related to the omission of relating the results shown to previous work.

23.1) What does the "Distr. set: AQ2" in the title of each panel refer to? Could it be removed?

(Note: now Figure 2) Yes, this was a working title which has been removed.

23.2) What is the statistics behind the data? What is the time period? At how many data per surface type do we look?

We now list these details in Appendix A.

23.3) What explains, to your opinion the fact, that GR3719 is a bit lower than is classically observed for open water and, particularly, for FYI (compare the tie point triangle used in the NASA-Team algorithm).

This probably depends on the choice of open water samples. We will elaborate on the sample choices anyway (see item 18. )

23.4) How do your values compare in general to tie points used by ordinary sea-ice concentration retrieval algorithms?

23.5) How do your backscatter values compare to values for C-Band radar backscatter of Antarctic sea ice cited in the literature?

We have included a brief discussion of these two points (L230ff.).

**(24.)**L201-203: While I am fine with using AMSR-E instead of AMSR2, I have concerns to simply replace ASCAT (C-Band) with QuikSCAT (Ku-Band) as signal penetration into and interaction with the snow / sea ice system differ - in addition to incidence angle and resolution.

Yes, correct, one cannot simply replace the scatterometer data. The respective distribution functions for the surface types are needed as well. Retrieval with QuickSCAT instead of ASCAT and with AMSR-E instead of AMSR2 has already been done (see Ye, 2016a) – we have pointed this out in the manuscript (end of section 2.3)

**(25.)** Instead of working with piece-wise available sea-ice drift products it might be a very good idea to use one consistent data set, namely the NSIDC one - unless you find an alternative with year-round coverage (IFREMER?); yes, NSIDC is not an optimal choice but

with that you avoid inconsistencies and jumps in your then much longer (by combining AMSR-E and AMSR2) time series. You might want to consider to simply delete these three lines here.

Yes, of course. If there is a consistent time series of drift data for the whole period, all data should be processed and reprocessed using that one. We keep theses three lines without extending them because we would just like to emphasize the potential/perspective of a longer time series.

**(26.)** Section 3.1:

(26.1)- How many Sentinel-1 SAR data from which dates were used? Where were these located (provide a map with the frames)? What was the time difference between SAR image acquisition and ECICE product? What is the "time stamp" of the ECICE products [0 UTC, 12 UTC]?

We use daily gridded brightness temperatures (see L166ff. of original manuscr.), interpolated from all swaths of one day (details on that will be included) – so there is no unique time stamp. This is now explained in the text.

(26.2)- Where were the Sentinel-1 SAR images taken from. Which type of SAR images was used (Wide Swath, Extended Wide Swath, ...)? How were the SAR image (pre-)processed for the evaluation? Was any drift correction applied to the SAR images?

We have included the missing information.

(26.3)- You decided to provide a qualitative intercomparison without computing radar backscatter (sigma_nought) values. Why? Wouldn't your results be much more credible and useful if you would come up with fractions of MYI and/or FYI derived based on a rough (by means of sigma_nought value) classification from the SAR images and compare those to the ECICE MYI concentration maps?

See response to item 26.7

(26.4)- L214-215: "In SAR images, MYI ... sub-surface layer" --> This very qualitative and not overly scientifically formulated sentence applies to the Arctic. Melt processes during summer in the Antarctic differ considerably from the Arctic and I doubt that one can speak of a "bubbly sub-surface" layer here. Please revise your wording taking int account the specifics of seasonal changes in microwave signatures in the Antarctic compared to the Arctic.

We have revised the sentence in question (now L284-288).

(26.5)- L221/222: "the iceberg ... as FYI" --> I don't agree. The SAR signatures inside that 70% FYI polygon encircling the iceberg are brighter than outside the polygon. The isoline does also not indicate at which side FYI concentrations are actually higher or lower. Given the fact that the area southwest of the iceberg is certainly dominated by FYI I suggest to re-phrase this statement along the lines that for that polygon both FYI and MYI concentrations are below 70% but that you don't know which is the dominant one. See also your Figure 5.

Yes, we agree and have corrected that error as suggested. Actually, looking at the individual maps of FYI and MYI, the icebergs tend to have 0% FYI concentration, which we now point out when discussing the comparison with SoD charts (now Figure 4).

(26.6)- For one grid cell, do partial concentrations sum up to 100%? I am asking because in the area indicated as > 50% YI fringing the Antarctic Peninsula there is evidence for MYI concentration > 50%. Did you actually check for maps like the one shown in Fig. 2 what the sum YI + FYI + MYI concentration is? It would interesting to see an example of this - perhaps in the appendix or in supplementary material.

See response to GC4 (4th paragraph). The sum is the total ice concentration (between 0% and 100%).

(26.7)- L222: This last sentence about the "quality" of this comparison I deem almost obsolete without information about how many SAR images of how many regions from which dates have actially been taken into account.

We will include more details about the number of scenes compared. We admit that the comparison is not very detailed, but rather an initial sanity check that focuses on the rather prominent MYI-FYI boundary usually found in the inner Weddell Sea – we will reword the sentence in this sense.

**(27.)** Section 3.2:

(27.1)- How are the weekly charts derived with respect to temporal availability of the input data? Is always the latest highest quality data set used for a respective grid cell (or pixel)? Or what is the compositing method used?

(27.2)- Does "microwave satellite imagers" include SAR? What is the dominant input data source for the charts you have used?

(27.3)- What does "analysis ... by experienced specialists" mean? Is this a manual analysis? Is the analysis done by one specialist or a team of specialists and what are the quality measures?

(27.4)- L228/229: What is the size of such a pixel? What is the grid that is used here? Is it the NSIDC polarstereographic one? Looking at Figures 3 and 4 I get the impression that in these ice charts the classification is not done pixel-by-pixel but rather in form of polygons that contain ice of similar characteristics and concentrations - such as done, e.g., by the Canadian and Danish Ice Services. Could you please check once again how the ice charts you show in your manuscript were generated, and if need be, re-phrase your description?

(27.5)- Were the input data projected into a common grid prior to ice chart generation?

(on 27.1 – 5) Ice charts are generated based on SAR image analysis as the prime data sources but combined with many sources of ancillary information. SAR images are analysed visually but ice analysis operators. One operator analyses each image, hence the analysis is subjective. However, those analysts are well trained and experienced. The ancillary data include climatic information about the area, the recent history of the ice filed, meteorological data, observations from ships and ice breakers Such charts consist of polygons, not pixels, thus we have reworded the sentence mentioning "pixels" which solves item (27.4)

(27.6)- L234/235: What is your "cold season"? What do you mean by "sporadic comparisons with data from other years"? How many, from where and which dates were these additional comparisons?

By "cold season" here we mean the period when the algorithm works because there is no widespread and sustained surface melt, i.e. March to October (inserted in the text now). We still consider adding all comparisons as supplementary data.

(27.7)- L243-246: Your observations of the different labeling of ice as FYI or MYI between AARI and NIC charts could be the result of different definitions of when FYI is re-labelled MYI ice by the producing agencies? Did you check that? I note that a switch in March / April disagrees with the WMO recommendation you mentioned further up in your manuscript.

We think this rather shows that the two different teams of ice analysts came up with different assessments of their data, maybe also because they use different data. This is hard to find out.

(27.8)- I note in addition that there are more fundamental differences between the AARI and the NIC ice charts in the Eastern Weddell Sea regarding the location of YI and FYI.

See above...

**(28.)** Figure 3:

- I suggest to use a title for the MYI concentration that is consistent with the other two ECICE results.

(Note: now Figure 4) We have removed the small titles of the sublplot and instead put the labels YI, FYI and MYIc into the centre of the subplots.

(28.2) - Putting the legend of the AARI ice chart into appendix is not a good solution. I suggest the following: You crop all maps to an area that excludes all the annotations in the AARI ice chart, put all ECICE results into the second row of panels and put the ice chart legend in the first row of panels next to the ice chart. In the second row of panels you could then also follow your approach from Fig. 5 and provide one legend with the title "Ice-type concentration", marking the ice type itself in the map (actually it is in the panels' titles but perhaps you consider to remove these anyways.).

We have considered the suggestion. However, note that the SoD charts use one "color family" for each of the three broad ice types: pink/purple hues for YI, yellow/green hues for FYI, brown hues for MYI, and this is the information needed here (given in the caption). We think the full color scale with all the subtypes would be too distracting here.

(28.3) - Finally, I note that you seem to use an old land mask to mask out Antarctica, still containing an overly long "Trolltunga" of the Fimbul Ice Shelf and the Mertz Ice Shelf. Given the fact that you focus here on AMSR2 data it might be a very good idea to use an more recent and hence more accurate land mask.

We have used the land mask that comes with the AMSR2 data (!), but see the need to improve that in the future.

(28.4) - L258 / Fig. 5: "with the summation of their fractions equal to 100%" --> When I look at Figure 5 I doubt that this statement holds. I guess it needs to be replaced by the actual total sea ice concentration that is obtained with the ECICE algorithm because there are quite some areas downstream of the Ronne-Filcher Ice Shelf polynya where YI conc. + FYI conc. + MYI conc. add up to something between 80 and 90%. I am sure you will get back to this in the discussion section. But it certainly does not hurt to either state that "theoretically" the partial concentration should add up to 100% but that this is not always the case, or correct your writing accordingly towards that the sum of the partial ice concentrations adds (of course) only up to the actually existing amount of sea ice. --> GC4

(Note: now Figure 6) As stated before, the sum is 100% only if the open water fraction is added as well. We have inserted "(including the open-water fraction)" to avoid misunderstandings.

(28.5) -L261/262 / Figure 5: Your maps do also reveal that ECICE seems to have a problem discriminating between YI and MYI because the area just next to the Ronne-Filcher Ice Shelf appears to be characterised by some YI, no FYI and some MYI as one can observe a fringe of non-zero MYI concentration in that area.

(Note: now Figure 6) This is another hint at revising/improving the input parameter distribution, in particular of YI - this is now mentioned in the discussion (now section 3.3, L394ff.).

**(29.)** Section 3.3:

(29.1) - Please provide more information about the PSSM maps. What are the grey areas masking parts of the maps shown? Where did you get the data from? What is their temporal and spatial resolution? How many of these maps did you use for which regions? The scope of this part of your intercomparison remains vague.

The grey areas are masked-out areas (land, of course, and apparently the sea further away from coasts, depending on the definition of the regions covered) - we now mention this in the text and the figure captions. The source of the data was already given above, section 2.3, (originally L175-180) and has been extended for the revision. We will put in a back-reference here.

(29.2) - Given the fact that you look at years 2017 and 2018 I assume it is SSMIS data and not SSM/I anymore, am I correct? Please correct your writing accordingly.

Yes, we have corrected that.

(29.3) - I would appreciate if you could comment on the quality and limitations of the PSSM based ice type maps. What the approximate thickness limit between thin ice and "other ice"? Does "thin ice" mean that there is 100% thin ice or could this potentially also be 50% thicker ice interspersed with open water?

We have inserted this information. The thickness limit is, according to the data provider (ICDC, CEN, Univ. of Hamburg), 10–20 cm.

**(30.)** Figures 6 through 9:

(30.1) - I suggest to reduce the size of the panels considerably. In particular I recommend to make the PSSM map the same size as the white box shown in the left panel denoting its location. Even better would be if you'd crop the maps in the left panels to the size of the PSSM map. That way would would be able to reduce the number of these figures from 4 to 2 or perhaps even 1. Did you try, in this context, to combine the information from both panels into one? Perhaps by extracting isolines from the PSSM maps and superpose these onto the YI concentration maps?

(Note: now Figures 7 to 10) We have considered the suggestions. However, If we crop the YI map on the left, to just the white box, it is not easy to see which part of the Antarctic is shown, therefore, we would rather keep the large map. If, instead, we reduce the size or the PSSM map on the right, we would not really save space as the smaller maps also need the right half of the figures, so why reduce the size.

(30.2) - Did you chose the dates shown in the manuscript arbitrarily? If so, make a note.

(30.3) - Were these the only PSSM maps you considered in your comparison? If not how did the comparison go for all the other maps? Did you derive any quantitative information?

We just show a few representative examples.

(30.4) - Looking at these YI concentration maps reminds me one more time the issue of how the ECICE ice-type concentration maps deal with cases of considerably less than 100% total sea-ice concentration because I note that all the YI concentration maps shown in Fig. 6-9 reveal lower YI concentrations in the core of the polynyas.

The partial ice concentrations of the three ice types sum up to the total ice concentration which is between 0% and 100% (this is now explicitly mentioned in section 2.1). We do not see a problem here.

**(31.)** L280: "given the limitation that a rigorous validation ..." --> Given the fact that you kind of advertise the data set obtained with this paper and given the fact that this is first attempt to provide such a data set, I don't take it as a positive sign of credibility of the data set produced, when this paper only deals with a very general, little quantitative evaluation. The results presented are partly very vague and the description of the physical background being the foundation for the approach used and the data set is not overly exhaustive and - at least for me - not convincing.

See response to GC4 (aim: proof of concept)

**(32.)** L281: "Large icebergs are often erroneously retrieved as FYI" --> Is this your result? You could state this more clearly. But, when doing so, please take into account my comment made to Fig. 2 with respect to this issue.

As already mentioned above (see item 26.5), this was an error on our side. We have corrected that statement (now L358ff.), and emphasised that icebergs and their discriminations are not our focus here.

L288-300: I was kind of expecting that you would run into problems with weather-induced variations in the snow physical properties and resulting microwave signatures. Since in your manuscript the physical foundation and description of the processes and properties resultung in specific microwave signatures is not overly detailed and mature, it is of course difficult to discuss these observations. I find that your attempt to explain your observations go into the correct direction but are far from being conclusive and is too vague. I'd say you could delineate the reasons that caused the MYI concentration over-estimation much better and much more specifically by means of checking the input data values and compare these with what is known from literature. It might make sense to take into account ERA-Interim and/or ERA5 data (you use them anyways) to discuss you observations also in the context of melt-refreeze, ice-snow interface flooding, slush refreezing, snow-ice formation and the like. I again recommend to take a look at the work of Voss et al. (2003) and the related doctoral thesis.

As to the physical foundations, see our response to GC1. We admit that we can improve on the interpretation of the results and thank the reviewer for the suggestions. However, any erroneous increase of MYI during the cold season because of changes in the snow and ice properties should be removed or at least greatly reduced by the drift correction — therefore our initial concern is why that does not work which has nothing to do with the physical properties of the sea ice and snow. As already mentioned later in the manuscript (L310ff. in original version), instead of working on the correction scheme, it might make more sense to prevent the misclassification by ECICE, which means to use better samples representing YI, FYI (and MYI) — here knowledge of the range of radiometric and scattering properties of FYI might indeed be useful.

**(33.)** Figure 11: Looking at that figure again makes me to think whether you ever tried to look at maps of YI conc + FYI conc + MYI conc? It appears to me that there are patches of spuriously large  MYI concentration that coincide with a total sum of partial concentrations above 100%.

(Note: now Figure 12) Well, as we only modify the MYI concentration in the correction schemes. which breaks the sum rule that the partial concentration *plus the open water fraction* add up to 100%. The problem is that we have more than 2 ice types, so if we increase/reduce MYI concentration, it is not clear how to distribute the corresponding  reduction/increase to FYI and YI, so we refrain from it. If we just had one ice type apart from MYI, it would be easy to preserve the sum by just mirroring changes of the MYI concentration.We have, by the way, checked that the sum of the uncorrected ice types is indeed the total ice concentration (see also the new section 3.3 on the time series).

**(34.)** L299: None of the references listed in this line deal with pancake ice and its backscatter. These are all references dealing with the snow cover and should be put into L298 behind"... MYI in that respect."

Yes, corrected.

**(35.)** L292: How credible are - to your opinion - these MYI occurrences "far offshore in the outer Ross Sea"? Which process can cause these?

The "streaks " of MYI be found in some years in this area (according to theNSIDC/AARI charts) seem to have drifted there from the East, so they originate as MYI near the coast of Wilkes Land and get to the outer Ross Sea by advection, which seems quite possible. However, their total area must be equal or less than the area of MYI of the source region.

Final question to L288-300: How did you compute the total MYI area shown in Fig. 10? Did you apply a threshold MYI concentration or did you count from 1 % onwards? What did you use as gridcell area to compute the total area?

As stated, we calculated the area,  which means we take into account the MYI concentration from 0% to 100% and the (space-dependent!) grid cell sizes of the polar stereographic (NSIDC) grid (clarified, now L365ff.).

**(36.)** L302-303: You can look yourself into the likelihood of (1) by checking the drift data you used. How did you cope with data gaps in the drift product? Did you include the quality flags?

Well, data gaps (e.g. NaN values because of insufficient cross-correlation) will cause the MYI domain to not be extended, causing a rather strict correction. As mentioned below (item 38.), the drift data are the next thing to be updated.

**(37.)** L306-307: "such seeding points" --> please explain this in more detail or delete it. Questions I would have is how this happens and why this should have an influence on the MYI concentration in particular and not on the other partial concentrations.

We have improved he explanation (L384ff.).

**(38.)** L309/310: I don't understand why you refer to an observation of Ted Makysm when you yourself used the data for the drift correction. Didn't you yourself take a look at the data once you suspected that these could include spurious drift estimates? This is inconclusive.

As now stated in the discussion, before doing extensive error search in the drift correction, the thing to do first is improve the distributions of YI and MYI in order to mitigate blatant misclassifications.

The next step would be then to use a newer version of drift data (OSI SAF or NSIDC); this is mentioned in the Summary/Conclusion.

**(39.)** Line 311-314: I suggest to not look into the data used but first try to understand which sea ice and snow physical properties you encounter during the course of one cold season and to further understand how the microwave signature looks like. This might require to look into 1-dimensional numerical modelling of microwave emissivities and of microwave backscatter as a function of sea ice and snow properties. There is a paper by Willmes et al. (2014) in the Cryosphere and there is work by Tonboe et al. that might help here.

We will look into that matter, but a dedicated modelling study is clearly beyond the scope of this manuscript (but is worth doing in the near future).

**(40.)** L319-322: "The most likely reason ..." --> While your observation from Fig. 10 seems to be credible, I am wondering whether this isn't an over-simplification of the situation. I agree, wettening of the snow cover can mask MYI so that it looks like FYI. But at the same time the re-freezing of the slush at the ice-snow interface, ice lenses, whatsoever causing larger grain sizes can have the adverse effect and making FYI looking like MYI. A deep snow pack and/or substantial deformation of FYI has the same effect as demonstrated by one of the co-authors for the Arctic ocean. In addition, and here the authors were right earlier of course, pancake ice is a nasty fellow and could possibly also likely to be misclassified as either of the two thicker ice types - adding to their partial concentration.

We agree with the reviewer's points. The physical processes of the snow and ice modulates the radiometric and the scattering data. However, the advantages of using the probability distributions of all possible values of a given observation from a given ice type warrants the inclusions of all possible conditions. Sure, the snow wetness and refreezing changes the observations but if the input distributions encompass all the possible changes, then the correct classification is warranted. As for the point of possible misclassification of pancake ice as MYI, we have not considered it. The two entities may not be misclassified using the present data set because while they have nearly same backscatter, their radiometric emission is different. This is an advantage of using the combination of passive and active microwave. Pancake ice is not part of the purpose of the study but its confusion with MYI should be considered. Here, ancillary information is required to avoid the inclusion of pancake ice areas. See also response to (18.6) above.

**(41.)** L323-326: Two more thoughts on this: Beginning in October the expansion of the Antarctic ice cover stops and the lateral movements switches to a retreat / compaction type. In addition, due to the dispersion the fraction of MYI per grid cell has decreased to a value that is likely not large enough anymore to be adequately detected by ECICE. I am sure this is something you can check in your data. One could hypothesize that computing the total ECICE MYI area is reasonable as long as ECICE is capable to derive the MYI concentration with high accuracy ... which I doubt is the case when the partial concentration has fallen below 30% and when the MYI coverage has dispersed in many small floes embedded in a mixture of YI and FYI.

Also any MYI that has arrived in the MIZ (in the Weddell Sea) is now likely to melt as air temperatures are not cold enough anymore out there to keep it alive. From that point of view I find a rather decay of the MYI area in the September / October time frame not overly surprizing.

The accuracy of the results from ECICE has not been estimated quantitatively because this requires in situ observations. This statement applies to other algorithms too (e.g., MYI concentration from NASA Team algorithm). We do not know the minimum concentration that can be estimated but the manuscript

provides data about the entire range of concentrations. Operational ice charts cannot be used for this purpose (in our opinion) because of their coarse resolution, subjective method and most importantly the conservative estimates in these charts. Given all this, we don't think that the partial concentration is inaccurate if it falls below 30%. The limit should be lower than that.

**(42.)** L337: "and melt" --> Where did I find examples of these in your manuscript?

Maybe the figure in the manuscript do not show this very well. When looking at the daily evolution of MYI over an entire season, this is quite obvious. In view of this comment, we still consider including an animation of one season in the supplementary material.

**(43.)** L343-344: "The new time series ... outweighs the shortcomings that still exist." --> I do not agree to this statement because of 1) the unmature physical foundation, 2) the vague interpretation of spurious ice type concentrations and 3) the very qualitative evaluation.

We still think that this data set, which we will term "preliminary", has its merits (it has actually already been used by Antarctic Cruises of the University of Cape Town) and serves as a proof of concept (see also response to GC4). We have rephrased accordingly.

**Typos / Editoral Comments:**

L27: I suggest to look for a more recent paper making this statement, e.g. Kwok 2018 in Environmental Research Letters.
L30: There should be another reference from Parkinson and DiGirolamo from 2022 in Remote Sensing of Environment.
L49: Typo: "...sea ice For ..." --> "... sea ice. For ..."
L79: "existing a ice chart" --> "existing ice chart"
L114: Typo: "coast" --> "cost"
L152/153: Please explain all the mathematical expressions that are used here for the first time.
L159-164: You have introduced the sensors' acronyms further up and can omit that here.

L207: I am sure a reader would appreciate to see 1-2 references here.  - Validation is just starting, no references yet..

L210/211: It might make sense to mention already here that the polynya maps used in the comparison are from a different year than those used for algorithm tuning.
L222: "Sentinel-1 scenes" --> "Sentinel-1 SAR scenes"
L256: Typo: "forth" --> "fourth"; see also L262
L266: Typo: "where" --> "were"
L286: "often" --> since you deal with a limited number of years in this paper you could perhaps mention all years during which you observe this increase.
L332: "outside the melt season" --> "during the freezing season"
"spatial" --> "grid"
L334/335: "... is well captured" --> You could add "by our ECICE results" to make clear that this is your result.
L334: "... in the Antarctic" --> add: "in addition to ship-based observations of the ice conditions."

We have corrected the listed typos and errors

**Response to RC2:**

Note that we quote the reviewer's comments and suggestions in red.

**Major comments:**

**Writing style**

The manuscript would benefit from (and strongly needs) a thorough tightening up of the writing style of the whole manuscript, including but not limited to:

● Use a clear structure with separated sections for the data and the methodology. At present, the input data and input products are presented "here and there" within the method section.

We have tried to make a clearer separation between data and methodology. Having two separate sections "Data" and "Methods", however, does not seem very useful to us, as mentioning all data *before* explaining the method is much information that cannot be used yet, and mentioning the data *after* describing the method would be too late. We have included a flow chart (at the end of section 2) that shows the methods (ECICE and the correction schemes) and the various input data and their flow.

● Avoid repetitions. Several repetitions occur and sometimes a simple re-ordering of the sentences would make the reading flow better

● Make sure that information given across the manuscript is in synergy with itself. For instance, it is very unclear whether the new product covers 2013-2019, 2013-2020, or 2013-present.

It is actually 2013 to 2021. Corrected in text.

● Make use of general spell checking.

We have tried to straighten the text (at the same time, however, incorporating a large number of suggestions by another reviewer), eliminated inconsistencies like the one mentioned, and, of course, have applied a spell checker again.

The presentation of this work should be clearly separated from possible improvements or potential future works. Future works or possible upgrades should rather be listed and discussed in a discussion section or the Conclusion.

We think that mentioning some possible improvements/potential future works in the course of the paper is unavoidable: For example. when discussing the wrong multiyear ice in the Ross Sea, we of course mention possible ways to mitigate that (L311 ff.). However, we agree that it makes sense to list all possible improvements and future work in one place and now do so at the end of the final Summary/Conclusion section.

**Algorithm presentation vs presentation of long-term time series**

The manuscript seems to have two main goals. On the one hand, it presents a methodology for mapping the Antarctic sea ice type from remote sensing data. And on the other hand, it presents, for the first time, a longer time series of Antarctic sea ice types. Both these topics are very relevant, however, at the present stage, the manuscript covers these in an inadequate manner.
In order to be a manuscript presenting a new method, the methodology is only superficially described. Below are some specific points to be considered:

We did not want to repeat the description of the ECICE algorithm and the two correction schemes here, as they are described in the respective publications, but we will try to follow the suggestions below.

● I suggest naming the full algorithm which is implemented at the University of Bremen. When reading the manuscript, it is unclear if "the new method" is ECICE, modified ECICE, or ECICE + post-processing. Examples of this:

○ L64: "Recently, a method has been developed ...". Which method is this? Add a reference.

(The reference is already at the end of that sentence.)

○ L66-69: "The method is based on ECICE ... and a later modification ..."
Unclear if the ECICE has been modified, or if post-processing includes
modifications of the outcome...
○ L72-74: "In this study, we have adapted this method to the Antarctic
conditions ...". Again unclear what "this method" is.
○ L85-86: "Our estimation of MYI concentration actually is a two-step procedure
that first uses ECICE and then applies two correction schemes ..." Assuming
that "our estimation" is coming from "the new method", here for the first time it
seems clear that the method is in fact the ECICE retrieval pluss some
post-processing (correction schemes).
○ L193: "The final result of the two-step retrival scheme ...".
○ etc.

Thank you for pointing this out. We agree that the naming must be consistent throughout the manuscript and have tried our best to do that.

● L114-116: it is mentioned that "the median is used as a measure of confidence of the
result for each surface type". However, this confidence field is never presented and it
is not clear if this information is provided together with the ice type product.

This information is saved along with the results of ECICE (we have included this information)

● Section 2.1 is describing the core/backbone of the classification algorithm. I would
have liked to see a few equations or illustrations (e.g. flow diagram) of the ECICE
methodology. Especially, the paragraph relevant for the Antarctic adaption (L105-117)
is hard to read as it is and would benefit from supplementing equations/illustrations.

We are now including a flow chart of ECICE and the correction schemes that in particular shows the various input data (new Figure 1). The ECICE algorithm is described in detail in the cited publications (Shokr et al., 2008 ; Shokr and Agnew, 2013) and we do not want to repeat too much of that here.

● L166: "For all input parameters, we use daily gridded data". How did you arrive at
these gridded data? Especially, for scatteromter data, a sentence on how the
angle-dependent swath data are gridded would be relevant.

We combine all AMSR swaths of one day and then interpolate to the grid using a distance-weighted near-neighbour approach (the one from Generic Mapping Tools with four sectors). The ASCAT swath data are converted to common incidence angle of 40° and then interpolated to the grid. We have added these details (section 2.3, below equ.(2)).

● L175: "Later in the season, sea ice that has formed ... away from MYI is FYI". How
do you account for the changing position of MYI during the season and thereby
collect only FYI data?

We made sure the used FYI areas are so far away from the start-of-the-season MYI that the latter cannot have drifted there. If we assume, e.g., 15 km of maximum daily drift (rough estimate from 1 year of OSISAF drift data), this would mean, e.g., at most 1500 km in June. We have added a short explanation.

● L196-197: "... retrieved the ice type concentrations for the months of Feb to Nov ...".
I did not find any comment on why summer months are omitted, or why exactly this
period has been chosen for the Antarctic product.

In general, under permanent melting conditions, the radiometric/backscattering properties of sea ice change considerably and differences between the ice type diminish or even vanish. Therefore using ECICE in summer does not yield reasonable ice types. We apologise for not having stated this important fact and have added this explanation to section 2.1 on ECICE.

● The final output is "corrected MYI" (L193), however the "uncorrected FYI, MYI, and YI" are also provided.
○ It is not clear if the uncorrected fields are "pure ECICE" outcome?
Yes, the uncorrected fields are the "pure ECICE" outcome. We added this were we mention the preliminary ice types (now L254ff., "without applying and correction")

○ uncorrected MYI (or Ex-MYI) is never presented, and maybe they should be shown?
We have actually already considered that and decided against it as Ex-MYI should first be investigated in more detail. We have renamed it "non-MYI" and now explain a bit more about it where is it first introduced (section 2.2.2, L177ff.).

Please note that other ice concentration algorithms that produce MYI do not apply corrections to account for anomalies in the locations of the MYI. The correction scheme used in ECICE can be used in any algorithm.

○ Could you include a comment whether MYIcorrected+FYI+YI or MYI+FYI+YI add opp to 100%? And if not, please comment on this as well.
The uncorrected ice type concentrations add up to the total ice concentration (which can be between 0% and 100%, of course), When correcting MYI, the amount added or subtracted is not subtracted or added from FYI or YI as we cannot say to which of the latter two it "belongs". Hence, MYIcorrected+FYI+YI cannot add up to the total ice concentration. We have added this explanation to the text.

● It is not clear if a threshold is used for the ice edge? Or are all surfaces with >0% ice concentration classified?
We do not use any threshold, but directly retrieve the concentration of the three ice types everywhere. In areas of open water (100%), all ice types have 0% of course.

● Several places it is announced that this computation can be extended to present time. What will the latency be for such a retrieval?
We have since implemented daily retrieval. The ECICE output is in near real time, within 1 day of receiving AMSR2 and ASCAT data. The latency of the corrected MYI is 16 days. We now mention this at the end of section 2.3.

In order to be a manuscript presentating a longer time series, it is surprising to see that there is a complete lack of presenting or showing any long-term (seasonal, interannual, regional) behaviour or variability. Only a few hand-picked days are shown and the year 2018 MYI total area time series. Since this is the first time a longer time-series of ice type is presented for the Antarctic, then it would be appropriate to show a full record plot and potentially discuss any trends and variabilities, (or missing trends).
We have added a section (3.3 Time Series 2013-2021) showing complete (2013-2021) time series and discuss them.

Several places it is mentioned the possibility of a record covering the period 2002 to present. However, the present record covers the period 2013-2019. Can you make this more clear - what defines the period you present and why is this period selected? And hereafter (best fitted in a discussion or conclusion section) mention possibilities for extending the time series and what this would require and what is the timeline for implementing this.
We have now stated more clearly (end of section 2.3): The period starts with the availability of AMSR2 data, in principle in July 2012. As the drift correction scheme needs to starts at the beginning of the cold season, we start to retrieve in 2013. The end of the period is now 2021 – we will change the text accordingly.

**Other comments**

General comment for the figures: Could you consider to label the sub-figures and thereby avoid " top-right", etc

The use of "top right" etc. seems common practice in many publications. However, we have now labeled the four panels with (a) to (d) which makes referencing easier.

Figure 1: Why not simply add the sensor type in the title of each subfigure instead of the "Dist. set: Aq2" which is a title that does not give any sense.

(Note: now Figure 2) We have removed the working title "Dist. set: Aq2". The parameter shown is clearly named in the *x*-axis labels.

Figure 1: Some of the shown density distributions clearly shows a double-peak. Can you comment on this and could this in fact indicate that more types are represented by the distribution?

(Note: now Figure 2) Yes, e.g. the distribution of MYI in GR17,37V, the red curve in the lower left plot, shows two almost distinct peaks. This might point at two subtypes. However, in order to retrieve one more ice type (by splitting "MYI" into two subtypes), ECICE would also need one more input channels as the number of input channels must be equal or larger than the number of retrieved surface types. We are not sure if is makes sense to add this discussion into the manuscript, though.

Figure 2: I find it difficult to locate this region. Either you could add an overview map or with words explain better where this is. E.g. " the inner part of ..."

(Note: now Figure 3)The region is explained in the caption ("Southeastern Weddell Sea, bordering the Antarctic Peninsula).

Figure 3: the sub-titles should be upgraded, preferably with one sub-title for each subplot. Also the sub-title of the lower-right should be updated/corrected.

(Note: now Figure 4) Usually there is just one caption under a figure. But we have put labels (YI, FYI, MYIc) into the centre of the subplots.

Figure 6-9: Here is used four figures for illustrating Young Ice. Potentially, these could be merge into fewer figures. The PSSM maps - the gray color should be defined in the caption. Even better, if the coast/land could be added for better orientation of the sub section.

(Note: now Figure 7-10) We have considered combining the figures, but think that this does not make them better to read. The meaning of the grey areas is now explained in the captions and in the text.

Figure 10/L286: What is shown in this figure? Is it the extent of all ice pixels that contain some concentration of MYI? Or is the MYI concentration or the full ice concentration taken into account?

(Note: now Figure 11) As mentioned in the caption, this is the full MYI area, taking the MYI ice concentration of each grid cell (0% to 100%) and the (variable) grid cell area into account. We now explicitly explain that in the text. So this is *not* the extent, which is commonly defined as the sum of the full area of all pixels with an ice concentration above 15%.

General comment for the tables: The layout of the table caption differs between the tables - some times it appears above and sometimes below.

Has been fixed.

**General comments:**

L25: you could add a reference to the YI definition (e.g. WMO Sea-Ice Nomenclature 2017)

Yes, we add reference to the WMO Sea Ice Nomenclature when we first introduce the ice types.

L33: explain or add a reference to why all MYI is SYI.

We will add the remark that almost all Antarctic sea ice will drift out into lower latitudes and meld within two years (now L36ff.).

L38-39: "Antarctic sea ice has strong region-dependent ..." This sentence stands a bit alone.
Could you say a few more words on this?
As this statement is not really relevant here, we rather delete it.

L43: Move "in the Antarctic" to the beginning: "Sea ice cover in the Antarctic..."
Yes.

L51: repeats that MYI typically is in Weddell Sea (L31)
Yes, but we see no problem in repeating this statement here as we here discuss the importance of the Weddell Sea in particular.

L62: "Total and partial sea ice concentration ..." when first time reading this, it was not clear
that "partial sea ice concentration" was referring to the ice types. I suggest re-wording this
whole paragraph.
We have reworded this: "The concentration of total sea ice and of the sea ice types"

L72: "Ye et al., 2019" is not accepted for publication. Can you find another reference?
A revision of the cited study is in progress, for the time being, there is only the discussion paper.

L74: replace "regularly" with "operationally"
Yes

L77: "brief account of ECICE and its adaption to Antarctic ..." Would it be more correct with
"brief account of ECICE, implemented correction schemes, and the adaption to Antarctic ..."
Yes, of course — we have corrected this.

L78: remove "first"?
L78-79: since the entire record is never presented, I would suggest deleting the period.
Simply "In sec 3, the outcome of the Antarctic sea ice type concentration mapping is
compared with ..."
We have rewritten the sentence as:  "In Section 3, results of the Antarctic sea ice type concentration mapping are compared with results from..."

L91: Any reason why SSM/I can be included but not SSMIS (and other passive microwave
radiometers)? Why mention SSM/I if it is not included in the present method/production?
After 2002, the preferred satellite instruments to be used are AMSR-E and AMSR2 as they have higher resolution than SSM/I and SSMIS. SSM/I was mentioned as it can extend the record backwards before the AMSR-E era. SSMIS can, of course also be used, and actually it can close the gap between AMSR-E (until Oct 2011) and AMSR2 (from July 2012). We have mentioned this here (beginning of section 2.1).

L100: "Most methods" please include references.
We have included the reference to the ice concentration algorithm comparison paper by Ivanova et al., (2015), this saves specifying 10 extra references.

L132: "If MYI .. drops at any location during a warm spell ...". Are there not any restrictions
on this drop to occur in the vicinity of where the warm anomaly appears?
We actually meant "at any location affected by the warm spell" and and have corrected the text accordingly.

L140: "After that, MYI can only drift ..." What exactly do you mean by this? If you mean that
no more MYI will be created (per theoretical definition) after this point, then say this more
clearly. In the same sentence is used the word "melting" which is not a part of "drifting"...

Please re-phrase this sentence.
We have slightly reworded the sentence: "After that, during the cold season, no new MYI can be generated. MYI can then only drift, and its concentration can only be changed by divergence, convergence, and melting."

L144: "... boundary of MYI cover ..." Please define the boundary of MYI cover. Is this where MYI conc = 0%?
We use a threshold of 20% MYI concentration to define the MYI boundary. We have inserted this information in the manuscript (now L171).

L153: "... sudden reductions ...", sudden reductions in time, I assume?
Yes, we use "sudden" in the standard, temporal, meaning, as the specification "(within one day)" suggests.

L156: "The values of the parameters ...". Please indicate from where these values are taken, e.g. include reference or discussion on how they have been chosen.
The values were empirically determined for the correction schemes in the Arctic and have been kept here. This is explained in Section 2.3. where we now also state why we have kept the Arctic values.

L160: Why is AMSR-E presented here when the ice type record covers 2013-2019?
AMSR-E data will be used in the next step. However, using them has already been implemented.

L167-168: Is the Melsheimer reference the right reference to add just after NSIDC?
Yes, it is, as the NSIDC grid in the context of sea ice type retrieval is described in more detail there.

L168-169: Could you please elaborate a bit more on this, e.g. by simply presenting the approximate spatial resolution of the used input data?
Yes, more explicit information has been inserted (section 2.3, below equ.(2)).

L174: As "ASI" is used only once, I suggest to just fully write the full name here, for easier readabilty.
This is why we have put the full name and the link and reference into a footnote.

L174-175: It is a bit unclear from what seasons the training data is collected. Is MYI data collected from only beginning of the freezing period. Please give a bit more details.
The MYI data are collected from the first months of the freezing season. There is now a new appendix (Appendix A) that gives details about the sample data.

L188: Are there any reasons for using ERA Interim instead of the newer ERA5?
When most of the presented work was done, ERA-5 was not available yet.

L190: When mentioning the potential NSIDC ice drift data, please include a comment on why OSI SAF ice drift data are chosen to be used, and whether NSIDC data have been tested out in the ice type retrieval. Also, please note that an ice drift climate data record from OSI SAF is in the pipeline for this spring 2022 (regarding L202-203).
NSIDC drift data have been used already for retrieval in the Arctic (see Ye at al., 2016b). At the time this study was started, the used OSISAF data seemed the best choice, but we are actually considering switching. This has been included in the text (L247, and at end of Summary/Conclusion)

L191-192: this is a repetion
Well, we once more reference Tables 2 and 3 (after referencing them in sections 2.2.1 and 2.2.2) and see no harm in that.

L218: Are these threshold procentages randomely chosen, or can you comment on why 50% is used for YI and 70% for the others.
They were empirically chosen.

L225: To my knowledge, NIS do not produce Stage of Development maps. Plese check this out.

NIS of the Norwegian Met. Institute is at least one of the three partners of that project. As far as we know, NIS has contributed some regional ice charts (but not SoD?). We have now left out NMI/NIS

L225: To my knowledge, no SoD maps are available on the webpage in 2014. Please chech this out.

Thank you for pointing this out, concentration maps start in December 2014, but SoD maps only in May 2015. We have corrected that.

L231-233: Is this information relevant (and is it true in practice?)?

We considered this information necessary because it makes some MYI (in the first half of the season, until end of June) appear in similar colours as FYI.

L235-236: when you write "an overall correspondence" - is this referring to results shown here, or did you check all available charts against the product. Please give more information on what is in overall correspondence and how this has been concluded.

We have compared at least one SoD chart per month (there is one per week) for two entire seasons.We consider publishing these comparisons as supplementary data. We will elaborate on that.

L279-280: The contribution from Ex-MYI is not shown or taken into account here. Please include this contribution to the discussion of FYI and YI.

As already mentioned above, we had decided against showing Ex-MYI (now called non-MYI, see above) as we have not investigated it thoroughly. We can only say that it is actually FYI or YI (but cannot tell which).We have added this information to the text where this is first discussed in section 2.2.2

Yes.

Yes, we have tried to rephrase and improve it (now L394ff.).

L315-316: I suggest that you include some mapping examples from September/October to better visualize for the reader why you see an increase in the MYI area.

In L315-316, we speak of a decline of MYI concentration, not an increase.

Yes. it should be "grid resolution".

L334: "(so far the only source of ice type information in the Antarctic)". This should be reworded. I assume that in situ data exists to some extent. Also, OSISAF ice type product exists and is processed on an operational basis.

We meant detailed and comprehensive ice type information (this OSISAF ice product has only FYI and MYI) that does not only rely on automatic satellite data processing. we have reworded to *"(so far the only source of detailed ice type information in the Antarctic, apart from ship-based observations of the ice conditions)"*, now L444.

L341-342: "The data become unstable toward the end of the freezing season in September/October, with MYI being underestimated" I think this has not been shown clearly in the result section. And how do you conclude that MYI is being underestimated?

See L315-322 (now L400ff.) in the Discussion section: We retrieve much less MYI than the SoD charts which, however, seem a bit "unstable" or inconsistent. In the discussion, we give a possible physical

explanation for an underestimation (now L404-407). Here, and in the Summary/Conclusion (L453), we have added a "probably" before "underestimated".

L344: I would remove the sentence "outweighs the shortcomings". This only put your product
in a bad light I would say.

Thank you for this encouragement! We have removed these words, but added a "preliminary" earlier in this sentence.

L346-350: Give better and more correct references to upcoming satellites/sensors. And
please add a comment on why 1.4 - 36.5 GHz is assumed to make scatterometer less important

We have revised these lines and added references (end of Summary/Conclusion), and also indicated why using CIMR's 1.4 GHz  channels is useful for MYI retrieval.

**Response to RC3:**

Note that we quote the reviewer's comments and suggestions in red.

51: One reason why most MYI is in the Weddell Sea is that the gyre that transports MYI away from the coast to the north and northwest also transports in ice from the north and northeast. This is seasonal ice that gets transport into the Weddell, where it compacts along the ice shelve and Antarctic Peninsula and, along with less solar insolation and colder temperatures, allows that FYI to survive into MYI. This seems a salient point to make here as it is the mechanism to form MYI.

We have added a statement on that in the text: "In turn, seasonal ice is transported into the Weddell Sea from the north and northeast, can be pressed and compacted against the ice shelves and the coast of the Antarctic peninsula where it survives the summer ad becomes MYI." (now L56ff.)

90-92: No SSMIS sensor data are used?

After 2002, the preferred satellite instruments to be used are AMSR-E and AMSR2 as they have higher resolution than SSM/I and SSMIS. SSM/I was mentioned as it can extend the record backwards before the AMSR-E era. SSMIS can, of course also be used, and actually it can close the gap between AMSR-E (until Oct 2011) and AMSR2 (from July 2012). We have mentioned this here (beginning of section 2.1).

174: How is the "beginning of the cold season" defined? Is it the minimum total extent? But at the minimum, there may be regional gains and regional ice losses occurring (the minimum marks when the gains start to outpace the losses). Ideally, you would use the minimum at given grid cell or at least regionally.

It is not feasible to define the beginning of the cold season grid-cell-wise (using reanalysis data), in particular as the drift correction is not grid-cell-wise but rather a neighborhood operation. This would also cause problems at the region boundaries if the beginning or the cold season were defined region--wise. Since we are here identifying sample areas for FYI and MYI, looking for the beginning of regrowing ice regionally is a reasonable approach. We have included this information now in a footnote.

187: How accurate are the ECMWF 2 m temperatures over the sea ice? There are several coastal stations that I assume provide observations, but over the sea ice, the observations are quite sparse, with few buoys (compared to the Arctic). It is reasonable to use ECMWF as that is what is available and better than nothing. But I think a mention on potential uncertainty is worthwhile here.

Yes, we have mentioned this (now L241ff.).

188-189: And likewise for the ice motions. Antarctic motions typically have higher errors because of the variability of the ice (flooding ice, etc.) and lack of buoy validation. Again, don't need to go into great detail, but a comment on the uncertainty would be helpful.

Here as well, we have mentioned this.

234-235: In what format are the SoD charts provided? It seems they are used here merely qualitatively. If they are just images, that makes sense. But if they are in some sort of data format (e.g., GeoTIFF), they could be used to do some quantitative comparison with the ECICE. And also, as noted below, they could be manipulated to consolidate the different ice classes into the main three with a clear color scale to more easily visually compare with ECICE.

We have used the maps (graphics files, PNG). Analysing this in more detail using the data in original SIGRID3 format is planned but would probably beyond the scope of this paper.

236, Figure 2: This figure seems a bit odd and confusing to me. It seems like there are two SAR images overlaid on the ECICE image. But they overlay, so block the ECICE. Once can see some continuity, so the performance looks reasonable, but it seems odd to show only one figure with one or the other (SAR or ECICE). The ECICE color scale seems to have several more gradations than the 5 indicated in the legend. The legend color scale should match the colors plotted. It seems

like creating a two-panel image – one with the ECICE and one with the SAR images and then overlay the contours on both – would be clearer?

(Note: now Figure 2) We have considered modifying the figure, splitting it into two panels. However, this figure is only to illustrate an episodic check of our first results and we want to keep it compact. Note that the legend gives only the most important color shades (the idea was not to clutter the plot with too much information), we now give a hint about that in the figure caption..

239, Figure 3: I guess it is okay to have the SoD color scale in the Appendix – at least the authors acknowledge that it isn't legible in the figure. But ideally, a better color scale would be included/added to the figure. And it's clear that the SoD figure has more categories than the ECICE, so it is a bit hard to directly compare, though the overall patterns are clear. It would be more work, but if it were possible to actually take the SoD and create a custom plot with the SoD categories combined into the three ECICE categories, that would be quite helpful.
267, Figure 5: As for Figure 3, it would be nice to have SoD in a simplified form with all types consolidated into the three ECICE types and with a color scale legend provided with the figure.

(Note: now Figure 4) As mentioned above, using the original data might be beyond the scope of this paper. Therefore, we have decided to keep the figure as is, in particular as there is already one "color family" for each of the three ice types: pink/purple hues for YI, yellow/green hues for FYI, brown hues for MYI.

**Minor Comments (by line number):**
45: I've seen "snow-ice" with a dash to connect the two nouns and denote a unique type. But this is perhaps simply more of an editorial/style decision.

We use the convention to write a two-word compound without a hyphen, just like "sea ice".

114: Typo, "cost" not "coast"

Yes.

174: Not sure why the ASI reference is given as a footnote? If that is The Cryosphere style guideline, I guess that's okay, but in my view, datasets should generally be cited as regular references.
(text)

We wanted to give the direct URL for the data on our server and also the reference to the PANGAEA data set, and avoid lengthy parentheses.

346: It seems like the chart color legend (Table A1) should be after the beginning of the Appendix text? But as noted, it would be helpful to create a new legend that combines the relevant classes into the three main types for the figures in the main text of the manuscript.

Figure placement has to be straightened in the final version any way...

---

## Author Response (AR2)

**Response to RC1:**

Note: We cite the reviewer's comments in blue and have numbered them

We again thank the reviewer for the critical eye and for the very detailed and constructive criticism.

Some aspects of the manuscript have improved compared to the version I reviewed. I still have a number of concerns, though, that make me to suggest another round of major reviews. The reasons for that I lay out in my general comments, further detailed by my specific comments.

**General Comments:**

**GC1:** The authors stated in their response to my 1st review that this is a proof-of-concept study and used this as an argument to keep a lot of the content of the manuscript at a level of depth where is was before. I therefore ask the authors to reflect the nature of this study in their title, in the abstract and also in the conclusions. See item (1.)

**GC2:** There are certain elements of the methodology that are still not described sufficiently well. To these belong how the initial MYI distribution is found, or how ice drift information is employed. In addition, the intercomparison of TBs and NRCS values from the sampling areas and periods with independent values of these parameters needs to be improved. At least the obvious discrepancies between the values used and those published in the literature should be mentioned properly and taken into account when discussing the limitations of the approach and the potential way ahead. About the initial MYI distribution, see item (16.a) below. How ice drift information is employed is quite explicitly explained in section 2.2. On the discussion of TB and NRCS values from our distributions vs. literature see (GC3) below, as well as items (19.), (20.), (20.a) and (21.).

**GC3:** The interpretation and discussion of the results still requires a more thorough incorporation of the physical properties of sea ice and its snow cover and their influence on the microwave properties. The discussion appears still to be biased to much towards improving the correction scheme rather than improving ECICE and the selection of adequate sampling areas and periods for the adoption of ECICE from Arctic to Antarctic conditions.

ECICE is a mathematical scheme, which does not care about physics, and in this sense it cannot be improved using knowledge of physical properties. What can and has to be improved is the selection of sampling areas – we have addressed that point already in the first revision, but we will elaborate more on that (cf. items (16.a), (16.b), (18.b)). But certain effects. namely temporal changes of the scattering and radiometric properties of the ice types, cannot be taken into account by better sampling, and this is exactly what the correction schemes are meant for. So one cannot say that "ECICE should rather be improved instead of the correction scheme" – both are important. We admit that in the original manuscript, we were biased towards the correction schemes, but we are now giving more room to the sampling problem as already mentioned.

**GC4:** There is a number of other smaller issues where the information provided is either not correct or needs to be complemented or where a more critical discussion of the own results appears to be useful for a more complete understanding of the main message of the manuscript. See "Specific Comments"

**Specific Comments:**

**(0.)** Title and abstract: You have decided to not build your paper on a solid review of our understanding of the different physical properties of the Antarctic sea ice compared to the Arctic sea ice and their implications on active and passive microwave satellite data. Fine. You have also decided to keep out expertise about snow on Antarctic sea ice. In that case, however, I cannot accept this paper without the following suggestions for amendments in the title and the abstract.

(1.) The title is far too general and points into the wrong direction. As you stated in your comments to my 1st review this is a "proof-of-concept" study. This should be mentioned in the title.

We have modified the title, it now reads: "First results of Antarctic sea ice type retrieval from active and

passive microwave remote sensing data". We do not use "proof-of-concept" in the title as we think the paper is more than that: It is a proof of concept and the presentation and discussion of first results.

(2.) You use ECICE and you discriminate young ice from first-year ice from multiyear ice - similar to what ECICE does in the Arctic - without modifying ECICE for the Antarctic for the reasons laid out in your response to my 1st review. Please mention ECICE in the title (there are Arctic ECICE papers, so it does not harm to mention it here). Hence a title such as: About a proof-of-concept study applying the ECICE algorithm to discriminate Antarctic multiyear ice from younger ice types" would be much more to the point.

We do not think is is a good idea to use an acronym like ECICE (which is not generally known, i.e., cannot be found in dictionaries) in the title of a paper.

(3.) In the abstract I suggest you write that ECICE uses probability distributions that decouple its application from the need of a deeper understanding and consideration of the emissive and backscattering microwave property differences between the two polar regions. This would lend much more credibility to the abstract and would fit much better to the content of the paper. Note that in L8-10 you explicitly write "Due to differences in physical and crystalline structural properties of sea ice and snow between the two polar regions, it has become difficult to identify ice types in the Antarctic". Since you do not deal with an exhaustive investigation of the different properties on the AMW and PMW signals but still come up with a solution - which is only possible thanks to the nature of the ECICE algorithm using probability distributions - I feel it is mandatory to tell the reader this piece of information, i.e. that you propose to, as a first proof-of-concept, apply a method based on probability distributions of the involved AMW and PMW data. See also my next comment.

Do we really have to justify that our algorithm (ECICE) "only" takes distributions of the input parameters for the surface type and does not need explicit physical knowledge of sea ice? Most methods for the retrieval of ice concentration do not even use distributions, but merely use *one value* per input channel per surface type, and no further physical details. Using distributions instead of single values does seem a bit closer to the physical reality. – We have inserted a sentence about the nature of ECICE, now L11/12.

**(4.)** The sentence "Until recently, no method ... time scales" in L10/11 should be deleted to my opinion. You did not develop a new method here. You applied a method that is known to work for Arctic conditions to Antarctic conditions to see whether it works.

Well, we meant that no other method has provided data for monitoring the distribution and temporal development of Antarctic ice types, particularly MYI throughout the freezing season and on time scales of several years. We admit that you can argue whether ECICE plus corrections schemes can be viewed as two different methods for the Arctic and Antarctic – if we replace "method" by "retrieval scheme" we can certainly say so. Note also that ECICE/correction schemes for retrieving Arctic MYI was published quite recently (*Ye et al.*, 2016a,b) – depending on how you define "recent". We now say: " [...] no retrieval scheme was ready for monitoring the distribution [...]", now L8/9

**(5.)** I also strongly recommend to re-formulate the sentence "Although there are ... sea ice types" in L16/17. This is not the first time we learn about the evolution and dynamics of Antarctic sea ice types. I would rather say: "Our results of applying ECICE to Antarctic sea ice conditions for the first time demonstrate its potential for Antarctic sea ice type discrimination. They also confirm existing knowledge about typical spatial pattern in the sea ice type distribution and their movement."

Thank you for your consideration.

This is not about the typical patterns and yearly cycles, but about the possible interannual variations. The typical patterns are known, but the year-to-year deviations from the typical pattern are hardly known. We have made it clear by adding the word "interannual" here (now in L16). See also next item.

**(6.)** L39/40: "Unlike the Arctic ... still unclear" --> I object to this statement. It is well known where sea ice typically survives summer melt. It is well known where it usually drifts. It is well known that a lot of the sea ice is either formed along the marginal ice zone in the so-called pancake ice cycle or in the ubiquituous coastal polynyas from where it drifts north or northeastward. Also the location and age of landfast sea ice is well known.

We agree and have rewritten the sentence. It now reads "While the general distribution and yearly cycle of the different sea ice types in the Antarctic are known to some extent, details, the interannual variations and possible long-term trends are still unclear." – now L39-41

**(7.)** L41-42: "In the austral summer ... 2012a)." --> What do studies by Hobbs et al. (2015) and Parkinson and diGirolamo (2021 and before) state in this regard?

They also mention an overall slight increase (until 2014), but with regional differences. Both Hobbs et al. (2015) and Parkinson and di Girolamo (2021) mention an overall slight increase (until 2014), the latter also shows the clear increasing trend in the austral summer months (Figure 2 (a) and (b) of that paper). However, we do not think that we need to include all this in the manuscript here as we just want to make the point that an observed slight increase of Antarctic sea ice extent might imply an increase in MYI extent, at least until 2014. We now additionally refer to Parkinson and di Girolamo (2021), now L43

**(8.)** L53/54: "For these reasons ... two regions" --> I don't agree to put this sentence here because you have not yet made any link between the physical properties that you just described and how these influence radiometric and backscattering observations.

We think it can be taken for granted that the structure of sea ice has an influence on its radiometric and scattering properties, even without going into detail here. We have replaced "are different" by "are expected to be different" (now L57).

**(9.)** L62: Please provide at least one reference each for these two important implications of landfast sea ice in the Antarctic.

It was the reviewer who gave us the information about landfast ice here (without references) and requested to mention it, which we had agreed to. We have added a recent reference on that (Fraser et al., 2021), now L65.

(10.a.) L76/77: "exclusively to the Arctic" --> Why is this the case? Because the radiometric signature between MYI and seasonal ice in the Antarctic was found to be not well suited to carry out a similar analysis in the Antarctic - at least when using the same channels as in the Arctic.

We have the impression the reviewer would like us to insert this explanation into the text, which we readily do as it adds clarity (now L79-81)

(10.b.)- "Besides, the retrieved ... see Section 3.2" --> This is a too global statement. First of all it needs to be stated whether you refer to MYI extent or area. It is the area which cannot increase during winter while the extent may increase during periods of MYI fracturing and dispersion during divergent conditions - depending on the threshold set to compute extent and depending on the capabilities of whatever algorithm used to reliably discriminate MYI from other ice types. Secondly, this tendency of MYI whatever to increase during the cold season is something that may occur but that not occurs regularly (I know this from own experience working with the NASA-Team MYI concentration data). Hence, I suggest to rephrase your statement.

We have specified that we mean the MYI *area* (now L81). Our wording "tends to increase" actually means that it often, but not always, increases. We do not say "has an increasing tendency" which would mean it always increases which is not the case as the reviewer pointed out.

(11.) L124/125: "the radiometric ... or even vanish" --> Since you are referring to Arctic conditions here it would be suitable to cite respective papers dealing with sea ice type retrieval in the Arctic, for instance the two Lindell and Long papers from 2016 in Transactions of Geoscience and Remote Sensing, 54, or Remote Sensing, 8. We now cite one of the papers (the one that includes ASCAT data) – now L132.

(12.) L175/176: "but is FYI ... of the ice" --> I suggest to add that this can also be FYI with a thick snow cover known to resemble radiometric properties similar to MYI ice (as is one of the problems using the 37 GHz / 19 GHz vertical polarization gradient ratio for snow thickness retrieval).

We have added "or because of a thick snow layer" - now L184/185

(13.) L179: "non-MYI" --> You write that you keep this new pseudo-ice type which is either FYI or YI - so simply seasonal sea ice. But I cannot see it in any of the maps you show later. To which of the concentrations is the concentration of this non-MYI added? If it is not added anywhere ... where does it remain? Or is this the "magic" missing 20% in some of the maps of areas with 100% sea ice concentration?

We admit one could do more with the non-MYI. If we just had two ice types, MYI and seasonal ice, we could simply add the non-MYI to the seasonal ice. However, as stated in the manuscript, (in the same line), we cannot tell how of it is FYI and how much is YI, so we just keep it and do not use it. One approach , e.g., would be to split the ex-MYI into FYI and YI according to the FYI and YI retrieved in that grid cell. We have inserted part of this discussion in a footnote that refers to what is now L188.

Speaking of "magic missing 20%" is imprecise and sounds even slightly polemic- it would have been good to stick to the facts instead: the total ice concentration from ECICE is between 80% and 100% in some areas

where the SoD charts show 10 tenths of ice cover, in particular in areas of YI and FYI in the inner Weddell Sea (cf. item (31.)).

(14.) L204/205: "This is the common grid used ..." --> I don't think that the information who uses this grid is required. Widely used as well is the EASE grid which has several advantages over the polarstereographic grid and is used, e.g., by OSI SAF for its sea ice concentration products. What is required, however, is to mention the latitude of the tangential plane of that polarstereographic grid.

We have mentioned the NSIDC just because the grid resulting from this specific polar stereographic projection with corners at at 39.23°S /42.24°W, 39.23°S /42.24°E, 41.45°S/135°E, and 41.45°S/135°W, standard longitude of 0° and standard latitude (latitude of the tangential plane) of 70°S, is very often just called "NSIDC grid", and it is still widely used, even though other, better grid projections (EASE grid) are increasingly being used. We did not want to put all these details here as they are not essential for understanding the paper and would make it clumsy, but instead refer to our data user guide. Note that the effect of different standard latitudes is just a different scaling of the whole grid, so the standard latitude for the "NSIDC grid" it is not needed at all to understand the maps. The standard latitude is essential, of course, to correctly interpret the meaning of "nominal grid resolution". So when a reader wants to actually use the data (e.g., our data), the accompanying documentation gives all the details needed.

(15.) L210/211: "using a simple linear approach ..." --> Did you develop this by yourself? In that case it would be appreciated to learn more about the method and its application frequency (daily? monthly? seasonally? How often are the linear regression coefficients updated? What is the data set used to define where there is sea ice and where there is open water? Is a sea ice concentration threshold used? What about summer conditions?). In case not please provide an appropriate reference.

This approach was developed in the group of C. Melsheimer and G. Spreen, based on the observation that the dependence of the NRCS (in dB) of sea ice on the incidence angle for ASCAT can be well fitted by a straight line (also observed for ERS Scatterometer by Gohin &Cavanié, 1994, DOI:10.1080/01431169408954156). The data were taken over regions with near 100% ice cover in January 2016 in the Arctic, and the slope *p* determined from the data is used to convert NRSC *s*(*t*) from incidence angle *t* to 40° according to:  $s(40^\circ)=s(t) + p^*(40-t)$ . While this is a rather simple method, it works well: daily composites of ASCAT NRCS data thus converted have not shown any visible swaths. We have included this information in the new Appendix C.

**(16.)** L214-217: "For FYI and MYI ... per day" --> These lines and your comments to my 1st review do not lay out sufficiently well what you did why. Please put yourself into the position of a student who wants to redo your analysis ... The following points need clarification:

**(16.a)** Why did you choose 2018 as the "master" year for your initial discrimination of sea ice types to find the probability distributions of the parameters laid out in Table 1?**

We started adaptation of the algorithm to the Antarctic in 2017/2018, so we chose rather recent data and ended up with 2018 (now mentioned in L231). We admit this is somewhat arbitrary. We have also improved the explanation why we have chosen this specific approach for FYI and MYI (now L227ff.).

(16.b) While you attempt to describe better in your footnote 2 how you define what MYI is, it is not sufficient. First of all, the moment when the sea ice cover starts to grow again, resulting in an increase of the sea ice extent depends pretty much on the drift conditions. I can envision - and this is also an issue in publications where melt and freeze onset are derived - that on a day-to-day basis you cannot adequately define when sea ice indeed begins to advance again. Those publications / methods use - for good reason - a period of 3 or even 5 consecutive days of the respective geophysical process to happen (in your case sea-ice advance) to define that the conditions have switched from summer melt to freeze-up. Secondly, you do that "regionally" ... okay ... What are the regions? How did you define them? How large is the difference between freeze-up (or ice-advance and hence MYI definition) days between different regions? I can envision that MYI is defined as such as early as mid / end of February in the Weddell Sea but as late as May in parts of the Bellingshausen Sea. Here, your manuscript lacks essential information.

Note that the approach to look for the sea ice to start grow again is just used to identify MYI sampling areas. This is not an analysis about where in the whole Antarctic the cold season starts on which specific day. We are aware of the complexity of the topic. Here we only want to deal with identifying a handful of regions (see Table A2) of ice that is left over from the previous season and make sure that refreeze has started. By observing daily maps over a few weeks from February to April it is very well possible to distinguish just drift

from real ice growth because only in the first case the concentration would go down if the extent increases. We have moved the text from the footnote into the text and elaborated more on this: "For this purpose, we have observed the daily ASI sea ice concentration maps in February and March [...] before possible drift blurs the picture, the areas of MYI can be identified." – now L232-237. We admit that there might have been a more sophisticated way and will address that, along with the general weaknesses of identification of sample areas, later in the discussion (end of section 3.2, L436-456).

We also have to address another possible misunderstanding: We do not define our "initial" MYI as all remaining ice at a certain day, or regionally at certain days. We always take the MYI that is output by ECICE, which is now clearly stated towards the end of section 2.3, L290ff.

However, the correction for snow metamorphosis, deformation, snow accumulation(i.e., processes that make non-MYI look like MYI), called drift correction, needs a reference MYI domain from the previous day. On the day which we use as starting day and which should be during the beginning of freeze-up, we use the (uncorrected) MYI from ECICE from the previous day for that. After that, we always use the corrected MYI form the previous day, of course. Note also that the mentioned processes that make FYI look like MYI permanently can be expected to be still weak in the first few weeks of the freezing season, therefore, the corrected MYI concentration) should not depend strongly on the beginning date for the drift correction. We have included this information in the manuscript, in section 2.2.2 "Drift correction" (now L187ff).

(16.c) "Later in the season ..." --> I probably understand what you did but it is not described properly. When sea ice is defined as FYI becomes not clear. Do you apply a sea-ice motion data set to actually track the border of the MYI defined in the previous step? Or did you apply a ball-park estimate of 10 km drift per day? In the latter case the question would be: i) into which direction and ii) every day?

This is an estimate of the drift, just to get a approximate upper limit., no sophisticated tracking involved. We have added a word to stress that this is just approximate (now L240).

**(17.)** L223/224: Did you check the weather conditions in the regions and during the time periods chosen to take your open water samples? Are these from predominantly quiet conditions (low wind speed, low atmospheric water vapor load, low cloud cover) or do these cover a wide (an as wide as possible) range of different conditions and hence atmospheric influences on the AMW and PMW observations? It would be useful to provide this information in the context of showing and describing the respective distribution functions.

As can be seen from the size of the OW sampling areas (about (600 km)2 and (250 km)2, respectively) as well as the time period (5 days, 4 days), they will definitely include widely diverse atmospheric and oceanic conditions. We have added a sentence stating this, now L247-249.

(18.) L224/225: "Details of all ..." --> Thank you for providing this information at least in form of a table. For the understanding and the credibility of your proof-of-concept study it would be much more useful, though, to have a map with the sea-ice concentration of, say, the first week of March 2018 (noting that several of your samples are dated around this period) superposed with the lat/lon boxes from where you obtained the data for the FYI and MYI samples. You could use such a map also to illustrate the regions that you apparently used to define freeze-up day and hence the date from which onwards a grid cell is defined MYI.

We do not want to overload the paper with illustrations and think it is credible enough to describe how we selected these areas, on which we have elaborated now, as stated in our detailed response to item (16.2) and the corresponding extra information in the manuscript. Note that we have not defined (and need not define) a regional "freeze-up day", see item (16.2) as well.

(18.a) - I note that YI samples are chosen across the entire freezing season and also FYI samples originate from three different time periods, increasing the likelihood to cover the potential range of surface conditions. Only for MYI you restricted the samples to March which I believe is sub-optimal. Particularly in the Weddell Sea Sentinel-1 SAR imagery allows to clearly separate MYI from FYI also later in the season (just take a look at the SAR images provided regularly by the Polarview consortium). This is a lack of rigorosity in the selection process of the MYI samples. I note in addition that you have substantially fewer MYI samples (about 1000) than other samples (FYI about 6000, OW about 14 000, YI about 2000). Given the fact that in the interpretation of your results you focus quite a bit on the MYI cover distribution this seems to be a sub-optimal and not sufficiently well-thought through way to generate a data set of samples to be used. MYI signatures can vary a lot during winter and particularly in March they might not be representative for conditions met later at all

Admittedly this is not ideal, but there is a reason for the seeming "lack of rigorosity": The problem is to find reliable MYI without using other data sources (which was our guideline). We see now that external data to identify MYI areas throughout winter might really improve the retrieval. However, it is not trivial to identify large enough areas of, if possible, pure MYI (100% MYI concentration). The sample area selection is now described in more detail (see items (16.a) and (16.b)), and further discussed towards the end of section 3.2, L436ff.

**(19.)** L233/234: I am glad that you put your observations into the context of values published in the literature. I am wondering however, whether you are fine with the fact that for FYI the GR3719 is well below zero ... which is indicative of what? Figuring this out will aid you in your interpretation / discussion of your results which is not yet convincing.

For FYI,  $GR_{37V19V}$  is close to zero for bare ice surface and becomes progressively negative as the snow depth increases. In other words,  $GR_{37V19V}$  is a function of snow depth on FYI, and used as such in a few algorithms to estimate snow depth. That is why the use of the range of this parameter for FYI in ECICE is useful (as opposed to using a single value). The use of the range of  $GR_{37V19V}$  does not help in interpretation of the results but it helps in producing more accurate classification of ice types.

**(20.)** L234-238: "The MYI tie points ... distributions." --> Your "well above" is something like 5 - 8 K which is not that impressive. Your observation that the modes of your MYI TB distributions do not match well with MYI tie points in the published literature could very likely be caused by the fact that you selected cases from March - a time of the freezing season where the MYI signature might not be representative of winter conditions. You could take this observation as a hint that your cases to derive MYI TB and sigma\_0 samples are not well chosen.

We have changed the text to "about 5 to 10 K above" (now L260). We now mention that our samples are from the beginning of the freezing season and might thus not represent the whole range (L262-264).

(20.a)- The comment you make in the last sentence does not really help here since Arctic MYI differs substantially in its physical and hence microwave structure from Antarctic MYI ice - simply because the melting process differs fundamentally. I therefore suggest to delete that last sentence. We agree. Deleted.

**(21.)** L238/239: You state that your C-Band NRCS values for Antarctic MYI are higher compared to what is observed in the Arctic. Did you check with results from the Antarctic? Haas (Annals of Glaciology, 33, 2001) reported C-Band NRCS values of Antarctic perennial sea ice of -16.3 dB compared to about -10.7 dB during summer. Gohin (International J. Remote Sensing, 16(11), 1995) reported winter-time NRCS values of MYI of -14 dB. Finally, looking into Arndt and Haas (The Cryosphere, 13(7), 2019) I find C-Band NRCS value time series for FYI and MYI. In short: There is a lot of literature you could have taken a look into to illustrate how well (or not well) your MYI NRCS values agree with published literature in the Antarctic; there is no need to look into the Arctic

We have removed comparison with Arctic data and replaces this by comparison with results by Arndt & Haas (2019), and we also point out that according to the time series in that paper (Fig. 4a) the highest MYI backscatter values occur in autumn and early winter which is the time from which our MYI samples stem.

**(22.)** L242: "the only source ..." --> How about NCEP/NCAR, MERRA-2, or JRA-55 re-analysis data? Are these not comprehensive and consistent as well?**

We agree, the wording was not good. We of course mean that meteorological reanalysis data are the only source of comprehensive and consistent temperature data over the Antarctic sea ice, and we have modified the text accordingly, now L268-270.

(23.) L248: "Sea ice drift data ..." --> why is that? Please provide a reference which backs up this notion.

Lavergne et al. (2021, DOI:10.5194/tc-15-3681-2021) found larger standard deviation in the Southern hemisphere when comparing ice drift from PMW with buoy data than they found for the Northern hemisphere (Table 2 and Fig. 5 of that paper). Possible causes: (1) larger drift speed in the Antarctic, so the motion tracking algorithms have to cover more area before they find the maximum cross-correlation which leaves more room for ambiguities in the motion field; (2) the processes that modify the ice in the Antarctic are often stronger than in the Arctic, hence the radiometric signature of the ice is more unstable which also hinders the tracking algorithms; (3) Antarctic sea ice is in lower latitudes than Arctic sea ice, so there is less overlap of the satellite swaths and, hence, less abundant coverage. Added the reference and a short remark (L275-276).

**(24.)** Figure 2: The color in the legend does not match with the description given in the caption and in the text. This has to be corrected. I guess you switched YI and MYI.

Yes, indeed! Thanks for finding this error which sneaked in when modifying the figure for the first revision. – Corrected.

**(25.)** - Please refer to appendix A and the relevant tables of the regions used to define these distributions and also provide the total sample numbers of the four different surface types shown.

We now refer to appendix A in the caption of Figure 2 as well, and also specify the total number of samples for all four surface types.

**(26.)** L285/286: "and the ice is more porous because ... scattering." --> Please cross-check this statement with the paper by Haas and Arndt and Haas I mentioned further up in this 2nd review of your paper.

We just want to give two possible reasons for the fact that in SAR images, MYI looks brighter than FYI, i.e., the NRCS of MYI is higher than that of FYI – we have added "e.g.," to make this clear (now L316).

**(27.)** L286: "we manually scaled ... classes." --> just a comment, no action required: This is another element of your work / paper which is not sufficiently transparent such that an interested student can redo the analysis the same way as you did it.

(28.) L287: "19 Sentinel-1 scenes" --> "Sentinel-1 SAR scenes" Corrected.

**(28.a)-** You must provide the source of this data. Added data source (https://scihub.copernicus.eu).

**(28.b)**- You write "in the Weddell Sea and near the Antarctic Peninsula". This applies to me that you used SAR images covering the Weddell Sea but also parts of the Bellingshausen Sea? ... As stated later on, a map would solve this lack of information (or ambiguity of the information provided).

We have added overview maps for the two days and included them in Figure 3.

(29.) Figure 3, L296/297: "Qualitatively, the ... is similarly good." --> What keeps you from showing a map into which you plot all 19 SAR images and superpose the ice type isolines as you did in Figure 3? One could get the impression that you picked the best example here and that in fact the agreement is less (even qualitatively) good for the remaining 18 images - simply because these might not focus that well on the wonderful MYI-FYI boundary shown in Figure 3. I know, it is just a proof-of-concept study and for this I do not expect a quantitative analysis of the SAR images used anymore and I do not expect something like a quantification of areas of misclassification (or good classification). But what I can expect is that you provide the reader with a sufficient amount of information that allows the reader to believe in the credibility of your results. In your case this means: show all SAR images (If you are in doubt of creating too many figures then --> merge Figs. 13 to 15 into one figure).

See above, we have added overview maps for the SAR scenes now.

**(30.)** L305-307: However, no ..." --> Just a comment: This makes a lot of sense because there are places around the Antarctic coastline where sea ice advance kicks in as late as July.

**(31.)** Figure 6 / L322-324: I commented on that in my 1st review and do not see it solved here: The color scale used in Figure 6 implies that the total sea ice concentration for the majority of the areas downstream of the Filchner-Roenne Ice Shelf is less than 100%; the color suggests something between 80% and 90% with higher FYI concentrations occurring actually rather at the boundaries of the FYI areas than within the areas themselves. Therefore, please tell the reader that the partial concentrations shown in Figure 6 do not need to sum up to 100% (even though I don't understand why.). Otherwise the reader sees that there are areas where MYI concentration = 0%, YI concentration = 0% and FYI concentration = 80% in regions where the SoD maps clearly indicate 100% and would at least have doubts about your product. It is pretty clear that in these areas OW concentration cannot be 20% (and if so, then your algorithm is not producing correct results here).

From the first review we had the impression the main problem was that we had failed to explain clearly enough that the partial ice concentrations of YI, FYI and MYI do *not* add up to 100% but only to the total ice concentration which can of course be less than 100%. We have made some effort to explain this clearly in the revised manuscript (several places) and also in our response to the review (notably, in our response to GC4, and in items 26.6, 28.4 and 30.4. of the first revision). Upon rereading the first review, we notice that another concern of the reviewer was that the sum of the partial concentrations seems to be 80% to 90%, while the SoD charts show 10 tenths of ice cover (they only indicate tenths, not per cent, which makes a difference as

to the precision meant). However, noticing that the AARI and NIC charts of the same day, 30 Mar 2017, shown in Figure 5, show 8 to10 tenths and 10 tenths. respectively, for the area in question, we did not think 90% total ice concentration to be a problem. Indeed, the total ice from ECICE shows indeed 80% to 95% (ASI maps show about 90%). We briefly discuss this issue now, L374-379.

**(32.)** L330/331: "The FYI areas ..." to 70%" --> I am not agreeing completely here because particularly in those regions where we have these strips FYI concentrations are often as high as 100%.

Inspecting the figure again, we do not see 100% FYI concentration where there are strips of about 20 % to 40% MYI concentration. However, the MYI concentration is sometimes as low at 20% in the MYI strips of the SoD charts and the strips are of course not fully congruent. This is not surprising as the SoD charts are weekly charts, the retrieved ice type maps are daily averages. We have modified the text in this sense, now L361-364.

(33.) L354: "no areas of more than 40% YI concentration ..." --> Since you refer to YI concentrations in the northeastern part of the region shown in the previous example I suggest to detail this statement a bit better here. We agree, so we have added:", with the exception of some narrow areas on the coast/shelf in the Northeastern part (top right) of the box" (now L395)

**(34.)** L359-361: Is this rather global statement at the beginning of this short paragraph relevant and correct? I would say, how large icebergs are classified depends on whether their radiometric signature resembles that of MYI or any of the other ice types and this signature changes with the season. Your results do not support that icebergs are generally classified as MYI (compare Fig. 3 and 4 as well as 6).

We agree and have changed the text accordingly (now L398-399)

**(35.)** L364: "increase around July" --> In view of Fig. 11 this is clearly an underestimation of the situation. If one would compute a running 30-day mean of the time series shown in Figure 11 one would get an upswing of the MYI area that begins in May/June and ends in August before again an increase in MYI area kicks in during October. I recommend to state the situation in a light that is more in line with what the figures shows.

We write "the total MYI area shows [...]in most years even an increase around July" – this is, of course, not a description of Fig. 11 (year 2018) but a succinct summary statement for all investigated years. We have now added a few words mentioning that in the 2018, shown in Fig. 11, the increase is earlier (now L405/406). Apart from that, we think it is clear that Fig. 11 is meant to underline our observation of large fluctuations as well as an increase around July in most years, and do not want or have to discuss all details of the Figure.

**(35.a)-** Did you check the minimum sea ice area at the end of summer in Feb/March 2018? Is your initial MYI area in line with the minimum sea ice area?

According to NSIDC, the average Antarctic sea ice area for February 2018 was about 1.6 Million km2, the ASI data record shows a mininum of about 1.5 Million km2, which is higher than the start of the MYI curve in Fig. 14. However, as mentioned already, we use the MYI extent retrieved with ECICE, and in late February, not all ice is retrieved as MYI: there are coastal polynyas that that are not MYI, and some ice in the southeastern Weddell Sea is (still?) retrieved as FYI (also visible on the NIC SoD chart: http://ice.aari.aq/antice/2018/02/20180222\_nic/nic\_antice\_20180222\_sd.png).

**(35.b)**- Before you begin with explanations what might cause this increase during winter I recommend to first briefly explain how you expect this curve should look. I am wondering in this context whether the intial decrease of the MYI area during March and April is not far too large and/or rapid.

We have stated one line above that we expect the curve not to rise. We think we cannot be more specific.

**(35.c)**- Finally, a piece of information that is (still) clearly missing is how accurate and credible your initial definition of MYI is. You did not come up with this and your footnote 2 a few pages up is not sufficient to understand how you regionally defined the starting point for your MYI occurrence. From what you wrote I can only speculate that the dates from which onwards ice present in a particular region is classified as MYI differs between the regions and that this could have had an impact on how Fig. 11 looks.

As stated above (item (16.b)), we do not define all remaining sea ice at a certain day (or in different regions on different days) as MYI. We always use the output MYI of ECICE. The start of the freezing season is needed in order to have a starting point of the correction for FYI and YI that looks like MYI but cannot have drifted there (drift correction). As discussed above, the result is not strongly dependent on the exact day.

**(36.)** L377/378: "Usually, snow ..." --> Please be more specific about the processes that can occur during winter on sea ice relatively close to the ice edge. It is deformation, increasing roughness. It is snow thickness increase due to snow accumulation, resulting in a decrease of the GR3719 towards values that resemble MYI (but I note that your FYI samples exhibit a quite negative GR3719 anyways ... unlike in the Arctic, see your Figure 2). It is snow metamorphism due to air-mass change induced melt-refreeze cycles. It is flooding of the ice-snow interface with formation of slush at the basal snow layer which eventually refreezes to become snow-ice - an ice type widespread in the Antarctic which radiometric properties not well studied.

We think that our wording "Usually, snow backscatter increases with time and emissivity decreases, making it resemble MYI in that respect" is an appropriate summary statement that gives a reason for spurious MYI and hints at possible problems with the distributions.

**(37.)** Figure 12: Also for these maps you must detail why the sum of the ice type concentrations does not need to sum up to 100% but can have any arbitrary value between - say 80% and 110 or even 120% - for areas with 100% sea ice concentration.

No, the sum of the ice type concentration cannot have "any arbitrary value between [...] 80% and [...] 120%". The reviewer seems to have misunderstood our explanations in L137 and L334 of revision 1 of the manuscript and our responses to several items in the first review (see also item 31. above) that the sum of the partial ice concentrations of YI, FYI and MYI *plus the open water fraction* is 100%. Thus, a MYI concentration of, say, 50%, does not mean that of 50% of the ice in that grid cell is MYI, but that a fraction of 50% of *the total area of that grid* cell is MYI. And, of course, the sum the YI, FYI and MYI concentrations can have values between 0% and 100% as it is just the total ice concentration.

Note that only in case the corrected MYI concentration is higher than the uncorrected one (which can only be caused by the temperature correction), the sum of YI, FYI and corrected MYI concentration might in principle be above 100%, but only if the total ice concentration is close enough to 100% that the increment by the temperature correction makes the sum go over 100%. Also note that the temperature correction is episodically and local and usually restricted to autumn.

**(38.)** L379/380: What about the emissivity of pancake ice? Backscatter is just one out of the four input parameters. In the distributions of the other input parameters, there is overlap between MYI and the other surface types, but the MYI backscatter (NRCS) distribution stands apart from those of all other surface types. Therefore, it is increased *backscatter* by pancake ice that is most likely to cause confusion. This is not meant as an exact analysis of the scattering and emission behaviour of ice types and subtypes, but about assumptions what could be insufficient in our sampling area selection (see response to GC3 above).

(38.a)- Also, I note from your sample maps of the YI distribution in Fig. 4 but also here in Fig. 12, that there is a fringe of elevated YI concentrations along the ice egde being indicative of the presence of pancake ice. It won't be any of the typical sheet ice types such as nilas or grey ice - these ice types are only interspersed in pocket like structures in the otherwise dominant pancake ice cover (see e.g. Ozsy-Cicek et al. 2011, Annals of Glaciol). Hence I would guess that your approach deals reasonably well with pancake ice.

Yes, it might be that our approach deals reasonably well with pancake ice, but this would come as a surprise: Unless there was a substantial portion of pancake ice in the coastal polynyas we sampled for the YI parameter distributions, these distributions do not contain pancake ice. Therefore it is appropriate to keep pancake ice as a concern here.

**(39.)** L393: The third reason can simply be that your correction approaches do not work the same way as they work in the Arctic because the processes that change sea ice and snow physical and hence microwave properties are more vigorous and different from those in the Arctic. Also their impacts could either be lasting longer or, what is possibly more probably, their frequency is too high to capture them properly with your correction method. Why should more frequent events not be captured with the correction schemes? We have added a statement that there might be changes to the sea ice in the Antarctic that cannot be corrected by the two correction schemes (L434/435).

(40.) L397/398: "The fact that ... polynyas" --> see my comment above about that your YI concentration maps seem to resemble the pancake ice areas well. See item (38.a)

**(41.)** L400-407: I doubt that in the weekly SoD charts of September and October there is an oscillation of areas classified as FYI or MYI in the Antarctic. If so, then you should perhaps consider not to use the SoD charts at all. I have doubts about this oscillation because these SoD charts are always analysed taking into account the distribution of SoD from the previous week, including knowledge about the past ice conditions and their persistence. In a way, your statement also misaligns with your earlier notion that SoD charts are developed by well-trained, well-experiences analysts.

We have not expressed clearly enough what me meant: In general, the weekly ice charts are produced alternately by NIC and AARI, and in addition occasionally they independently produce maps for the same week. There seems to be a slight bias between them as to distinguishing WMO types 2.5 and 2.6. This hints at how similar these ice types appear even to a trained human analyst. We would not, however, as suggested by the reviewer, draw the conclusion not to use the SoD charts at all, in particular as there are virtually no other sources of ice type information for the whole Antarctic. We have tried to improve the text, now L450-455.

(41.a)- Both, September and October are months of freezing conditions. Melt conditions occur, if at all, still episodic and in association with warm air advection ahead of low pressure systems - at it does all year / winter. The same applies to some extent to November. In my eyes, the most likely reason for the misclassification is that your MYI cases used to train ECICE are from March only and very likely do not reflect the deeper, more layered snow pack conditions encountered in September / October - the months where also the NASA-Team algorithm has severe difficulties to correctly retrieve the sea ice concentration of Antarctic sea ice - because of the layering of the snow (see Comiso et al., 1997 in Remote Sensing of Environment, a paper that initiated the development of the NT2 algorithm to get rid of the substantial biases). In addition to the samples used for MYI you might also consider that the FYI samples might not represent the deeper snow pack conditions often encountered in late winter / spring - aka September / October. Yes, the restricted sampling of MYI at the start of the season is a main concern and is being addressed now – (see items (16.b) and (18.a).)

**(41.b)-** The strong decrease in MYI area shown in Fig. 11 during September / October could also simply be the rebound to "normal" conditions with all the artificial MYI area discussed in the context of Fig. 12 vanishing. What puzzles me more is the increase in MYI towards spring, i.e. October / November.

We agree that the part of the decline is the rebound when the erroneous MYI vanishes (but why does it vanish?). However, the strong decline continues into October, and the SoD charts do not show a substantial decay of the remaining MYI in that time, which is why we assume misclassification.

According to the OSISAF team (see the ATBD and the validation report of their sea ice type product, Aaboe et al., 2021a and b, reference list of the manuscript), the variability of their retrieved MYI in the Antarctic in September and October is already so large that they classify the September and October data as unreliable. Note that they define "variability" as the standard deviation of the different of daily extent minus an 11-day moving average.

We doubt that the total MYI extent in November is meaningful. In the Arctic, the MYI retrieval becomes unreliable in late May, so it is quite likely that this happens in the Antarctic already in early November, considering that much of the Antarctic sea ice is at lower latitudes than its Arctic counterpart.

(41.c)- In conclusion, I strongly recommend to revisit your strategy to choose sampling location and their time periods, take a look into publications that describe what we know about the physical processes of Antarctic sea ice and snow during the course of winter and how these influence AMW and PMW properties of the sea ice; I recommend to again read Voss et al., 2003; Arndt et al., 2016; Haas et al 2001; Arndt and Haas, 2019; Willmes et al., 2014, Willmes et al. 2011, and others and based on what you learn from studying these publications critically discuss what might be wrong in ECICE as used in your manuscript. As you stated 1-2 paragraphs further up in the context of Fig. 12: It is not the correction which partly jeopardize your results but the complex nature of the Antarctic sea ice and its snow cover that is not adequately taken into account into "tuning" ECICE for Antarctic conditions by simply using local TB and NRCS distributions.

As already mentioned (items (16.a), (16.b), (18.a)), we have put more emphasis now on the selection of sample areas. Note also (see our response to GC3 above) that this is not about finding "what is wrong in ECICE". The problems are the samples for the needed distributions, and possible problems with the correction schemes. in particular the drift correction.

**(42.)** L418: "A first check is ... total sea ice" --> Certainly this is a good idea - however, the examples you present in your manuscript do not credibly support such an approach since - at least from visiual inspection of the maps shown - even for 100% sea ice cover the ice type fractions do not add up to a total sea ice concentration of 100%. This needs to be clarified better / corrected before you can consider to carry out the step described in this line.

Firstly, with "100% total ice concentration", does the reviewer mean the 10 tenths of total ice concentration on the SoD charts? As mentioned above, there is some uncertainty in the ice cover information from the SoD charts, so this should not be overrated.

Secondly, note that when comparing time series of some geophysical variable such as sea ice area or concentration from different data sources/algorithms, systematic differences are almost inevitable (think of the ASI sea ice extent time series versus the NSIDC one), but the main thing is the anomaly, or, in other words, the time evolution and the ranking of maxima and minima (see, e.g., Nature 478(7368):188) usually contains the main message.

**(43.)** L421/422: I don't understand what you mean by "as the standard one" and it certainly does not surprize you that I suggest to add to the plot at least one time series of a sea-ice concentration product based sea ice extent, e.g. OSISAF (not the one from Tonboe et al., 2016 but the newer one from Lavergne et al., 2019, i.e. OSI-450 / OSI-430-b). We have introduced our meaning of "standard" two sentences before. Changed to "as the ``standard" ones (here: ASI)" – now L472.

**(44.)** L425: You sum up these areas using all non-zero concentrations, am I correct? What I mean by this is that you do not use a 15% threshold.

Yes, of course, as we calculate the area here. We have inserted "0% to 100%" in the parenthesis for extra clarification (now L478).

(45.) L429: "mainly caused by ..." --> I suggest to re-formulate this sentence because it is very evident in your manuscript that this correction is most likely not the main reason but the fact that your choice of sampling areas and periods for MYI but potentially also for FYI is not representative enough for the full range of conditions encountered during the course of the winter. I suggest not to blame the correction at first place. Yes, we have reformulated the sentence, now L481-483.

**(46.)** L431/432 / Fig. 15 (and also Fig. 14): What is the minimum sea-ice area of the respective years? As mentioned in item (42.), we consider the ranking of the minima and maxima more important than the absolute values.

- A "." is missing at the end of the sentence. Corrected.

(47.) L452/453: "The data become unstable ... underestimated." --> Why is that? Shouldn't the MYI area in September / October be smaller than the MYI area you begin with in March? I'd say that a substantial amount of the MYI found in the Weddell Sea end of summer has melted by then in the Southern Ocean facing the Atlantic and should continue to melt in November. Hence the low September / October MYI area does not seem unrealistic to me but the increase in MYI in November does.

See item (41.b)

**(48.)** L459-462: May I propose a slightly corrected version of this? For (1) I clearly see the need to incorporate also again MYI into the revision of the distributions. Step (2) should be a thorough evaluation of these TB and NRCS distributions with values available in the literature. Step (3) should be a quantitative evaluation of the ECICE results before any correction with a considerably enhanced and better analyzed set of evaluation data which may or may not contain in-situ and ship data (I would say the latter are most useful for the identification of pancake ice locations). Then, step (4) could be an application of the corrections with more up-to-date data. Step (5) would be a final evaluation. Only step (6) can then be an extension of the current data product using AMSR-E and respective AMW sensors plus - again - an evaluation of the extended product.

We thank the reviewer for the suggestions and have included MYI into the revision of the distribution , and slightly modified the other points as well (now L512-517)

**Typos / Editoral comments:**

(if not otherwise noted, the correction was done as suggested)

L43/44: "In the absense of ... still unknown" --> Are you referring to different surface topography and hence impulse fluxes or to different thicknesses of sea ice and its snow cover and hence heat fluxes? We wrote "energy fluxes" and had mainly the heat flux in mind. Changed to "heat flux" (now L45/46)

L47: "and thus" --> "and the ice-snow interface thus" And further: "... flooded. The slush layer resulting from flooding of the basal snow layer with sea water eventually refreezes, becoming snow ice."

L50/51: "to higher wind that triggers" --> "to more variable wind conditions that cause"

L98-103: Please stress more that the following two sections, 2.1 and 2.2, are valid for the ECICE as developed in the Arctic - including the two correction schemes.

We have inserted the note "originally developed for the Arctic" in L108/107 and L108. The ensuing description of ECICE (section 2.1) and the correction schemes (section 2.2.), however, is a description of the general algorithm and not just valid for the Arctic. They become region-specific as soon as ECICE is given the distribution functions and the correction schemes get their tuning parameters.

L302: "and reconnaissance" --> is this air reconnaissance? Then please add this important detail.

We gave checked the documentation of NIC on how the charts are produced

(https://usicecenter.gov/Resources/AnalystProcedures) and actually found no mention of reconnaissance data. However, we found no information for the procedure at AARI (neither on the English nor the Russian web pages), so we delete the mentioning of "reconnaissance", but insert the above link to the NIC procedure

L312: "plus correction schemes" can be deleted as you write "corrected MYI concentration"

When we delete this, the statement would read "corrected MYI concentration from ECICE" which is wrong as ECICE does not output corrected MYI concentration. Therefore, the "plus correction scheme" cannot be deleted.

L332: "one sea ice type class" --> add "such as in the SoD charts"

L333: "retrieval where" --> "retrieval such as ECICE where"

L358: "given the limitation ... yet." --> I suggest to delete this part of the sentence; it does not add relevant information.

L364: "form" --> "from" --

Figure 13: You can remove the title. Both, the y-axis title as well as the legend state what is shown. A "." is missing at the end of the caption.

Figure 14: As in Fig. 13 you can remove the title.

- You could also plot the total area and compare it with the total area derived from the OSI-450 / OSI-430-b product. This would lend the figure more credibility.

Figure 15: The figure is cropped at the left side. Noted. Will be fixed for final version.

Like in the previous two figures the title can be removed.

L434: What is "maxima of total ice" --> please be more specific in your wording. What you mean are "maxima of the total sea ice concentration".

L455: "i" --> "in"

L457-459: This sentence should be merged with the next paragraph with (currently) point (4).

L468/469: One mentioning of "at 1.4 GHz" seems enough here.